# Truth over Tricks: Measuring and Mitigating Shortcut Learning in Misinformation Detection

**Herun Wan[1,2], Jiaying Wu[3], Minnan Luo[1,2,*], Zhi Zeng[1,4], Zhixiong Su[1,4]**

[1]School of Computer Science and Technology, Xi'an Jiaotong University, China
[2]Ministry of Education Key Laboratory of Intelligent Networks and Network Security, China
[3]National University of Singapore, Singapore
[4]Shaanxi Province Key Laboratory of Big Data Knowledge Engineering, China
wanherun@stu.xjtu.edu.cn  minnluo@xjtu.edu.cn
https://github.com/whr000001/TruthOverTricks

## Abstract

Misinformation detectors often rely on superficial cues (i.e., *shortcuts*) that correlate with misinformation in training data but fail to generalize to the diverse and evolving nature of real-world misinformation. This issue is exacerbated by large language models (LLMs), which can easily generate convincing misinformation using simple prompts. We introduce TRUTHOVERTRICKS, a unified evaluation paradigm for measuring shortcut learning in misinformation detection. TRUTHOVERTRICKS categorizes shortcut behaviors into intrinsic shortcut induction and extrinsic shortcut injection, and evaluates seven representative detectors across 14 popular benchmarks, along with two new factual misinformation datasets, NQ-Misinfo and Streaming-Misinfo. Empirical results reveal that existing detectors suffer severe performance degradation when exposed to both naturally occurring and adversarially crafted shortcuts. To address this, we propose the Shortcut Mitigation Framework (SMF), an LLM-augmented data augmentation framework that mitigates shortcut reliance through paraphrasing, factual summarization, and sentiment normalization. SMF consistently enhances robustness across 16 benchmarks, forcing models to rely on deeper semantic understanding rather than shortcut cues.

## 1 Introduction

Real-world misinformation, such as fake news [70] and rumors [36], appears in highly diverse forms. It varies in sentiment, writing style, and topical focus, and often adapts quickly to emerging events. The emergence of large language models (LLMs) further amplifies this diversity. With simple prompts such as "Rewrite the news article to make it more convincing", LLMs can generate persuasive misinformation that closely mimics credible content [5].

To ensure factual integrity in this dynamic online landscape, misinforma-

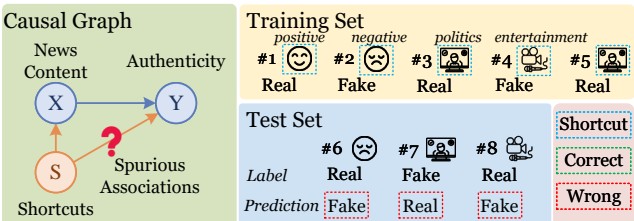

Figure 1: Illustration of shortcut learning in misinformation detection. As depicted in the causal graph, misinformation detectors often learn spurious associations between truthfulness and task-irrelevant features (i.e., shortcuts) during training. For instance, features such as *sentiment* or *domain* may lead detectors to infer that positive-toned or politics-related content is more likely to be real. As a result, models fail to generalize and produce incorrect predictions when presented with more realistic or diverse test instances.

---

[*]Corresponding author

39th Conference on Neural Information Processing Systems (NeurIPS 2025).

tion detection systems must generalize across a wide range of expressions. However, existing systems are typically trained on static datasets with limited variation in style and semantics. As a result, they tend to rely on *shortcuts* [12] – superficial cues that are correlated with misinformation labels in the training data but are not causally informative. This reliance leads to brittle predictions that fail to generalize to novel or intentionally disguised misinformation, as illustrated in Figure 1. Although recent work has begun to explore shortcut learning in this domain—analyzing surface features such as specific entity mentions [71] or publisher-related stylistic traits [59], these investigations are often narrow in scope, focusing on specific shortcut types or misinformation types. Fully addressing this challenge requires a systematic and comprehensive measurement framework that captures the full spectrum of shortcut behaviors across both naturally occurring and adversarially induced settings.

To this end, we propose TRUTHOVERTRICKS, a unified evaluation paradigm for diagnosing shortcut learning in misinformation detection systems. TRUTHOVERTRICKS is grounded in a novel taxonomy of misinformation shortcuts, which distinguishes between two key sources: (1) *intrinsic shortcut induction*, which captures shortcuts that arise naturally within existing benchmarks, and (2) *extrinsic shortcut injection*, which introduces adversarial variations crafted by LLMs to obscure misinformation and challenge model robustness. In the intrinsic setting, we systematically investigate whether four commonly used misinformation indicators – sentiment, style, topic, and perplexity – may inadvertently induce shortcut reliance. In the extrinsic setting, we curate six types of LLM-generated reframings that simulate realistic adversarial attacks on misinformation detectors.

With TRUTHOVERTRICKS, we evaluate seven detectors from three main categories of misinformation detectors (i.e., LLM-based, LM-based, and debiasing approaches) on 14 popular misinformation detection benchmarks. To assess whether models can detect misinformation that requires factual knowledge (for example, "The capital of Greece is Thessaloniki"), we construct two new benchmarks: NQ-Misinfo and Streaming-Misinfo. These are derived by distorting instances from two existing question answering (QA) benchmarks, Natural Questions (NQ) [26] and StreamingQA [31], to create factual misinformation examples. Empirical results show that current detectors experience significant and consistent performance degradation under both intrinsic shortcut induction and extrinsic shortcut injection, with accuracy dropping to 0% in the most challenging cases. Notably, misinformation detectors trained on sentiment-injected data can even obtain reasonable capability as a sentiment classifier, although the training objective differs significantly.

To redirect model attention from superficial shortcuts to deeper semantic signals, we propose the Shortcut Mitigation Framework (SMF), an LLM-augmented framework for mitigating misinformation shortcuts and promoting more generalizable detection behavior. To better capture the diversity of real-world misinformation, our framework enhances data variation through three complementary strategies: paraphrase generation, factual summarization, and sentiment normalization. These augmentation techniques encourage models to rely on robust, content-based reasoning rather than shortcut cues, thereby improving resilience to distributional shifts and adversarial manipulations. We integrate our framework with five representative misinformation classifiers and evaluate its effectiveness across all 16 benchmark datasets under extrinsic shortcut injection from the TRUTHOVERTRICKS suite. Empirical results show that our framework consistently improves model robustness against a wide range of externally injected shortcuts, demonstrating its effectiveness and general applicability.

## 2 TRUTHOVERTRICKS

TRUTHOVERTRICKS evaluates shortcut learning in misinformation detection through two complementary perspectives: (1) *intrinsic shortcut induction* and (2) *extrinsic shortcut injection*. An overview of the framework is shown in Figure 2.

### 2.1 Intrinsic Shortcut Induction

This component of TRUTHOVERTRICKS investigates the impact of shortcuts that naturally emerge within existing misinformation benchmarks. Specifically, we evaluate whether four commonly used misinformation indicators, namely *sentiment*, *style*, *topic*, and *perplexity*, may inadvertently introduce shortcut biases, despite their frequent use in enhancing model performance.

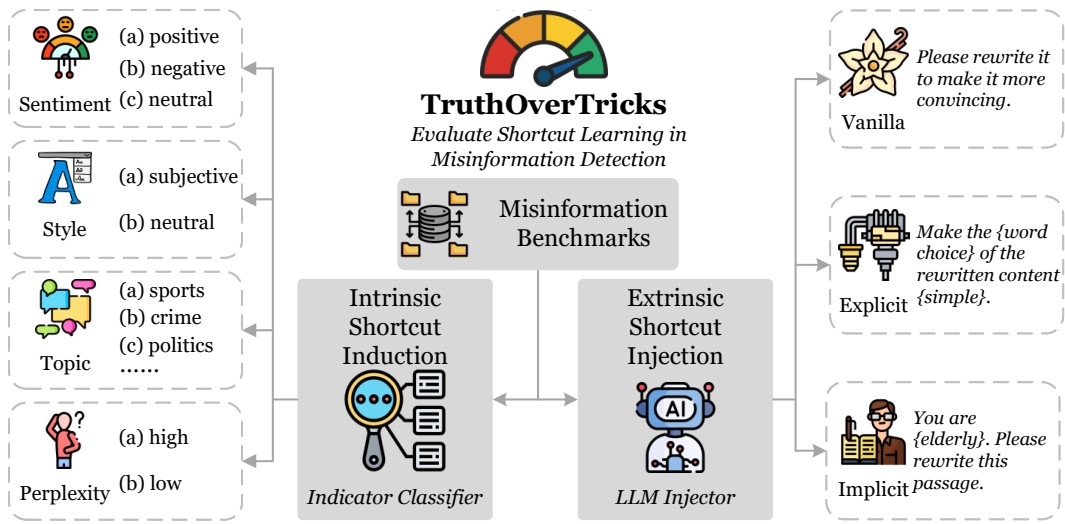

Figure 2: TRUTHOVERTRICKS evaluates misinformation detectors under two types of shortcuts: (1) *Intrinsic Shortcut Induction*, which captures spurious correlations that naturally emerge within existing benchmarks; and (2) *Extrinsic Shortcut Injection*, which introduces adversarially crafted variations from LLMs designed to obscure misinformation and challenge detector robustness.

- **Sentiment.** Prior work has shown that leveraging sentiment features can improve detection performance [69, 42]. However, we argue that sentiment may function as a shortcut, as the presence of negative or positive emotion does not causally determine whether content is true or false.

- **Style.** Misinformation is often more subjective in tone, while real news is presumed to be more neutral [23]. Yet in practice, especially on social platforms, real content can also adopt subjective styles to increase engagement [7], making stylistic cues a potentially misleading signal.

- **Topic.** Many detectors struggle to generalize across diverse topics or domains [29, 60], suggesting that models may overfit to topical patterns rather than learning content-based signals.

- **Perplexity.** Token distribution patterns, often captured via perplexity, are increasingly used to detect machine-generated content [39, 2]. This may cause detectors to learn surface-level language regularities rather than semantic truths.

To evaluate the shortcut-inducing potential of each indicator, we first train dedicated classifiers (e.g., to label sentiment as positive or negative), as detailed in Appendix D. We then create data splits where the joint distributions of the ground-truth misinformation label and the indicator label differ between training-validation and test sets. For example, the training-validation set may contain mostly negative-toned misinformation and positive-toned real news, whereas the test set reverses this pairing. This setting allows us to test whether detectors rely on shortcut cues that do not generalize. Details of the intrinsic setup and indicator statistics are provided in Appendix C and Appendix I, respectively.

## 2.2 Extrinsic Shortcut Injection

The advancement of LLMs introduces new risks in the form of LLM-based injection attacks. Malicious actors may exploit LLMs to generate misinformation that embeds superficial cues, or *shortcuts*, to evade existing detectors. To simulate such scenarios, TRUTHOVERTRICKS employs LLMs to explicitly or implicitly inject predefined shortcut-inducing factors into misinformation content.

We construct zero-shot prompts consisting of an *instruction text* $p_{inst}$ and *input texts* $p_{input}$ to guide the LLMs. The full list of prompts is presented in Table 5 (Appendix E.1), and representative outputs are shown in Table 15 (Appendix E.2). The evaluation setup mirrors the intrinsic shortcut setting, as detailed in Appendix C. We design three injection strategies:

- **Vanilla.** We adopt three prompts from [5]: *paraphrase*, *rewriting*, and *open-ended*, to rewrite texts in a general manner. Unlike the original setup, we rewrite both real and misinformation instances to avoid introducing confounding label-specific artifacts. This setting evaluates whether detectors overfit to generic LLM rewriting patterns. Here, $p_{input}$ includes the original text $s$.

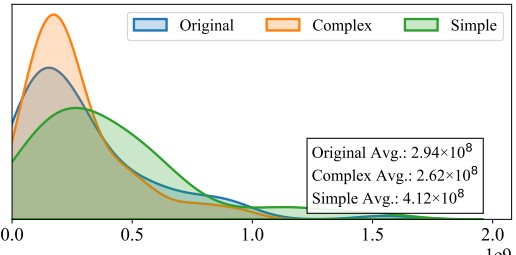
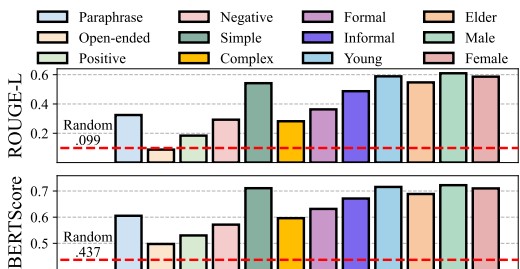

Figure 3: Token count distributions before and after **Word Choice** injection. The distinct distribution patterns and differences in average token counts show that LLMs can effectively manipulate word choices.

Figure 4: Token-level and semantic-level similarities of original-rewritten pairs. "Random" denotes that we shuffle the original-rewritten pairs as the baseline. Results show that rewritten texts preserve similarity to the originals.

- **Explicit.** This strategy instructs the LLM to incorporate predefined stylistic or linguistic attributes into the rewritten text. We select three such factors, with $p_{input}$ including both the original text $s$ and a factor description $f$: (i) **Sentiment**: *positive* and *negative*; (ii) **Tone**: *formal* and *informal*; and (iii) **Word Choice**: *simple* and *complex*.

- **Implicit.** Inspired by [1], this strategy prompts the LLM to simulate content authored by individuals with specific demographic traits. It evaluates whether detectors are sensitive to implicit author attributes. We consider two common dimensions, where $p_{input}$ includes the original text $s$ and author description $f$: (i) **Age**: *young* and *elder*; and (ii) **Gender**: *male* and *female*.

**Shortcut Injectors.** We leverage the open-source Meta-Llama-3-8B-Instruct coming from this link to inject shortcuts. The temperature is set to $\tau = 0$ to ensure reproducibility.

**LLMs can inject predefined factors without altering content authenticity.** As LLMs are known to suffer from hallucination [22], we first investigate whether they can reliably inject predefined shortcut-inducing factors. Taking **Word Choice** as an example, we use the infini-gram score [32] to assess token commonality and visualize token count distributions before and after injection in Figure 3. The clear distributional shift indicates that LLMs can effectively modify word usage patterns. Additional analyses for other shortcut types are provided in Appendix F.1. To further validate injection fidelity, we conduct a human evaluation with three expert annotators (Appendix F.2). For explicit shortcut injections, the average Cohen's Kappa between the annotated ground truth and expert judgments is 0.902, confirming high agreement. In contrast, implicit injections (e.g., simulating author age or gender) yield a much lower average Kappa of 0.035, indicating that LLMs struggle to simulate subtle author attributes. We also observe frequent refusals from the LLM to rewrite misinformation under implicit prompts (e.g., "I cannot rewrite the passage as it contains misinformation"), likely influenced by reinforcement learning from human feedback (RLHF) [41].

Next, we examine whether the rewritten outputs preserve authenticity. We compute ROUGE-L and BERTScore to assess token-level and semantic-level similarity, respectively, of original-rewritten pairs (Figure 4). Results show high similarity scores, substantially higher than a randomized baseline, suggesting that the meaning of the original content is largely preserved. Additionally, we assess detector performance on the rewritten datasets under the standard setting (Appendix F.3; Tables 24 and 25), where only slight performance differences are observed. A complementary human evaluation (Appendix F.4) confirms that authenticity is maintained post-injection, with an average accuracy of 91.2% and a Fleiss' Kappa of 0.838. We further employ an LLM-based evaluation approach similar inspirit to FactScore [38] (Appendix F.5), where GPT-4-turbo believes that on average 92.5% of the rewritten instances do not alter the authenticity.

These findings confirm the validity and effectiveness of LLM-based injection attacks, establishing them as a practical and credible threat to misinformation detectors.

### 2.3 Experiment Settings

**Detectors.** We apply TRUTHOVERTRICKS to evaluate seven misinformation detectors spanning three major categories:(i) **LLM-based Detectors**: We follow the prompting strategy of F3 [35]

Table 1: Accuracy of DEBERTA under four intrinsic shortcut settings. **Shortcut** refers to evaluation using TRUTHOVERTRICKS, while **Random** uses a randomly sampled training set of equal size to simulate the standard setting for fair comparison. **Difference** shows relative performance change; "–" indicates undefined values. We highlight (in blue color) the top three datasets with the largest performance drops per shortcut. Results show that existing detectors experience significant degradation when exposed to intrinsic shortcuts.

| DEBERTA | | D01 | D02 | D03 | D04 | D05 | D06 | D07 | D08 | D09 | D10 | D11 | D12 | D13 | D14 | D15 | D16 |
|---|---|---|---|---|---|---|---|---|---|---|---|---|---|---|---|---|---|
| **Sentiment** | Random | 0.0 | 89.7 | 85.0 | 75.0 | 88.9 | 100.0 | 90.0 | 47.6 | 57.1 | 68.6 | 96.1 | 82.0 | 100.0 | 100.0 | 90.0 | 50.0 |
| | Shortcut | 0.0 | 31.0 | 35.0 | 25.0 | 77.8 | 87.5 | 85.0 | 57.1 | 42.9 | 25.7 | 90.2 | 66.0 | 95.7 | 100.0 | 30.0 | 11.5 |
| | Difference | - | 65%↓ | 59%↓ | 67%↓ | 12%↓ | 12%↓ | 6%↓ | 20%↑ | 25%↓ | 63%↓ | 6%↓ | 20%↓ | 4%↓ | 0% | 67%↓ | 77%↓ |
| **Style** | Random | 60.0 | 85.4 | 90.6 | 88.2 | 100.0 | 100.0 | 92.9 | 82.5 | 71.4 | 63.0 | 100.0 | 50.0 | 98.4 | 88.9 | 46.7 | 97.8 |
| | Shortcut | 50.0 | 70.7 | 62.5 | 41.2 | 100.0 | 100.0 | 85.7 | 70.0 | 92.9 | 92.6 | 100.0 | 100.0 | 98.4 | 88.9 | 0.0 | 14.1 |
| | Difference | 17%↓ | 17%↓ | 31%↓ | 53%↓ | 0% | 0% | 8%↓ | 15%↓ | 30%↑ | 47%↑ | 0% | 100%↓ | 0% | 0% | 100%↓ | 86%↓ |
| **Topic** | Random | 81.8 | 79.7 | 73.9 | 78.9 | 63.2 | 95.5 | 100.0 | 88.2 | 80.7 | 55.7 | 100.0 | 84.1 | 100.0 | 98.7 | 82.1 | 48.9 |
| | Shortcut | 63.6 | 64.1 | 47.8 | 52.6 | 78.9 | 90.9 | 94.3 | 85.3 | 35.1 | 37.7 | 100.0 | 81.8 | 95.9 | 98.7 | 69.2 | 71.1 |
| | Difference | 22%↓ | 20%↓ | 35%↓ | 33%↓ | 25%↑ | 5%↓ | 6%↓ | 3%↓ | 57%↓ | 32%↓ | 0% | 3%↓ | 4%↓ | 0% | 16%↓ | 45%↑ |
| **Perplexity** | Random | 47.6 | 82.5 | 80.8 | 64.7 | 79.2 | 100.0 | 97.1 | 79.1 | 68.0 | 70.3 | 92.7 | 87.0 | 87.5 | 100.0 | 59.5 | 96.0 |
| | Shortcut | 9.5 | 42.1 | 30.8 | 41.2 | 83.3 | 96.9 | 77.1 | 25.6 | 38.0 | 34.4 | 87.3 | 45.7 | 87.5 | 98.6 | 42.9 | 26.0 |
| | Difference | 80%↓ | 49%↓ | 62%↓ | 36%↓ | 5%↑ | 3%↓ | 21%↓ | 68%↓ | 44%↓ | 51%↓ | 6%↓ | 48%↓ | 0% | 1%↓ | 28%↓ | 73%↓ |

to prompt MISTRALV3 [24] and LLAMA3 [14].(ii) **LM-based Detectors**: We adopt BERT [9] and DEBERTA [20], using frozen embeddings followed by multi-layer perceptrons (MLPs) for classification. (iii) **Debiasing Detectors**: We include three representative debiasing methods: CMTR [18], DISC [11], and CATCH [50]. These detectors collectively cover a wide spectrum of existing approaches. Detailed descriptions of all baselines are provided in Appendix H.

**Evaluation Benchmarks.** To comprehensively evaluate shortcut learning on real-world data, we use 14 popular misinformation benchmarks covering *fake news*, *rumors*, and *disinformation*, numbered **D01** through **D14** in order of publication: **RumourEval** [8], **Pheme** [4], **Twitter15**, **Twitter16** [36], **Celebrity**, **FakeNews** [44], **Politifact**, **Gossipcop** [51], **Tianchi** published [here] in 2022, **MultiLingual** [25], **AntiVax** [19], **COCO** [27], **Kaggle1** published [here] in 2024, and **Kaggle2** published [here] in 2024.

To assess whether detectors can identify factually incorrect claims, we introduce two new benchmarks for factual misinformation detection: **NQ-Misinfo** and **Streaming-Misinfo**, labeled as **D15** and **D16**, respectively. Inspired by [43], both are derived from existing question answering datasets: Natural Questions (NQ) [26] and StreamingQA [31]. We construct real claims by converting question–answer pairs into declarative sentences, and generate fake claims by distorting the original answers. Detailed construction steps are described in Appendix B.

Comprehensive dataset descriptions and statistics are provided in Appendix I.

# 3 Observations from TRUTHOVERTRICKS

## 3.1 Intrinsic Shortcut Induction

**Existing datasets contain shortcuts irrelevant to the authenticity.** We begin by examining the distributions of four commonly used indicators: sentiment, style, topic, and perplexity, between misinformation and non-misinformation. Figure 5 shows the distributions for **Pheme** as an example, while results for all datasets are provided in Figure 14 and Table 17 in Appendix J.1. The distributions of fake and real news are largely similar across these factors, suggesting that they are not causally linked to authenticity. In other words, detectors should not rely on such attributes—for instance, assuming that a news item with negative sentiment is fake, or one with positive sentiment is real.

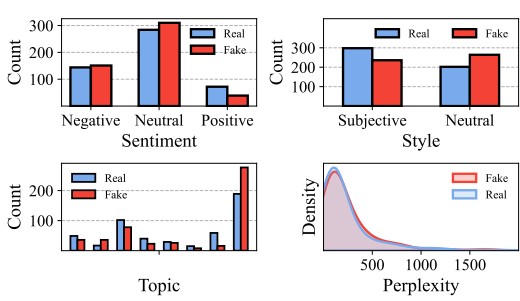

Figure 5: Distributions of sentiment, style, topic, and perplexity for fake and real instances in **Pheme**. The similarity between distributions suggests that these factors are not causally linked to authenticity.

Table 2: Average accuracy of baseline detectors on 16 datasets under various LLM-generated extrinsic shortcuts. "Original" denotes performance under the standard training setting. We report relative accuracy changes compared to the original. Results show that LLMs can successfully inject predefined factors that induce shortcuts, leading to a performance drop (in blue color) in trainable detectors. Explicit shortcut injection is significantly more harmful than implicit injection. In contrast, LLM-based detectors remain robust, motivating our mitigation approach.

| Shortcuts | | LLM-based Detectors | | LM-based Detectors | | Debiasing Detectors | | |
| --- | --- | --- | --- | --- | --- | --- | --- | --- |
| | | MISTRALV3 | LLAMA3 | BERT | DEBERTA | CMTR | DISC | CATCH |
| Original | | 53.9 | 58.8 | 78.1 | 80.2 | 76.9 | 78.4 | 78.2 |
| Vanilla | | 52.6 (2.3% ↓) | 54.8 (6.9% ↓) | 17.3 (77.8% ↓) | 11.2 (86.0% ↓) | 22.9 (70.3% ↓) | 14.9 (81.0% ↓) | 16.9 (78.3% ↓) |
| Explicit | Sentiment | 51.9 (3.6% ↓) | 52.6 (10.6% ↓) | 9.1 (88.4% ↓) | 19.2 (76.0% ↓) | 10.7 (86.1% ↓) | 8.9 (88.7% ↓) | 17.0 (78.2% ↓) |
| | Tone | 54.3 (0.7% ↑) | 56.4 (4.2% ↓) | 8.8 (88.8% ↓) | 6.7 (91.6% ↓) | 10.8 (86.0% ↓) | 9.0 (88.5% ↓) | 14.9 (80.9% ↓) |
| | Word Choice | 54.3 (0.8% ↑) | 56.9 (3.4% ↓) | 7.0 (91.0% ↓) | 6.4 (92.0% ↓) | 8.9 (88.4% ↓) | 6.8 (91.3% ↓) | 16.1 (79.5% ↓) |
| Implicit | Age | 54.5 (1.1% ↑) | 53.8 (8.5% ↓) | 73.4 (6.0% ↓) | 74.6 (6.9% ↓) | 73.6 (4.4% ↓) | 73.7 (5.9% ↓) | 73.1 (6.5% ↓) |
| | Gender | 53.0 (1.6% ↓) | 53.6 (8.9% ↓) | 74.3 (4.9% ↓) | 74.0 (7.7% ↓) | 73.6 (4.3% ↓) | 74.8 (4.6% ↓) | 74.6 (4.6% ↓) |

**Existing detectors suffer from intrinsic shortcut learning.** We report the performance of DE-BERTA under both the intrinsic shortcut setting and a random control setting in Table 1. Results for all detectors are presented in Table 18 (Appendix J.1). We observe substantial performance degradation under the shortcut setting, with some detectors producing systematically incorrect predictions. This indicates that models may incorrectly associate shortcut features with truthfulness.

**Shortcut effects vary across datasets.** In certain datasets, the distributions of specific indicators differ notably between misinformation and non-misinformation, and performance does not always decline—occasionally it even improves. This suggests that in some cases, shortcut features may coincidentally align with authenticity, reinforcing the model's reliance on them. These indicate that detectors are prone to associating auxiliary features with authenticity, regardless of whether such associations are spurious or valid. This validates the need to further investigate the risks posed by extrinsic shortcut injection.

## 3.2 Extrinsic Shortcut Injection

We report the average accuracy of all detectors on 16 datasets under the standard training setting and under the TRUTHOVERTRICKS evaluation in Table 2. Full per-dataset results are provided in Tables 19 and 20 in Appendix J.2. Detailed experiment settings are provided in Appendix J.

**LLM-based detectors are robust to shortcuts.** While LLMs without fine-tuning are generally limited in their ability to detect misinformation, we observe that they are substantially more robust to shortcut-based attacks. This suggests that LLMs are less prone to relying on superficial correlations and can instead capture deeper semantic signals. These findings support our motivation to use LLMs in SMF (proposed in §4) to rewrite data in a way that removes shortcut cues while preserving authenticity.

**Explicit injection attacks severely degrade detector performance.** Compared to the standard setting, performance under explicit shortcut injection drops by an average of approximately 80%, with the most extreme case reaching a 92.0% drop. Even under the *Vanilla* rewriting strategy, which alters prompt formats without explicitly targeting shortcut attributes, detectors exhibit notable performance degradation. While debiasing methods outperform standard LM-based detectors, they too experience substantial drops. These results indicate that addressing shortcut reliance purely through model design is insufficient, reinforcing the need for data-centric solutions such as our LLM-based data augmentation framework introduced in §4.

**Detectors are less affected by implicit injection.** In contrast to explicit shortcut attacks, implicit injection leads to minimal performance degradation. This aligns with our earlier findings in §2.2, where we observed that LLMs struggle to reliably simulate author-specific attributes such as age or gender. Consequently, the injected features are less effective in inducing shortcut learning. It's notable that in Table 2 we retained instances even when the LLMs partially or fully refused to follow the injection prompt. This choice preserved label distributions and ensured a consistent dataset size

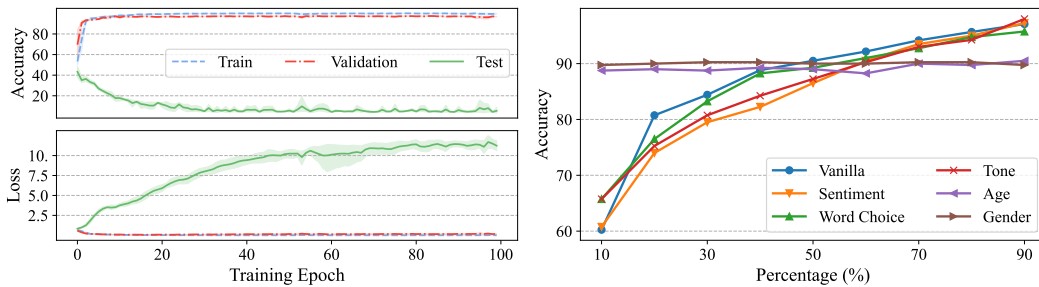

Figure 6: Training loss and accuracy of DE-BERTA on **AntiVax** under the shortcut setting, averaged over five runs. Performance degrades over time, indicating that the detector increasingly overfits to shortcuts rather than learning authenticity.

Figure 7: Performance of DEBERTA on **An-tiVax** as the ratio of shortcut-to-authenticity associations increases. Accuracy rises sharply with shortcut prevalence, approaching 100% when the ratio nears 100%, indicating strong reliance on shortcuts.

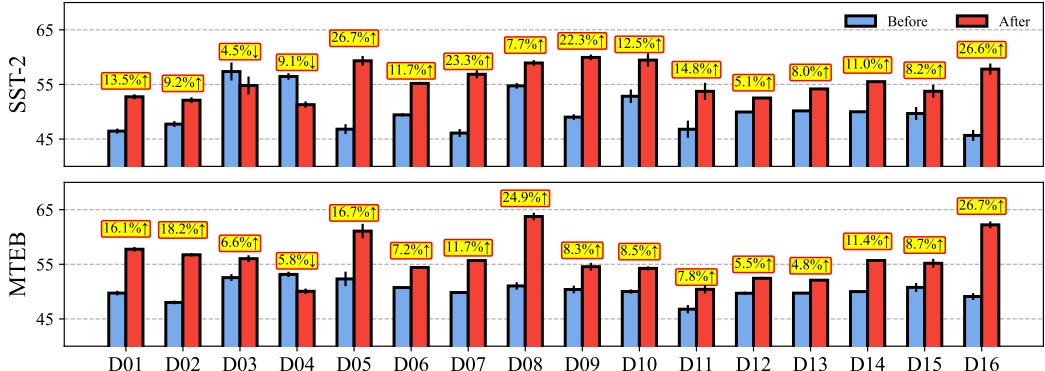

Figure 8: Performance of DEBERTA on two sentiment classification datasets before and after training under the **Sentiment** injection attack. Accuracy improvements indicate that detectors mistakenly learn sentiment as a proxy for authenticity, confirming the presence of shortcut learning.

across original and shortcut conditions, allowing for a direct comparison. To rigorously assess the influence of these refusals, we conducted an additional analysis in Appendix K.

**Misinformation detectors readily learn shortcuts.** Previous results demonstrate that detector performance drops significantly under shortcut settings. To further investigate what detectors actually learn during training in such conditions, we analyze their learning dynamics and transfer behavior.

We begin by tracking training loss and test accuracy under the shortcut setting, as shown in Figure 6. Test performance deteriorates as training progresses, indicating that detectors overfit to spurious correlations rather than learning authenticity. To examine what information the models acquire, we evaluate detectors trained under the **Sentiment** injection setting on two external sentiment classification datasets: SST-2 [53] and MTEB [40]. As shown in Figure 8, accuracy on these sentiment tasks increases by up to 26.7% after shortcut-based training, suggesting that detectors trained on sentiment-injected misinformation acquire nontrivial sentiment classification ability.

We also study how performance varies as the ratio of shortcut-to-authenticity associations increases, with results presented in Figure 7. As this ratio rises, detector accuracy improves dramatically, approaching 100% in most settings except for implicit injection, where LLMs struggle to simulate author attributes. These findings confirm that detectors learn to rely on shortcut features, even when such associations are spurious or non-causal.

Overall, these results indicate that detectors do not learn to assess authenticity per se. Instead, they internalize correlations between shortcut features and labels, regardless of whether these associations reflect genuine truthfulness.

Table 3: Average accuracy of baseline detectors on 16 datasets under extrinsic shortcut settings, with and without data augmentation from our proposed shortcut mitigation framework (SMF). "w/o aug." denotes original performance under shortcut attacks. We report relative improvements and highlight (in blue color) the best results. SMF effectively mitigates shortcut learning, with different variants excelling under different types of injected shortcuts.

| Shortcut | Augmentation | LM-based Detectors | | Debiasing Detectors | | |
|---|---|---|---|---|---|---|
| | | BERT | DeBERTa | CMTR | DisC | CATCH |
| **Vanilla** | w/o aug. | 17.3 | 11.2 | 22.9 | 14.9 | 16.9 |
| | Paraphrase | 38.8 (123.7% ↑) | 30.2 (169.0% ↑) | 43.2 (89.0% ↑) | 38.6 (159.9% ↑) | 43.0 (153.7% ↑) |
| | Summary | 44.1 (154.4% ↑) | 36.2 (222.5% ↑) | 47.3 (106.8% ↑) | 44.2 (197.0% ↑) | 49.0 (189.4% ↑) |
| | Neutral | 29.9 (72.2% ↑) | 21.3 (89.4% ↑) | 35.7 (56.0% ↑) | 28.1 (88.9% ↑) | 25.7 (51.9% ↑) |
| **Sentiment** | w/o aug. | 9.1 | 19.2 | 10.7 | 8.9 | 17.0 |
| | Paraphrase | 14.7 (62.0% ↑) | 22.7 (18.0% ↑) | 18.0 (68.6% ↑) | 14.4 (62.8% ↑) | 23.9 (40.6% ↑) |
| | Summary | 40.0 (340.7% ↑) | 45.4 (136.7% ↑) | 43.2 (304.1% ↑) | 40.0 (350.8% ↑) | 51.0 (199.4% ↑) |
| | Neutral | 34.9 (284.6% ↑) | 44.0 (129.4% ↑) | 40.0 (274.3% ↑) | 36.1 (307.3% ↑) | 47.6 (179.4% ↑) |
| **Tone** | w/o aug. | 8.8 | 6.7 | 10.8 | 9.0 | 14.9 |
| | Paraphrase | 58.6 (567.3% ↑) | 52.5 (679.3% ↑) | 62.6 (479.9% ↑) | 55.3 (516.0% ↑) | 62.9 (321.0% ↑) |
| | Summary | 55.9 (536.7% ↑) | 48.6 (621.9% ↑) | 58.4 (441.2% ↑) | 56.4 (528.2% ↑) | 63.1 (322.8% ↑) |
| | Neutral | 57.1 (549.5% ↑) | 50.6 (651.9% ↑) | 59.8 (453.5% ↑) | 57.4 (539.4% ↑) | 60.5 (304.9% ↑) |
| **Word Choice** | w/o aug. | 7.0 | 6.4 | 8.9 | 6.8 | 16.1 |
| | Paraphrase | 63.4 (800.5% ↑) | 56.9 (783.1% ↑) | 65.3 (633.6% ↑) | 63.9 (838.4% ↑) | 67.7 (321.6% ↑) |
| | Summary | 45.2 (542.0% ↑) | 33.6 (421.4% ↑) | 47.8 (437.6% ↑) | 45.7 (570.2% ↑) | 57.5 (258.1% ↑) |
| | Neutral | 49.1 (598.1% ↑) | 41.5 (542.8% ↑) | 52.9 (494.3% ↑) | 49.7 (629.3% ↑) | 58.9 (266.9% ↑) |
| **Age** | w/o aug. | **73.4** | **74.6** | **73.6** | **73.7** | **73.1** |
| | Paraphrase | 71.1 (3.1% ↓) | 71.9 (3.7% ↓) | 71.4 (3.0% ↓) | 71.7 (2.7% ↓) | 71.5 (2.2% ↓) |
| | Summary | 71.7 (2.3% ↓) | 69.5 (6.9% ↓) | 72.4 (1.5% ↓) | 72.1 (2.3% ↓) | 71.0 (2.9% ↓) |
| | Neutral | 71.8 (2.2% ↓) | 72.2 (3.2% ↓) | 72.4 (1.6% ↓) | 73.3 (0.6% ↓) | 72.6 (0.8% ↓) |
| **Gender** | w/o aug. | **74.3** | **74.0** | **73.6** | **74.8** | **74.6** |
| | Paraphrase | 71.5 (3.8% ↓) | 71.9 (2.8% ↓) | 70.6 (4.0% ↓) | **72.3 (3.2% ↓)** | 71.8 (3.7% ↓) |
| | Summary | 70.5 (5.2% ↓) | 69.8 (5.6% ↓) | 71.3 (3.1% ↓) | 71.1 (5.0% ↓) | 71.2 (4.5% ↓) |
| | Neutral | 71.1 (4.4% ↓) | 72.8 (1.6% ↓) | 70.7 (3.9% ↓) | 72.3 (3.3% ↓) | 71.9 (3.6% ↓) |

**Shortcut learning erodes detector credibility.** A robust misinformation detector must not only make accurate predictions but also provide well-calibrated confidence estimates to support reliable content moderation. We employ the expected calibration error (ECE) [17] to quantify the calibration of misinformation detectors. Figure 9 shows the calibration curves of detectors before and after shortcut injection. Calibration degrades significantly for most detectors under attack, except for **LLM-based detectors**, which remain relatively stable. Moreover, we observe that detector confidence increases even as prediction quality declines, further confirming that detectors are learning shortcuts rather than semantic authenticity.

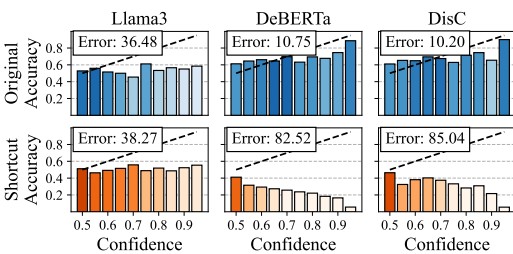

Figure 9: Calibration of detectors before and after explicit injection attacks. "Error" indicates the expected calibration error ($\times 100$), where lower values denote better calibration. Results show that explicit injection attacks substantially degrade calibration quality.

# 4   Shortcut Mitigation Framework (SMF)

Our observations from TRUTHOVERTRICKS reveal that misinformation detectors often rely on spurious shortcut features rather than learning authentic signals, which not only harms generalization but also degrades calibration and interpretability. To address this, we introduce the Shortcut Mitigation Framework (SMF), a simple yet effective data-centric approach designed to reduce shortcut reliance and enhance model robustness.

## 4.1   Methodology

Given that LLMs can inject shortcut-inducing features, we hypothesize they can also be leveraged to remove factors irrelevant to authenticity. To this end, we propose SMF, an LLM-based data augmentation method that rewrites texts prior to both training and inference. SMF is model-agnostic

and can be applied to any detector. We design three variants and provide their prompts in Table 6 (Appendix G.1) and example outputs in Table 16 (Appendix G.2).

- **Paraphrase.** We reuse the same prompt as the *paraphrase* variant from the **Vanilla** setting in the extrinsic shortcut injection.

- **Summary.** We ask LLMs to summarize a given text and remove irrelevant factors. After summarizing, we expect the text will only contain information about authenticity.

- **Neutral.** This variant targets a specific shortcut (e.g., sentiment). Since LLMs can inject a desired factor, we ask them to overwrite an existing shortcut by injecting a neutral alternative (e.g., neutral sentiment). This re-balances distributional mismatches between training and test splits, potentially mitigating shortcut reliance.

### 4.2 Results

We first verify that SMF can successfully remove injected shortcuts, with supporting results provided in Appendix L.1. Table 3 presents the average accuracy of detectors across 16 datasets with and without SMF augmentation under shortcut injection. Full results for each variant are available in Tables 21, 22, and 23 (Appendix L.2). We exclude LLM-based detectors from this evaluation in the main text since they are already robust to shortcut attacks, and we provide the results of LLM-based detectors in Appendix M.

**SMF significantly improves detection performance.** All three augmentation variants lead to notable performance gains, with improvements reaching up to 838.4% under explicit injection attacks. This demonstrates the effectiveness of data-centric mitigation over model-side adjustments. Debiasing detectors outperform standard LM-based ones, and combining debiasing with SMF further enhances robustness. Notably, performance decreases slightly under implicit injection, likely because LLM rewriting may subtly alter authenticity when no shortcut is present.

**Each variant excels against specific shortcut types.** The **Paraphrase** variant performs best against the **Word Choice** shortcut, while **Summary** consistently yields the highest overall improvement. These results suggest that summarizing texts to retain only authenticity-relevant content is a particularly effective mitigation strategy.

**Combining SMF variants further enhances robustness.** Since each SMF variant targets different shortcut types, we explore whether combining them yields additive benefits. As shown in Table 4, joint augmentation leads to further performance gains up to 54.7% improvement, demonstrating that integrated augmentation can more comprehensively remove diverse shortcut cues.

Table 4: Performance of DEBERTA on the **Anti-Vax** dataset using single vs. multiple data augmentations. Combining multiple augmentations further mitigates shortcut learning and leads to improved accuracy.

| Shortcuts | Neutral | Neutral+Summary | Summary | Summary+Neutral |
|---|---|---|---|---|
| Vanilla | 23.2 | 34.2 (47.3% ↑) | 31.8 | 33.1 (3.9% ↑) |
| Sentiment | 44.8 | 69.2 (54.7% ↑) | 58.0 | 66.8 (15.1% ↑) |
| Tone | 73.0 | 76.8 (5.1% ↑) | 70.8 | 79.0 (11.7% ↑) |
| Word Choice | 62.0 | 76.0 (22.6% ↑) | 57.2 | 74.2 (29.7% ↑) |

## 5 Related Work

Shortcut learning refers to the phenomenon where neural models achieve high performance on standard benchmarks by exploiting spurious correlations, rather than learning the underlying task [12, 37]. In the context of misinformation detection, we argue that detectors often rely on superficial features (e.g., sentiment, source, or style) rather than identifying factual authenticity. Prior work has examined various forms of bias in this domain, including entity bias [71, 67], source bias [54, 33], and domain bias [52, 66]. However, a systematic evaluation of shortcut learning in misinformation detection remains largely unexplored. This paper aims to fill that gap by providing a comprehensive evaluation paradigm and mitigation framework. Additional related work on misinformation detection is discussed in Appendix A to further contextualize our contributions.

# 6  Conclusion

We propose TRUTHOVERTRICKS, a unified evaluation paradigm for diagnosing shortcut learning in misinformation detection systems. It encompasses two key components: *intrinsic shortcut induction*, which captures spurious correlations naturally present in existing benchmarks, and *extrinsic shortcut injection*, which introduces adversarially crafted shortcut variations using LLMs. Using TRUTHOVERTRICKS, we evaluate seven representative detectors across 16 misinformation detection benchmarks. Our findings reveal that current detectors suffer substantial performance degradation when exposed to both shortcut types. To address this, we introduce SMF, a unified LLM-based data augmentation framework designed to mitigate shortcut learning. SMF significantly improves robustness across diverse shortcut types, demonstrating the effectiveness of data-centric mitigation strategies in enhancing generalization and authenticity awareness in misinformation detection.

## Acknowledgements

This work is supported in part by the National Natural Science Foundation of China (No. 62192781, No. 62272374), the Natural Science Foundation of Shaanxi Province (No. 2024JC-JCQN-62), the State Key Laboratory of Communication Content Cognition under Grant No. A202502, the Key Research and Development Project in Shaanxi Province (No. 2023GXLH-024). This research is supported by the Ministry of Education, Singapore, under its MOE AcRF TIER 3 Grant (MOE-MOET32022-0001). The China Scholarship Council also supports this research.

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

## Limitation

This paper aims to explore shortcut learning in misinformation detection systematically. Specifically, we focus on exploring shortcut learning in textual content-based misinformation detection with datasets including textual content and detectors analyzing texts. There are detectors employing additional information, such as social context, where the additional information may also induce shortcuts.

We have tried our best to cover mainstreaming misinformation datasets and detectors (including 16 datasets and 7 detectors). However, as proven in the main text, existing datasets contain diverse and various shortcuts, and LLMs could inject numerous shortcuts. It is impossible to cover every shortcut, every injection attack, and every mitigation method. Thus, we explore four representative shortcuts in existing datasets, six representative attack methods, and three representative mitigation methods.

## Ethics Statement

Identifying misinformation on social platforms ensures online safety. This paper aims to explore shortcut learning in misinformation detection. We have proven that detectors suffer significant performance drops under the shortcut setting and have designed six LLM injection attacks. The research findings and resources inevitably increase the risks of dual use. We aim to mitigate such dual use by employing controlled access to our research data, trying our best to ensure that the data is only employed for research purposes. Meanwhile, our findings illustrate that detectors are vulnerable under the shortcut setting. Thus, we argue that the decision of automatic detectors should be considered as an initial screen, while content moderation decisions should be made with related experts.

Meanwhile, we employ LLMs to inject predefined factors into texts, while we do not directly employ LLMs to generate misinformation. We also argue that LLMs should not be employed to generate misinformation, and researchers should make an effort to limit the misuse of LLMs. Meanwhile, due to the inherent social bias and hallucinations of LLMs, the texts after injection inevitably contain biased content. We emphasize that the data can only be used for research purposes.

## A  Related Work

**Misinformation Detection** Identifying *misinformation* (*fake news*, *disinformation*, *rumor*, *etc.*) is critical to ensure cybersecurity. The most common misinformation detectors analyze news content including text [61, 13], images [67, 68], or videos [56, 3] to identify misinformation. They also extract additional information such as sentiments [34], topics [48], or patterns [10] to enhance performance. However, we argue that these factors will induce shortcuts, where detectors may wrongly associate these factors with authenticity. Only few works have explored the shortcuts (biases) in misinformation detection as discussed in §5. Thus, this paper systematically explores shortcut learning in misinformation detection, bridging the gaps.

Besides analyzing the news content, detectors would employ social context (evidence) [15, 6], such as user profiles [55, 62], user reactions [63], or news environments [49, 65], to enhance detection performance. However, this paper does not focus on exploring this type of detector because it has various irrelevant factors (in content or social context), and we do not aim to improve the performance.

Recently, the advances of LLMs have brought opportunities and risks for misinformation detection. LLMs can be employed to enhance misinformation detection [16, 21, 58, 64]. On the other hand, LLMs can be leveraged to generate misinformation [5] or attack existing detectors [59, 57]. Thus, faced with the misuse of LLMs, we explored LLM injection attacks for shortcut learning in this paper.

Meanwhile, numerous datasets [29, 45, 30] have been proposed to support the development of misinformation detectors. However, these datasets are not designed for shortcut learning. This paper has explored LLM-based injection attacks, providing useful resources to promote the development of the misinformation detector against shortcut learning.

# B NQ-Misinfo and Streaming-Misinfo

**Dataset Construction**   Existing misinformation datasets are mostly written by humans, containing numerous factors not causally related to authenticity. For example, authors might express their sentiments or attitudes in the news, which might cause the wrong associations between authenticity and factors. Thus, we propose two misinformation datasets that try to exclude other factors. Namely, detectors should learn fact-related knowledge to conduct the right identifications on the datasets.

Inspired by [43], we employ two question-answering datasets, **NQ** [26] and **Streaming** [31] to construct the novel misinformation datasets.

An instance from the question-answering datasets contains a question $q$ and a correct answer $a_r$. To construct misinformation, we employ LLMs (Meta-Llama-3-8B-Instruct) to generate the wrong answer $a_w$. We prompt LLMs in a zero-shot format using the following prompt from [43]:

```
Generate a false answer to the given question.  It should be of short (less
than five words in general) and look plausible, compared to the reference
answer.
Question:  Question

Reference Answers:   Answer
```

So far, we have obtained a correct pair $(q, a_r)$ and a wrong pair $(q, a_w)$. Next, we construct misinformation and real ones based on these pairs. Unlike [43] who employ LLMs to only generate misinformation, we generate misinformation and real ones to avoid other factors (especially hallucinations that come from LLMs) based on rules. We first filter in questions beginning with "When", "Where", "What", and "Who". We then fill in answers (correct or wrong) in questions and rewrite questions into declarative sentences (real or misinformation) based on grammar rules. For example, we transfer ("*When was the last time anyone was on the moon?*", "*December 1972*") into "*The last time anyone was on the moon in December 1972.*" Sentences generated based on rules inevitably make grammar mistakes, thus we employ CoEdIT [47] to fix grammatical errors with prompt "*Fix grammatical errors in this sentence:* "[1]. We process the above example and obtain "*The last time anyone was on the moon was in December 1972.*"

**Dataset Analysis**   We present the basic statistics of these two datasets in Appendix I.

We further employ AutoBencher [28] to quantitatively evaluate the proposed datasets, where we employ *Difficulty* and *Separability* metrics. We consider the selected detectors as $\mathcal{M}$ and **NQ-Misinfo** and **Streaming-Misinfo** as $\mathcal{D}_c$. For **NQ-Misinfo**, the *Difficulty* is 0.340 and *Separability* is 0.025; For **Streaming-Misinfo**, the *Difficulty* is 0.385 and *Separability* is 0.028.

**Detector Performance**   Although existing detectors achieve remarkable performance on standard benchmarks, they have the following two limitations, which inspire us to explore shortcut learning in misinformation detection.

We first evaluate existing detectors on our proposed datasets **NQ-Misinfo** and **Streaming-Misinfo** and present the results in Figure 10. Compared to the standard dataset **AntiVax**, detectors can not successfully identify misinformation in datasets requiring fact-related knowledge, where the performance drops by up to 39.8%. Meanwhile, detector performance on the datasets is only slightly higher than random guessing (50%). Thus, we can conclude that existing detectors cannot identify misinformation requiring fact-related knowledge.

To further evaluate the generalization ability of existing detectors, we train DEBERTA on a specific dataset and evaluate it on another, where Figure 11 presents the results. It illustrates that the generalization ability of existing detectors is limited. We speculate that detectors do not learn the authenticity but learn the associations between authenticity and other factors. Thus, when the joint distributions of authenticity and other factors change (other datasets), detectors will make the wrong judgments.

# C TRUTHOVERTRICKS Settings

---

[1]Model checkpoints come from this link.

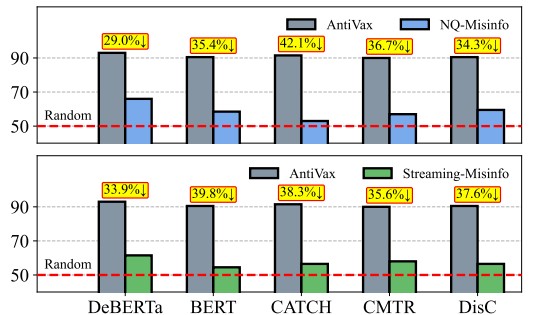

Figure 10: The performance of existing detectors on a standard dataset **AntiVax** and our proposed datasets **NQ-Misinfo** and **Streaming-Misinfo**. Detectors fail to achieve acceptable performance on datasets requiring fact-related knowledge and are only slightly better than random guessing.

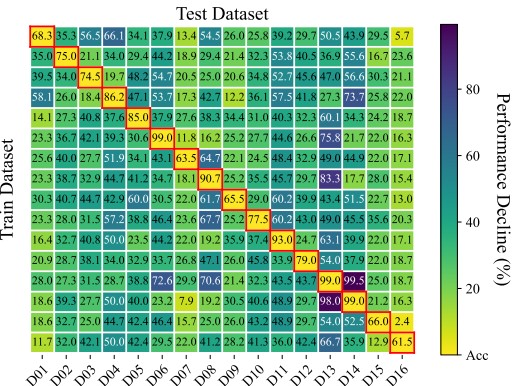

Figure 11: The performance of DEBERTA under out-of-dataset settings. We train the detector on a specific dataset and evaluate it on another. Detectors suffer from a significant performance drop.

Here we formally define the standard setting of misinformation detection and the shortcut settings, where Figure 12 presents an overview of them.

**Misinformation Detection** In this paper, we focus on content-based misinformation detection. Namely, the data instance is formalized as $d = \{t, y\}$, where $t$ denotes the textual content and $y \in \{0, 1\}$ denotes the misinformation label (0 for real and 1 for fake). The misinformation detector $f(\cdot)$ with learnable parameters $\theta$ aims to judge a piece of text $\hat{y} = f(t \mid \theta)$, where $\hat{y}$ denotes the prediction, and $\theta$ is optimized to maximize the detection performance.

**Standard Setting** For each dataset $\mathcal{D} = \{d_i = \{t_i, y_i\}\}_{i=1}^{n}$, we first randomly partition the dataset into training, validation, and test sets $\mathcal{D}_{training}$, $\mathcal{D}_{validation}$, and $\mathcal{D}_{test}$. Due to the random partitioning, the data distributions in each set are similar. Then we optimize detectors on the train-

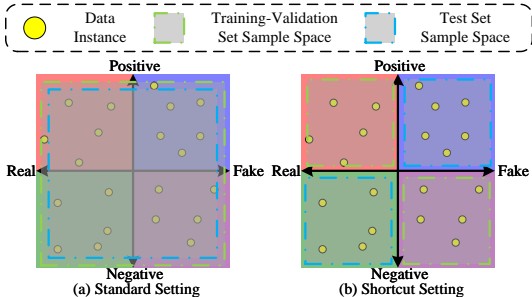

Figure 12: The overviews of standard setting and the shortcut setting. The horizontal axis represents authenticity, while the vertical axis represents other factors (taking *positive* and *negative* sentiment as an example). The main difference is that the shortcut setting induces other factors, and the joint distribution of authenticity and other factors is different between the training-validation set and the test set.

ing set, select the best detector statements on the validation set, and report the performance on the test set.

**Intrinsic Shortcut Induction** Here we first introduce the other factor (sentiment, style, topic, perplexity) $e$ into each data instance $d = \{t, e, y\}$. We then partition the dataset into training, validation, and test sets based on the joint distribution $P(e, y)$. The joint distribution of the training set is similar to the validation set but different from the test set. For example, the training-validation sets contain instances with $\{e = 0, y = 0\}, \{e = 1, y = 1\}$, but the test set contains instances with $\{e = 1, y = 0\}, \{e = 0, y = 1\}$. In this setting, the factor $e$ is obtained by classifiers introduced in Appendix D.

**Extrinsic Shortcut Injection** The only difference from intrinsic induction setting is that we employ LLMs to inject predefined factor $e$ into each instance $d$. Namely, in intrinsic shortcut induction, we group instances into $d_0 = \{t, e = 0, y\}$ and $d_1 = \{t, e = 1, y\}$, but here, we generate instance

$\tilde{d}_0 = \{\tilde{t}_0, e = 0, y\}$ and $\tilde{d}_1 = \{\tilde{t}_1, e = 1, y\}$ base on $d = \{t, e, y\}$. The rewritten $\tilde{t}$ is rewritten by LLMs based on $t$, and we have proven that the authenticity $y$ does not alter after rewriting.

**SMF Setting**   The only difference from the extrinsic injection setting is that we employ an LLM $\mathcal{G}$ to augment each text $\tilde{t}$ into $\hat{t} = \mathcal{G}(\tilde{t})$.

# D    Misinformation Indictor Classifiers

To explore potential shortcuts in existing misinformation datasets, we select sentiment, style, topic, and perplexity, four factors. These factors are widely used to enhance detection performance on standard benchmarks, thus we argue they are likely to induce shortcut learning.

**Sentiment**   We employ a five-class sentiment classifier to identify the sentiments in texts[2], where the labels include *very negative*, *negative*, *neutral*, *positive*, and *very positive*. We then group *very negative* and *negative* into *negative*, and group *very positive* and *positive* into *positive*, obtaining three classes. We set $e = 0$ for *negative*, set $e = 1$ for *positive*, and ignore *neutral* in the shortcut setting.

**Style**   We employ a binary classifier[3] to classify texts into *subjective* and *neutral*. We set $e = 0$ for *subjective* and $e = 1$ for *neutral* in the shortcut setting.

**Topic**   We employ an eight-class topic classifier to obtain the topics of texts, where the labels include "*sports*", "*arts, culture, and entertainment*", "*business and finance*", "*health and wellness*", "*lifestyle and fashion*", "*science and technology*", "*politics*", and "*crime*". We set $e = 0$ for the first 4 topics and $e = 1$ for others.

**Perplexity**   We employ GPT2 [46] to calculate the perplexity of each text[4]. We set $e = 0$ for texts with perplexity in the top 50% and $e = 1$ for others.

# E    Extrinsic Shortcut Injection

## E.1    Prompts

We provide the prompts we employed to inject predefined factors in Table 5.

## E.2    Case Study

We provide some cases of texts before and after injection attacks in Table 15. We also highlight the parts showing the predefined factors. It qualitatively illustrates that LLMs can explicitly inject other factors but struggle to implicitly inject factors.

- **Vanilla**. **Paraphrase** is the most faithful strategy, which just paraphrases the sentence without changing or adding any details. **Rewriting** strategy sometimes will add some fabricated details and **Open-ended** strategy will add a lot of fabricated details. The results in Figure 4 coincide with this conclusion.
- **Sentiment**. These two strategies do not change authenticity. However, LLMs would add some words with obvious emotional tendencies, such as "*surprise*" for **positive** and "*absolutely disgusting*" for **negative**.
- **Tone**. Like **Sentiment**, these strategies will change the tone of texts without changing authenticity.
- **Word Choice**. LLMs can employ simple (common) words for **Simple**, such as "*plane*" (with the infini-gram count of 52,239,855), and employ complex (uncommon) words for **Complex**, such as "*airliner*" (with the infini-gram count of 1,455).
- **Age** and **Gender**. These implicit strategies cannot present the characteristics of the author profiles. Although there are slight differences, we cannot distinguish the specific attributes.

---

[2]Model checkpoints come from this link.

[3]Model checkpoints come from this link.

[4]Model checkpoints come from this link.

Table 5: The prompts to inject predefined factors. We `highlight` the special parts of each prompt, where highlighted parts illustrate the main motivation behind each strategy.

| Shortcut | Instruction Text $p_{inst}$ |
|---|---|
| **Vanilla $p_{input}$**: *information content $s$.* | |
| **Rewriting** | *Given a passage, Please* `rewrite` *it to make it more convincing. The content should be the same. The style should be serious, calm and informative. The passage is: $s$* |
| **Paraphrase** | *Given a passage, please* `paraphrase` *it. The content should be the same. The passage is: $s$* |
| **Open-ended** | *Given a sentence, please* `write a piece of news`. *The sentence is: $s$* |
| **Explicit $p_{input}$**: *information content $s$; text attribute $f$* | |
| **Sentiment** | *Rewrite the following passage, ensuring the content is the same. Meanwhile, make the passage $f$* `emotionally`. *The passage is: $s$* |
| **Word Choice** | *Given a passage, please rewrite it without any explanations. The content should be the same. Make sure the* `word choice` *of the rewritten passage is $f$. The passage is: $s$* |
| **Tone** | *Given a passage, please rewrite it without any explanations. The content should be the same. Make sure the* `tone` *of the rewritten passage is $f$. The passage is: $s$* |
| **Implicit $p_{input}$**: *information content $s$; author attribute $f$* | |
| **Age** | *You are $f$. Given a passage, please rewrite it without any explanations. The content should be the same. The passage is: $s$* |
| **Gender** | *You are $f$. Given a passage, please rewrite it without any explanations. The content should be the same. The passage is: $s$* |

Table 6: The prompts to mitigate the shortcut learning, removing other factors. We `highlight` the special parts of each prompt, where highlighted parts illustrate the main motivation behind each strategy.

| Strategy | Instruction Text |
|---|---|
| **Paraphrase** | *Given a passage, please* `paraphrase` *it. The content should be the same. The passage is: $s$* |
| **Summary** | `Summarize` *the following passage, ensuring it contains the most information about the* `facts`. *Meanwhile, avoid including sentiments, intonation, and other factors affecting the factuality. The passage is: $s$* |
| **Neutral** | *Rewrite the following passage, ensuring the content is the same. Meanwhile, make the passage* `neutral emotionally`. *The passage is: $s$* |

# F  LLMs Could Inject Shortcut without Altering Authenticity

## F.1  Quantitative proof that LLMs can inject shortcuts

To quantitatively prove that LLMs can inject predefined factors, we employ **Word Choice** and **Sentiment**:

- **Word Choice**. We employ infini-gram to calculate the count of a specific token appearing in numerous documents, where a lower value means the word (n-gram token) is more uncommon (complex). Given a piece of text (token sequence), we could obtain a sequence of token count. To compare sequences with different lengths, we calculate the geometric mean (to avoid the influence of extreme values) to represent a sentence.

- **Sentiment**. We employ the same sentiment classifier in Appendix D to obtain the sentiments before and after injection and present results in Figure 13. It obviously illustrates that after injection, the texts with a specific sentiment (*positive* or *negative*) significantly increase, and texts with *neutral* sentiment or opposite sentiments decrease. Thus, we can assert that LLMs can inject the predefined factors.

## F.2 Human evaluation for injection

We further conduct a human evaluation to evaluate the injection attacks. We evaluate both explicit (**Tone** and **Word Choice**) and implicit (**Age** and **Gender**) attacks, where the implicit attacks are hard to evaluate through metrics. We randomly sample 50 data pairs from all datasets for each attack, where each pair contains texts with contrary attributes. For example, for **Word Choice**, each pair contains a sentence with *simple* attribute and a sentence with *complex* attribute, where their original texts (texts before injection) are the same. We ask experts (annotators) to check which one is more in line with a certain attribute. For example, we ask experts to check "which of the following two sentences

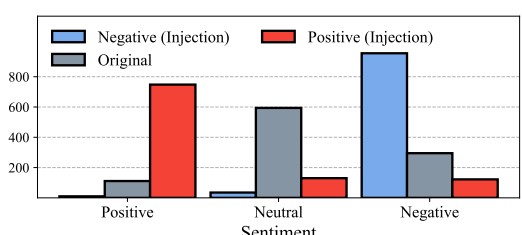

Figure 13: The sentiment distributions in dataset **Pheme** before ("Original") and after **Sentiment** injection attack. It illustrates that LLMs can inject sentiment factors.

has a more complex word choice?" We consider it a three-class classification task (0 for the first sentence, 1 for the second sentence, and 2 for uncertainty). Then we employ Cohen's Kappa between the ground truth of LLM injections and experts for quantitative evaluation, where a higher value denotes humans can better distinguish the attributes and LLMs are more successful in injecting this attribute. The Kappa scores of the three experts are (i) **Tone**: 0.735, 0.878, and 0.842; (ii) **Word Choice**: 1.000, 1.000, and 0.960; (iii) **Age**: -0.009, -0.003, and 0.015; and (iv) **Gender**: 0.000, 0.026, and 0.026. Meanwhile, the experts choose the answer 2 many times for the implicit attacks. The results prove that LLMs can successfully inject explicit shortcuts but fail to inject implicitly.

## F.3 Performance under standard setting after injection

We provide the detector performance under the standard setting and present the results in Tables 24 and 25. For each injection attack, we report the average performance of different attributes. It illustrates that the changes in performance are minor, which proves that authenticity does not alter after injection attacks.

## F.4 Human evaluation for authenticity

We conduct a human evaluation to evaluate whether the authenticity alters, namely, whether text before injection and text after injection present the same fact. We randomly sample 100 positive data pairs from all datasets and all injection attacks. Each positive pair contains an original text (before injection) and a corresponding text after injection. For comparison, we also randomly sample 100 negative data pairs and all sentences are randomly sampled for comparison. We ask experts (annotators) to evaluate whether these two sentences describe the same fact. We consider it a binary classification task (0 for false and 1 for true). We calculate the accuracy between ground truth and expert answers, where a higher accuracy score denotes experts can identify the facts more successfully and LLMs do not alter the authenticity. The accuracy scores of all experts are 0.940, 0.870, and 0.925. We also calculate the Fleiss' Kappa between all experts and the value is 0.838.

## F.5 LLM-based evaluation for authenticity

We employ GPT-4-turbo to conduct an LLM-based evaluation approach similar in spirit to FactScore [38]. Specifically, we prompted GPT-4-turbo as follows:

Table 7: The percentage of "yes" responses across categories.

| Category | Random | Rewriting | Paraphrase | Open-Ended | Positive | Negative | Simple |
|---|---|---|---|---|---|---|---|
| Percentage | 0.12 | 0.92 | 0.88 | 0.98 | 0.80 | 0.82 | 0.96 |
| Category | Complex | Formal | Informal | Young | Elder | Male | Female |
| Percentage | 0.94 | 0.96 | 0.90 | 0.98 | 0.96 | 0.94 | 0.98 |

```
Text 1:   Original Article
Text 2:   Injected Article
Do these two texts describe the same event?  Please answer yes or no.
```

We sample 50 original-injected pairs of each Category, and we ensure each injection is successful. We also sample a "Random" Category where each pair is randomly shuffled and consider it baseline. We present the percentage of "yes" responses across categories in Table 7. The results present that injection does not alter the labels evaluated by LLMs. Notably that LLMs judge that 12% of random shuffle pairs describe the same event, which shows that LLM-as-evaluator might not be reliable in this task. It coincides with the idea of utilizing metric measures and human evaluation.

# G  SMF Details

## G.1  Prompts

We provide the prompts of the mitigation method variants in Table 6.

## G.2  Case Study

We provide some cases of texts before and after the data augmentation in Table 16. We also highlight the parts showing the effectiveness of each method. It qualitatively illustrates that LLMs can remove other factors to mitigate shortcut learning. Generally, after the data augmentation, all texts tend to be similar and do not contain specific attributes.

- **Paraphrase**. After paraphrasing, texts tend to employ similar phrases, such as "*following the Paris shooting*".
- **Summary** can remove words showing specific attributes and maintain the information about authenticity.
- **Neutral** tries to present the authenticity neutrally.

# H  Detectors

We employ three categories of misinformation detectors (including seven detectors) to evaluate the shortcut learning within them. We believe these detectors could cover most mainstream misinformation detectors. Meanwhile, we have publicly published this work's resources, making it easy to generalize the evaluations to novel detectors. We conduct all experiments using eight RTX 4090 GPUs with 24GB of memory. Specifically, all LLM-related experiments can be held on two GPUs, and others can be held on one GPU.

Table 8: The hyperparameters of misinformation detectors. To obtain a fair comparison, we leverage the same hyperparameters for every setting. Meanwhile, we have run five times and reported the best performance to avoid randomness.

| Hyper | BERT | DEBERTA | CMTR | DISC | CATCH |
|---|---|---|---|---|---|
| Optimizer | Adam | | | | |
| Metrics | Accuracy | | | | |
| Weight Decay | 1e-5 | | | | |
| Dropout | 0.5 (if employed) | | | | |
| Hidden Dim | 512 | 512 | 512 | 256 | 512 |
| Learning Rate | 1e-3 | 1e-3 | 1e-3 | 1e-3 | 1e-4 |
| Batch Size | 256 | 256 | 128 | 32 | 16 |

- MISTRALV3[5] and LLAMA3[6] are two widely-used LLMs. We prompt them in a zero-shot fashion using a prompt from [35]:

---

[5]Model checkpoints come from this link.

[6]Model checkpoints come from this link.

```
Claim
```
Please check the above claim true or false.  Just output 'True' or
'False'. We set *max_new_tokens* as 1000 and *temperature* as 0.0 (*do_sample* as false).

- BERT and DEBERTA are two widely-used encoder-based LMs, where most misinformation detectors employ them as the backbone to extract features. Thus, we also employ them to extract features and leverage an MLP layer to identify misinformation. We believe they could present most misinformation detectors.

- CMTR employs the text summarization technology to solve longer sequences and capture additional contextual information. It employs extractive and abstractive summarization, which can remove other factors and maintain authenticity, thus we leverage it as a baseline.

- DISC aims to learn disentangled causal substructure for graphs to construct a debiasing graph neural network. Since there are many works model texts as graphs [15] to classify text, we leverage DISC as a baseline to present this type of detector.

- CATCH is designed to detect cross-platform hate speech detection through causality-guided disentanglement. CATCH is a unified method to disentangle causally related representations for texts. Thus, we employ it as a baseline to present this type of detector.

To ensure reproducibility and promote the development of misinformation detectors, we present the hyperparameters in Table 8. Meanwhile, we have publicly published the resources.

## I  Datasets

We employ 16 misinformation datasets (including two datasets we proposed requiring fact-related knowledge) to evaluate shortcut learning. These datasets are published from 2017 to 2024, and we believe they can cover most mainstream misinformation datasets. Meanwhile, we have publicly published this work's resources, making it easy to construct shortcut learning for novel datasets.

Since the original data size of each dataset is different and the label distribution may be uneven, in order to make a fair comparison (especially since we report the average performance of each dataset in Tables 2 and 3 in the main text), we first sample each dataset. The sampling strategy makes sure that the labels of each dataset are evenly distributed and that each dataset contains at most 1,000 instances. We provide the basic statistics of each dataset in Table 9. Generally, we divide each dataset into training, validation,

Table 9: The basic statistics of each dataset. "# fake" and "# real" denote the count of fake and real instances, "Avg. length" denotes the average character length of each instance, and "Avg. words" denotes the average word count (split by spaces) of each instance.

| Datasets | # fake | # real | Avg. length | Avg. words |
|---|---|---|---|---|
| **RumourEval** | 131 | 184 | 88.1 | 14.7 |
| **Pheme** | 500 | 500 | 82.1 | 13.8 |
| **Twitter15** | 255 | 252 | 78.3 | 13.2 |
| **Twitter16** | 151 | 168 | 75.2 | 12.7 |
| **Celebrity** | 250 | 250 | 2445.6 | 424.1 |
| **FakeNews** | 240 | 240 | 737.7 | 122.0 |
| **Politifact** | 252 | 116 | 6443.2 | 1093.3 |
| **Gossipcop** | 500 | 500 | 4571.3 | 767.2 |
| **Tianchi** | 500 | 500 | 3238.1 | 543.4 |
| **MultiLingual** | 500 | 500 | 198.8 | 33.2 |
| **AntiVax** | 500 | 500 | 100.7 | 17.9 |
| **COCO** | 500 | 500 | 263.3 | 45.8 |
| **kaggle1** | 500 | 500 | 2496.3 | 410.2 |
| **kaggle2** | 500 | 500 | 2684.9 | 437.9 |
| **NQ** | 500 | 500 | 58.6 | 10.7 |
| **Streaming** | 500 | 500 | 69.9 | 12.0 |

and test sets in a ratio of 6:2:2. For intrinsic shortcut induction, to construct the shortcut setting (group instances into different groups), the size is different for different factors. We present the size of the training, validation, and test sets in Table 10. For others, since we employ LLMs to inject factors without modifying the instance index, we follow the same split as the standard setting ("Original" in Table 2).

## J  Complete Observations and Settings

### J.1  Intrinsic Shortcut Induction

To support the finding that existing datasets contain shortcuts irrelevant to authenticity, we present the distributions of the four factors across misinformation and real ones in Figure 14 and Table 17.

Meanwhile, we provide the complete performance of the selected detectors in Table 18. For the shortcut setting, the training-validation sets contain instances with $(e = 0, y = 0)$ and $(e = 1, y = 1)$,

Table 10: The statistics of the shortcut setting in intrinsic shortcut induction. It illustrates that some test sets only contain few instances, where the results may be due to randomness. Thus, we have run the experiments five times and reported the best performance in the main text. Meanwhile, it coincides with our motivation to explore shortcut learning in misinformation detection, promoting the development of misinformation detectors.

| Datasets | Sentiment | | | Style | | | Topic | | | Perplexity | | |
|---|---|---|---|---|---|---|---|---|---|---|---|---|
| | # training | # validation | # test | # training | # validation | # test | # training | # validation | # test | # training | # validation | # test |
| **RumourEval** | 45 | 12 | 9 | 132 | 33 | 10 | 123 | 31 | 11 | 116 | 29 | 21 |
| **Pheme** | 130 | 33 | 29 | 328 | 83 | 41 | 336 | 85 | 64 | 364 | 91 | 57 |
| **Twitter15** | 110 | 28 | 20 | 163 | 41 | 32 | 180 | 46 | 23 | 186 | 47 | 26 |
| **Twitter16** | 80 | 21 | 8 | 94 | 24 | 17 | 95 | 24 | 19 | 125 | 32 | 17 |
| **Celebrity** | 76 | 19 | 9 | 212 | 54 | 2 | 196 | 49 | 19 | 163 | 41 | 24 |
| **FakeNews** | 87 | 22 | 16 | 207 | 52 | 16 | 178 | 45 | 22 | 131 | 33 | 32 |
| **Politifact** | 28 | 7 | 20 | 160 | 40 | 14 | 88 | 23 | 35 | 104 | 26 | 35 |
| **Gossipcop** | 192 | 48 | 21 | 364 | 91 | 40 | 360 | 91 | 34 | 370 | 93 | 43 |
| **Tianchi** | 237 | 60 | 14 | 392 | 98 | 14 | 396 | 99 | 57 | 328 | 82 | 50 |
| **MultiLingual** | 218 | 55 | 35 | 369 | 93 | 27 | 364 | 91 | 61 | 342 | 86 | 64 |
| **AntiVax** | 109 | 28 | 51 | 362 | 91 | 14 | 358 | 90 | 16 | 340 | 85 | 55 |
| **COCO** | 216 | 54 | 50 | 379 | 95 | 2 | 353 | 89 | 44 | 320 | 80 | 46 |
| **kaggle1** | 156 | 40 | 23 | 235 | 59 | 63 | 324 | 82 | 73 | 518 | 130 | 24 |
| **kaggle2** | 112 | 28 | 44 | 504 | 127 | 9 | 341 | 86 | 77 | 192 | 48 | 70 |
| **NQ** | 106 | 27 | 10 | 329 | 83 | 92 | 364 | 92 | 39 | 377 | 95 | 42 |
| **Streaming** | 107 | 27 | 26 | 324 | 82 | 92 | 364 | 91 | 45 | 358 | 90 | 50 |

while the test set contains instances with $(e = 1, y = 0)$ and $(e = 0, y = 1)$. Meanwhile, to provide a baseline performance for comparison, we randomly sample a set with the same size as the training-validation sets.

## J.2 Extrinsic Shortcut Injection

We present the complete performance of all selected detectors in Tables 19 and 20. For the shortcut setting, we employ LLMs to inject predefined factors, which is more free, for example, we can inject positive or negative into the same instance. As a result, we report the average performance of all possible permutations and combinations. For example, assuming $\|e\| = 2$, namely, $e \in \{0, 1\}$, the possible permutations and combinations include (i) the training-validation sets contain instances with $(e = 0, y = 0)$ and $(e = 1, y = 1)$, while the test set contains instances with $(e = 1, y = 0)$ and $(e = 0, y = 1)$; and (ii) the training-validation sets contain instances with $(e = 1, y = 0)$ and $(e = 0, y = 1)$, while the test set contains instances with $(e = 0, y = 0)$ and $(e = 1, y = 1)$. Similarly, for $\|e\| = 3$, namely, $e \in \{0, 1, 2\}$ (**Vanilla**), there are 6 possible settings.

The results enhance the findings that explicit LLM injection attacks significantly harm the detector performance but implicit attacks fail. Meanwhile, LLM-based detectors are robust to the shortcuts, although they struggle to achieve good performance under the standard setting, which coincides with our idea to employ LLMs to remove shortcuts and maintain authenticity.

**Evaluation on sentiment classification datasets**  We have proven that detectors cannot learn authenticity during training under the shortcut setting (Figure 6). We aim to explore what detectors learn and assume that detectors learn shortcuts. Thus, we evaluate detectors before and after training under the **Sentiment** injection attack using two sentiment classification datasets. For each dataset, we sampled 1,000 positive instances and 1,000 negative instances to evaluate. It is notable that we do not directly train detectors using the sentiment labels. However, after training under the shortcut setting, misinformation detectors have the ability to identify sentiments (performance increase shown in Figure 8). Thus, we assert that detectors could learn the corresponding shortcuts.

**The ratio of shortcuts to authenticity associations**  We have described the shortcut setting under LLM injection attacks in Appendix J.2. It is an extreme situation in which the joint distributions are absolutely different for training-validation and test sets. Thus, we set a ratio $\alpha$ to simulate the situations in which the distributions are partially similar. It denotes that $\alpha$ instances in the training-validation sets are from the same joint distribution as the test set, representing the ratio of shortcuts to authenticity associations. To ensure the fairness of the comparison, we keep the test set the same.

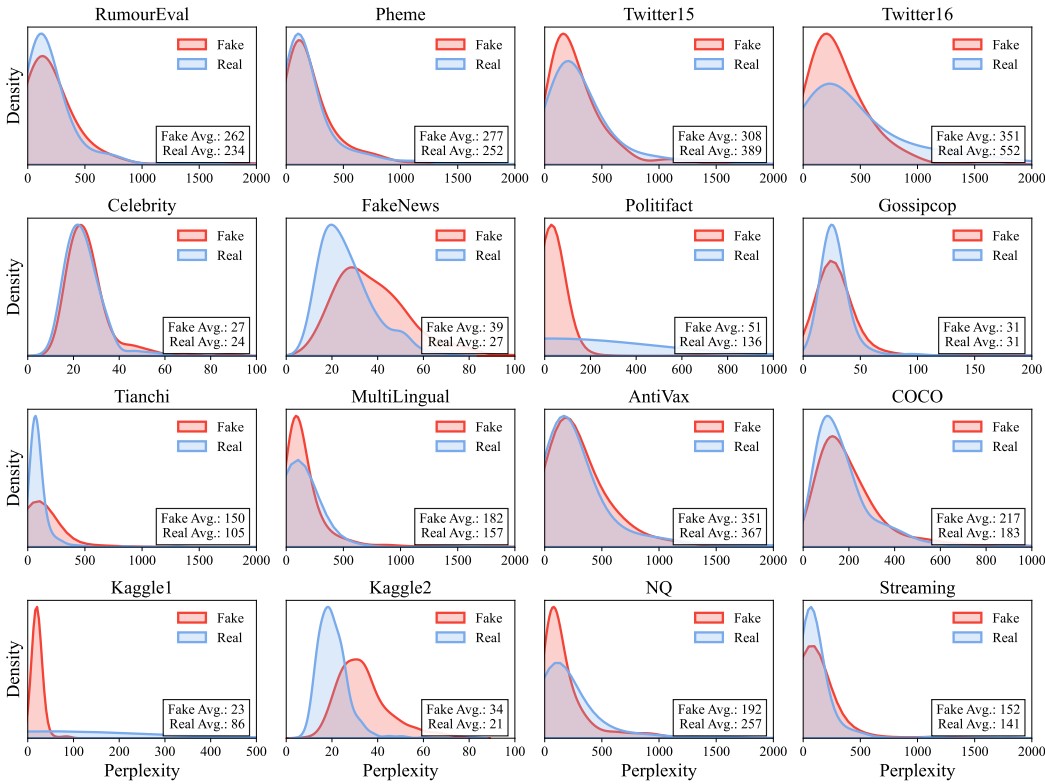

Figure 14: The perplexity distributions of fake and real instances on the employed 16 datasets. We report the average perplexity of fake and real instances. The distributions between fake and real instances are similar on most datasets, which proves that these factors are not causally related to authenticity and are potential shortcuts.

## J.3 Calibration Details

We evaluate the calibration of misinformation detectors after LLM injection attacks. A well-calibrated detector should give an exact confidence score, where a confidence of 80% denotes the estimated accuracy should be 80%. For LLM-based detectors, we employ the probability of the prediction token ("*false*" or "*true*") as the confidence score. For other trainable detectors, we employ the Softmax operator to process logits (the output of the MLP layer) to obtain the confidence score, where we employ the larger value as the confidence score. We then bin them into 10 buckets and calculate the expected calibration error (ECE) [17].

## K Influence of Refusal

In Table 2 in the main text, we retained instances even when the LLMs partially or fully refused to follow the injection prompt. This choice preserved label distributions and ensured a consistent dataset size across original and shortcut conditions, allowing for a direct comparison. To rigorously assess the influence of these refusals, we conducted an additional analysis. We filtered out all instances in which the LLM refused to perform the implicit injection. To maintain fairness, we also removed the corresponding original (non-injected) instances, ensuring pairwise alignment. The number of refusals per dataset is shown in Table 11.

Since these refusals could skew comparisons, we filtered out the refused samples as well as their original counterparts. We conduct further experiments and present the results in Table 12. The results are consistent with our observation in the main text: detectors are generally less affected by implicit injection. This further supports our interpretation that such injections are more subtle in nature, and that current detectors are not easily influenced by these minor shifts in writing style or framing. These

Table 11: The number of refused instances.

| Dataset | # Instances | # Refused (age) | # Refused (gender) | Dataset | # Instances | # Refused (age) | # Refused (gender) |
|---------|-------------|-----------------|--------------------|---------|-------------|-----------------|--------------------|
| **D01** | 315 | 90 | 79 | **D09** | 1000 | 296 | 293 |
| **D02** | 1000 | 283 | 236 | **D10** | 1000 | 193 | 203 |
| **D03** | 507 | 160 | 151 | **D11** | 1000 | 394 | 348 |
| **D04** | 319 | 127 | 122 | **D12** | 1000 | 589 | 552 |
| **D05** | 500 | 28 | 33 | **D13** | 1000 | 191 | 195 |
| **D06** | 480 | 59 | 55 | **D14** | 1000 | 215 | 196 |
| **D07** | 368 | 136 | 128 | **D15** | 1000 | 12 | 14 |
| **D08** | 1000 | 39 | 39 | **D16** | 1000 | 46 | 58 |

Table 12: The results of detectors after filtering the refused instances.

| **Detector** | MISTRALV3 | LLAMA3 | BERT | DEBERTA | CMTR | DISC | CATCH |
|--------------|-----------|--------|------|---------|------|------|-------|
| Original Age | 55.0 | 57.8 | 78.2 | 78.2 | 77.7 | 79.1 | 79.3 |
| Shortcut Age | 56.9 | 56.6 | 74.9 | 74.9 | 75.2 | 75.5 | 76.4 |
| Original Gender | 55.6 | 57.3 | 78.0 | 78.8 | 77.0 | 77.6 | 78.6 |
| Shortcut Gender | 55.6 | 56.1 | 74.8 | 74.5 | 74.0 | 74.8 | 76.8 |

findings strengthen our understanding of the limits of shortcut learning in subtle, implicit contexts and highlight the importance of continuing to explore such dimensions in future work.

# L SMF Result Details

## L.1 SMF can Remove Shortcuts

We first calculate the similarities between text pairs with distinct attributes (*positive* and *negative*) before and after the mitigation methods. We present the results in Table 13, which illustrates that the similarities increase after the mitigation methods. As a result, we assert that LLMs could also remove the shortcuts.

Besides, we also evaluate the ability to remove the specific shortcut (take **Sentiments** as an example) and present the results in Figure 15.

Table 13: The similarities between texts with contrary attributes before and after mitigation methods. The similarities generally increase after mitigation methods, proving that the methods could remove shortcuts.

| Shortcuts | Original | Paraphrase | Summary | Neutral |
|-----------|----------|------------|---------|---------|
| Vanilla | 0.637 | 0.677 (6.33% ↑) | 0.717 (12.57% ↑) | 0.639 (0.31% ↑) |
| Sentiment | 0.622 | 0.643 (3.26% ↑) | 0.723 (16.11% ↑) | 0.663 (6.59% ↑) |
| Tone | 0.688 | 0.760 (10.48% ↑) | 0.798 (16.02% ↑) | 0.719 (4.63% ↑) |
| Word Choice | 0.679 | 0.750 (10.51% ↑) | 0.776 (14.23% ↑) | 0.698 (2.83% ↑) |
| Age | 0.915 | 0.886 (3.14% ↓) | 0.895 (2.18% ↓) | 0.876 (4.28% ↓) |
| Gender | 0.933 | 0.905 (3.09% ↓) | 0.915 (2.00% ↓) | 0.897 (3.91% ↓) |

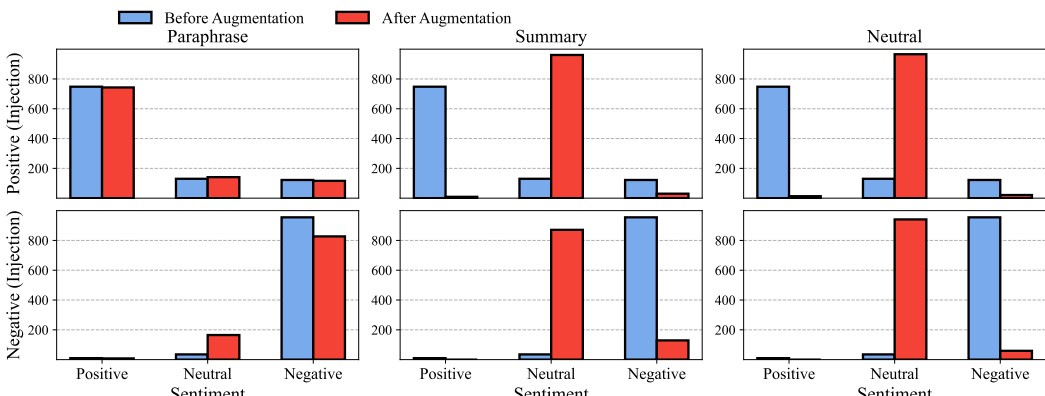

Figure 15: The sentiment distributions under the **Sentiment** injection attack and distributions after the three mitigation methods. It illustrates that the mitigation methods except for **Paraphrase** can remove this shortcut.

Table 14: Average accuracy of LLM-based detectors enhanced with SMF.

| Shortcut | Augmentation | LLM-based Detectors | |
|---|---|---|---|
| | | MISTRALV2 | LLAMA3 |
| Vanilla | w/o aug. | 52.6 | 54.8 |
| | Paraphrase | 52.2 | 55.5 |
| | Summary | 51.5 | 54.2 |
| | Neutral | 51.9 | 52.6 |
| Sentiment | w/o aug. | 51.9 | 52.6 |
| | Paraphrase | 51.0 | 52.8 |
| | Summary | 52.0 | 53.0 |
| | Neutral | 51.0 | 51.6 |
| Tone | w/o aug. | 54.3 | 56.4 |
| | Paraphrase | 53.0 | 55.9 |
| | Summary | 52.2 | 54.2 |
| | Neutral | 52.7 | 54.3 |
| Word Choice | w/o aug. | 54.3 | 56.9 |
| | Paraphrase | 52.8 | 55.5 |
| | Summary | 51.8 | 53.8 |
| | Neutral | 51.3 | 53.6 |
| Age | w/o aug. | 54.5 | 53.8 |
| | Paraphrase | 53.1 | 53.5 |
| | Summary | 51.2 | 52.1 |
| | Neutral | 54.0 | 52.8 |
| Gender | w/o aug. | 53.0 | 53.6 |
| | Paraphrase | 53.3 | 52.9 |
| | Summary | 51.2 | 52.5 |
| | Neutral | 53.7 | 52.0 |

## L.2 Complete Results

We present the complete performance of three mitigation methods in Tables 21, 22, and 23. The results highlight that every augmentation method could effectively improve performance and different variants show advantages for specific shortcuts.

## M SMF Results of LLM-based Detectors

We extend the evaluation to include Mistral-v3 and LLaMA-3 under the SMF setup, where Table 14 presents the results. These results show that applying SMF to LLM-based detectors does not lead to significant gains or drops in performance. This supports our original hypothesis in the main text: LLM-based detectors are generally less affected by shortcut-based perturbations, likely due to their more holistic text understanding.

Table 15: Some cases of the LLM-based injection attacks. We highlight the parts highly related to the intent of specific injection attacks. It qualitatively illustrates that explicit injection attacks can successfully inject predefined factors but implicit attacks fail.

| Original | After Injection |
|---|---|
| Hostages are being held and a siege is taking place at Sydney's Lindt Chocolat Cafe in Martin Place. | **Rewriting**: A critical situation is unfolding in Sydney's Martin Place, where a Lindt Chocolat Cafe has been taken over and a standoff is currently underway. Reports indicate that hostages are being held captive, and authorities are working diligently to resolve the situation safely and efficiently. |
| | **Paraphrase**: A standoff is currently underway at the Lindt Chocolat Cafe in Martin Place, Sydney, where individuals are being held captive and a siege is in progress. |
| | **Open-ended**: **BREAKING: Siege Underway at Sydney's Lindt Chocolat Cafe** 
 SYDNEY, AUSTRALIA - A tense standoff is unfolding at the Lindt Chocolat Cafe in Martin Place, Sydney, as police and emergency services respond to a hostage situation. 
 According to eyewitnesses , a group of people are being held captive inside the cafe, with reports suggesting that a gunman is among the hostages. The situation is believed to have unfolded around 2:00 pm local time , with police receiving multiple calls reporting the incident. 
 Fabricated Details 
 This is a developing story, and we will provide updates as more information becomes available. |
| Believe it or not, this is a shark on the freeway in Houston, Texas. | **Positive**: Wow, can you believe it?! RT : This incredible photo is making waves! A shark was spotted cruising down the freeway in Houston, Texas - talk about a fin-tastic surprise! |
| | **Negative**: Ugh, can you believe this?! RT : It's absolutely disgusting , but apparently, a shark was spotted on the freeway in Houston, Texas. |
| Sydney airspace wasn't closed. A second terror suspect wasn't arrested. Myths around debunked. | **Formal**: It is hereby clarified that Sydney airspace remained operational and was not subject to closure. Furthermore, a second terror suspect was not apprehended. Additionally, various misconceptions and myths surrounding these events have been thoroughly debunked . |
| | **Informal**: So, to set the record straight : Sydney airspace wasn't shut down, and another terror suspect wasn't hauled in. All those rumors and myths have been busted . |
| passenger plane crashes in French Alps with onboard | **Simple**: A plane carrying many people crashed in the French Alps. |
| | **Complex**: A commercial airliner suffered a catastrophic failure, precipitously plummeting to the ground in the rugged French Alps, resulting in a calamitous loss of life. |
| I bet. You never seen this rare natural phenomena. Lighting hits a River. What a sight. Incredible indeed. | **Young**: I'm willing to take the bet. You've never seen this rare natural phenomenon - lightning hitting a river. What a sight! It's truly incredible. |
| | **Elder**: I wager. You've never witnessed this extraordinary natural occurrence. Lightning strikes a river. What a spectacle. Truly astonishing. |
| ICYMI - Virgin Islands Allows National Guard To Seize Guns, Ammo Ahead Of Hurricane Irma | **Male**: ICYMI - The Virgin Islands Allow National Guard To Seize Guns, Ammo Before Hurricane Irma |
| | **Female**: ICYMI - The Virgin Islands Allow National Guard to Seize Guns and Ammo Before Hurricane Irma |

Table 16: Some cases of the LLM-based data augmentation methods. We `highlight` the parts highly related to the intent of specific injection attacks. After data augmentation by LLMs, texts with different attributes tend to be similar and do not present different attributes.

| Texts with Shortcuts | After Data Augmentation |
|---|---|
| **For Paraphrase strategy.** | |
| **Formal**: Reports have `emerged of multiple attacks` against mosques in France in the aftermath of the Paris shooting. | Following the Paris shooting, there have been numerous reports of violent incidents targeting mosques in France. |
| **Informal**: Mosque `attacks in` France after Paris shooting | Following the Paris shooting, a series of attacks took place at mosques in France. |
| **For Summary strategy.** | |
| **Positive**: "Behind the scenes, there's a different story unfolding. These remarkable young men are not only helping to restore order, but they're also demonstrating `the power of resilience and community`. As they work tirelessly to clean up the aftermath of the riot, they're showing us that even in the face of chaos, there is always `hope and always a way forward`. Their `selfless efforts` are a testament to the human spirit, and a reminder that together, we can overcome any obstacle." | The passage describes a group of young men who are involved in restoring order after a riot. They are working to clean up the aftermath of the event. |
| **Negative**: This is what they're `hiding from` you on the media. These young men, `forced to` clean up the mess left behind by the senseless destruction and chaos of the riot, are a stark reminder of the devastating consequences of unchecked violence and anarchy. | The passage describes a situation where young men are forced to clean up the aftermath of a riot, which has caused destruction and chaos. |
| **For Neutral strategy.** | |
| **Positive**: Wow, can you believe it?! RT : This `incredible` photo is making waves! A shark was spotted cruising down the freeway in Houston, Texas - talk about a fin-tastic `surprise!` | A recent photo is generating interest. According to reports, a shark was spotted on a freeway in Houston, Texas. This unusual sighting has caught the attention of many. |
| **Negative**: Ugh, can you believe this?! RT : It's `absolutely disgusting`, but apparently, a shark was spotted on the freeway in Houston, Texas. | A recent report has surfaced indicating that a shark was spotted on a freeway in Houston, Texas. |

Table 17: Sentiment, style, and topic distributions of fake and real instances on the employed 16 datasets. "Different" denotes the ratio of the number of fake instances to the number of real instances, the closer it is to 1, the more similar they are. "T1" to "T8" represent "Sports", "Arts, Culture, and Entertainment", "Business and Finance", "Health and Wellness", "Lifestyle and Fashion", "Science and Technology", "Politics", and "Crime" respectively. The distributions between fake and real instances are similar on most datasets, which proves that these factors are not causally related to authenticity and are potential shortcuts.

| Category | | D01 | D02 | D03 | D04 | D05 | D06 | D07 | D08 | D09 | D10 | D11 | D12 | D13 | D14 | D15 | D16 |
|---|---|---|---|---|---|---|---|---|---|---|---|---|---|---|---|---|---|
| **Sentiment** | | | | | | | | | | | | | | | | | |
| Positive | Fake | 17 | 39 | 33 | 43 | 64 | 62 | 25 | 164 | 176 | 92 | 62 | 92 | 31 | 81 | 99 | 42 |
| | Real | 16 | 72 | 36 | 20 | 85 | 58 | 15 | 108 | 169 | 72 | 223 | 76 | 74 | 39 | 91 | 45 |
| | Difference | 1.06 | 0.54 | 0.92 | 2.15 | 0.75 | 1.07 | 1.67 | 1.52 | 1.04 | 1.28 | 0.28 | 1.21 | 0.42 | 2.08 | 1.09 | 0.93 |
| Neutral | Fake | 72 | 310 | 138 | 74 | 111 | 107 | 118 | 257 | 222 | 237 | 199 | 152 | 343 | 213 | 352 | 346 |
| | Real | 119 | 284 | 90 | 85 | 135 | 121 | 89 | 291 | 183 | 199 | 221 | 197 | 213 | 372 | 358 | 333 |
| | Difference | 0.61 | 1.09 | 1.53 | 0.87 | 0.82 | 0.88 | 1.33 | 0.88 | 1.21 | 1.19 | 0.90 | 0.77 | 1.61 | 0.57 | 0.98 | 1.04 |
| Negative | Fake | 42 | 151 | 84 | 34 | 75 | 71 | 109 | 79 | 102 | 171 | 239 | 256 | 126 | 206 | 49 | 112 |
| | Real | 49 | 144 | 126 | 63 | 30 | 61 | 12 | 101 | 148 | 229 | 56 | 227 | 213 | 89 | 51 | 122 |
| | Difference | 0.86 | 1.05 | 0.67 | 0.54 | 2.50 | 1.16 | 9.08 | 0.78 | 0.69 | 0.75 | 4.27 | 1.13 | 0.59 | 2.31 | 0.96 | 0.92 |
| **Style** | | | | | | | | | | | | | | | | | |
| Subjective | Fake | 86 | 236 | 123 | 59 | 225 | 173 | 193 | 340 | 415 | 382 | 404 | 484 | 194 | 442 | 42 | 60 |
| | Real | 82 | 298 | 147 | 95 | 197 | 144 | 86 | 365 | 437 | 365 | 429 | 480 | 459 | 210 | 44 | 66 |
| | Difference | 1.05 | 0.79 | 0.84 | 0.62 | 1.14 | 1.20 | 2.24 | 0.93 | 0.95 | 1.05 | 0.94 | 1.01 | 0.42 | 2.10 | 0.95 | 0.91 |
| Neutral | Fake | 45 | 264 | 132 | 92 | 25 | 67 | 59 | 160 | 85 | 118 | 96 | 16 | 306 | 58 | 458 | 440 |
| | Real | 102 | 202 | 105 | 73 | 53 | 96 | 30 | 135 | 63 | 135 | 71 | 20 | 41 | 290 | 456 | 434 |
| | Difference | 0.44 | 1.31 | 1.26 | 1.26 | 0.47 | 0.70 | 1.97 | 1.19 | 1.35 | 0.87 | 1.35 | 0.80 | 7.46 | 0.20 | 1.00 | 1.01 |
| **Topic** | | | | | | | | | | | | | | | | | |
| T1 | Fake | 4 | 36 | 36 | 9 | 16 | 40 | 9 | 33 | 21 | 34 | 24 | 7 | 2 | 8 | 111 | 91 |
| | Real | 8 | 49 | 26 | 12 | 8 | 40 | 2 | 32 | 13 | 18 | 48 | 10 | 12 | 2 | 110 | 83 |
| | Difference | 0.50 | 0.73 | 1.38 | 0.75 | 2.00 | 1.00 | 4.50 | 1.03 | 1.62 | 1.89 | 0.50 | 0.70 | 0.17 | 4.00 | 1.01 | 1.10 |
| T2 | Fake | 6 | 36 | 5 | 1 | 39 | 20 | 2 | 119 | 30 | 12 | 2 | 0 | 1 | 6 | 68 | 20 |
| | Real | 3 | 17 | 5 | 0 | 49 | 23 | 1 | 96 | 5 | 2 | 1 | 4 | 10 | 1 | 59 | 16 |
| | Difference | 2.00 | 2.12 | 1.00 | inf | 0.80 | 0.87 | 2.00 | 1.24 | 6.00 | 6.00 | 2.00 | 0.00 | 0.10 | 6.00 | 1.15 | 1.25 |
| T3 | Fake | 28 | 78 | 44 | 18 | 102 | 75 | 38 | 150 | 125 | 164 | 101 | 218 | 87 | 76 | 60 | 104 |
| | Real | 29 | 102 | 78 | 40 | 78 | 76 | 26 | 172 | 102 | 166 | 55 | 257 | 57 | 76 | 64 | 107 |
| | Difference | 0.97 | 0.76 | 0.56 | 0.45 | 1.31 | 0.99 | 1.46 | 0.87 | 1.23 | 0.99 | 1.84 | 0.85 | 1.53 | 1.00 | 0.94 | 0.97 |
| T4 | Fake | 24 | 23 | 51 | 18 | 2 | 1 | 10 | 11 | 29 | 34 | 270 | 80 | 2 | 8 | 63 | 56 |
| | Real | 12 | 40 | 42 | 23 | 4 | 4 | 3 | 10 | 3 | 40 | 361 | 62 | 4 | 1 | 78 | 65 |
| | Difference | 2.00 | 0.57 | 1.21 | 0.78 | 0.50 | 0.25 | 3.33 | 1.10 | 9.67 | 0.85 | 0.75 | 1.29 | 0.50 | 8.00 | 0.81 | 0.86 |
| T5 | Fake | 6 | 26 | 5 | 11 | 20 | 0 | 2 | 35 | 0 | 9 | 1 | 0 | 0 | 0 | 93 | 67 |
| | Real | 6 | 29 | 11 | 11 | 28 | 0 | 0 | 32 | 1 | 8 | 4 | 0 | 2 | 0 | 83 | 71 |
| | Difference | 1.00 | 0.90 | 0.45 | 1.00 | 0.71 | - | - | 1.09 | 0.00 | 1.12 | 0.25 | - | 0.00 | - | 1.12 | 0.94 |
| T6 | Fake | 12 | 8 | 21 | 4 | 1 | 15 | 15 | 10 | 13 | 14 | 17 | 5 | 2 | 7 | 47 | 37 |
| | Real | 6 | 15 | 8 | 16 | 2 | 17 | 4 | 8 | 7 | 17 | 7 | 10 | 6 | 0 | 44 | 35 |
| | Difference | 2.00 | 0.53 | 2.62 | 0.25 | 0.50 | 0.88 | 3.75 | 1.25 | 1.86 | 0.82 | 2.43 | 0.50 | 0.33 | - | 1.07 | 1.06 |
| T7 | Fake | 17 | 16 | 9 | 25 | 9 | 64 | 70 | 6 | 107 | 128 | 38 | 87 | 214 | 167 | 33 | 69 |
| | Real | 12 | 59 | 18 | 32 | 6 | 63 | 47 | 25 | 117 | 231 | 12 | 100 | 231 | 213 | 39 | 73 |
| | Difference | 1.42 | 0.27 | 0.50 | 0.78 | 1.50 | 1.02 | 1.49 | 0.24 | 0.70 | 1.09 | 3.17 | 0.87 | 0.93 | 0.78 | 0.85 | 0.95 |
| T8 | Fake | 34 | 277 | 84 | 65 | 61 | 25 | 106 | 136 | 175 | 105 | 47 | 103 | 192 | 228 | 25 | 56 |
| | Real | 108 | 189 | 64 | 34 | 75 | 17 | 33 | 125 | 216 | 132 | 12 | 57 | 178 | 207 | 23 | 50 |
| | Difference | 0.31 | 1.47 | 1.31 | 1.91 | 0.81 | 1.47 | 3.21 | 1.09 | 0.81 | 0.80 | 3.92 | 1.81 | 1.08 | 1.10 | 1.09 | 1.12 |

Table 18: The accuracy of the selected detectors except DEBERTA under four potential shortcuts. We only report the accuracy for LLM-based detectors since they do not require training and the test sets are the same.

| BERT | | D01 | D02 | D03 | D04 | D05 | D06 | D07 | D08 | D09 | D10 | D11 | D12 | D13 | D14 | D15 | D16 |
|---|---|---|---|---|---|---|---|---|---|---|---|---|---|---|---|---|---|
| **Sentiment** | Random | 0.0 | 86.2 | 85.0 | 87.5 | 77.8 | 0.0 | 95.0 | 71.4 | 64.3 | 40.0 | 86.3 | 74.0 | 100.0 | 97.7 | 100.0 | 11.5 |
| | Shortcut | 0.0 | 6.9 | 15.0 | 12.5 | 55.6 | 56.2 | 80.0 | 28.6 | 28.6 | 8.6 | 72.5 | 46.0 | 69.6 | 90.9 | 10.0 | 7.7 |
| | Difference | - | 92%↓ | 82%↓ | 86%↓ | 29%↓ | - | 16%↓ | 60%↓ | 56%↓ | 79%↓ | 16%↓ | 38%↓ | 30%↓ | 7%↓ | 90%↓ | 33%↓ |
| **Style** | Random | 50.0 | 92.7 | 84.4 | 82.4 | 50.0 | 87.5 | 100.0 | 57.5 | 42.9 | 55.6 | 92.9 | 50.0 | 98.4 | 88.9 | 37.0 | 98.9 |
| | Shortcut | 40.0 | 73.2 | 84.4 | 70.6 | 100.0 | 93.8 | 92.9 | 65.0 | 92.9 | 59.3 | 85.7 | 100.0 | 100.0 | 88.9 | 0.0 | 6.5 |
| | Difference | 20%↓ | 21%↓ | 0% | 14%↓ | 100%↑ | 7%↑ | 7%↓ | 13%↑ | 117%↑ | 7%↑ | 8%↓ | 100%↑ | 2%↑ | 0% | 100%↓ | 93%↓ |
| **Topic** | Random | 72.7 | 79.7 | 91.3 | 94.7 | 68.4 | 86.4 | 100.0 | 85.3 | 64.9 | 50.8 | 87.5 | 86.4 | 97.3 | 98.7 | 79.5 | 22.2 |
| | Shortcut | 36.4 | 25.0 | 56.5 | 63.2 | 68.4 | 31.8 | 77.1 | 50.0 | 24.6 | 29.5 | 100.0 | 56.8 | 90.4 | 85.7 | 48.7 | 55.6 |
| | Difference | 50%↓ | 69%↓ | 38%↓ | 33%↓ | 0% | 63%↓ | 23%↓ | 41%↓ | 62%↓ | 42%↓ | 14%↑ | 34%↓ | 7%↓ | 13%↓ | 39%↓ | 150%↑ |
| **Perplexity** | Random | 76.2 | 84.2 | 92.3 | 100.0 | 70.8 | 78.1 | 100.0 | 55.8 | 72.0 | 62.5 | 98.2 | 87.0 | 91.7 | 98.6 | 100.0 | 100.0 |
| | Shortcut | 28.6 | 64.9 | 80.8 | 82.4 | 29.2 | 62.5 | 85.7 | 34.9 | 34.0 | 45.3 | 89.1 | 37.0 | 87.5 | 94.3 | 38.1 | 50.0 |
| | Difference | 62%↓ | 23%↓ | 12%↓ | 18%↓ | 59%↓ | 20%↓ | 14%↓ | 38%↓ | 53%↓ | 28%↓ | 9%↓ | 58%↓ | 5%↓ | 4%↓ | 62%↓ | 50%↓ |

| CMTR | | D01 | D02 | D03 | D04 | D05 | D06 | D07 | D08 | D09 | D10 | D11 | D12 | D13 | D14 | D15 | D16 |
|---|---|---|---|---|---|---|---|---|---|---|---|---|---|---|---|---|---|
| **Sentiment** | Random | 11.1 | 89.7 | 75.0 | 87.5 | 77.8 | 0.0 | 100.0 | 42.9 | 71.4 | 71.4 | 88.2 | 66.0 | 100.0 | 90.9 | 100.0 | 23.1 |
| | Shortcut | 0.0 | 27.6 | 20.0 | 12.5 | 55.6 | 31.2 | 100.0 | 52.4 | 21.4 | 11.4 | 66.7 | 46.0 | 82.6 | 86.4 | 10.0 | 7.7 |
| | Difference | 100%↓ | 69%↓ | 73%↓ | 86%↓ | 29%↓ | - | 0% | 22%↑ | 70%↓ | 84%↓ | 24%↓ | 30%↓ | 17%↓ | 5%↓ | 90%↓ | 67%↓ |
| **Style** | Random | 70.0 | 95.1 | 93.8 | 88.2 | 100.0 | 56.2 | 78.6 | 80.0 | 57.1 | 44.4 | 92.9 | 50.0 | 98.4 | 88.9 | 100.0 | 100.0 |
| | Shortcut | 40.0 | 70.7 | 78.1 | 64.7 | 100.0 | 100.0 | 92.9 | 50.0 | 100.0 | 92.6 | 100.0 | 100.0 | 100.0 | 88.9 | 0.0 | 12.0 |
| | Difference | 43%↓ | 26%↓ | 17%↓ | 27%↓ | 0% | 78%↑ | 18%↑ | 38%↓ | 75%↑ | 108%↑ | 8%↑ | 100%↑ | 2%↑ | 0% | 100%↓ | 88%↓ |
| **Topic** | Random | 81.8 | 79.7 | 91.3 | 100.0 | 63.2 | 50.0 | 97.1 | 88.2 | 100.0 | 60.7 | 93.8 | 84.1 | 95.9 | 97.4 | 100.0 | 13.3 |
| | Shortcut | 0.0 | 34.8 | 34.8 | 57.9 | 78.9 | 36.4 | 68.6 | 79.4 | 24.6 | 36.1 | 93.8 | 54.5 | 75.3 | 70.1 | 46.2 | 35.6 |
| | Difference | 100%↓ | 55%↓ | 62%↓ | 42%↓ | 25%↑ | 27%↓ | 29%↓ | 10%↓ | 75%↓ | 41%↓ | 0% | 35%↓ | 21%↓ | 28%↓ | 54%↓ | 167%↑ |
| **Perplexity** | Random | 76.2 | 86.0 | 92.3 | 100.0 | 54.2 | 93.8 | 94.3 | 65.1 | 56.0 | 60.9 | 98.2 | 78.3 | 95.8 | 97.1 | 90.5 | 100.0 |
| | Shortcut | 61.9 | 66.7 | 84.6 | 82.4 | 50.0 | 3.1 | 94.3 | 62.8 | 58.0 | 56.2 | 92.7 | 37.0 | 91.7 | 87.1 | 31.0 | 40.0 |
| | Difference | 19%↓ | 22%↓ | 8%↓ | 18%↓ | 8%↓ | 97%↓ | 0% | 4%↓ | 4%↑ | 8%↓ | 6%↓ | 53%↓ | 4%↓ | 10%↓ | 66%↓ | 60%↓ |

| DISC | | D01 | D02 | D03 | D04 | D05 | D06 | D07 | D08 | D09 | D10 | D11 | D12 | D13 | D14 | D15 | D16 |
|---|---|---|---|---|---|---|---|---|---|---|---|---|---|---|---|---|---|
| **Sentiment** | Random | 0.0 | 82.8 | 65.0 | 87.5 | 66.7 | 0.0 | 95.0 | 71.4 | 64.3 | 60.0 | 92.2 | 78.0 | 100.0 | 93.2 | 100.0 | 26.9 |
| | Shortcut | 22.2 | 24.1 | 20.0 | 12.5 | 66.7 | 56.2 | 80.0 | 14.3 | 28.6 | 11.4 | 86.3 | 46.0 | 73.9 | 93.2 | 0.0 | 7.7 |
| | Difference | - | 71%↓ | 69%↓ | 86%↓ | 0% | - | 16%↓ | 80%↓ | 56%↓ | 81%↓ | 6%↓ | 41%↓ | 26%↓ | 0% | 100%↓ | 71%↓ |
| **Style** | Random | 90.0 | 92.7 | 75.0 | 82.4 | 100.0 | 75.0 | 100.0 | 60.0 | 50.0 | 55.6 | 78.6 | 100.0 | 98.4 | 88.9 | 97.8 | 77.2 |
| | Shortcut | 60.0 | 73.2 | 81.2 | 70.6 | 100.0 | 93.8 | 71.4 | 42.5 | 92.9 | 88.9 | 85.7 | 100.0 | 100.0 | 88.9 | 7.6 | 14.1 |
| | Difference | 33%↓ | 21%↓ | 8%↑ | 14%↓ | 0% | 25%↑ | 29%↓ | 29%↓ | 86%↑ | 60%↑ | 9%↑ | 0% | 2%↑ | 0% | 92%↓ | 82%↓ |
| **Topic** | Random | 63.6 | 90.6 | 91.3 | 94.7 | 63.2 | 95.5 | 100.0 | 82.4 | 63.2 | 63.4 | 93.8 | 75.0 | 97.3 | 98.7 | 100.0 | 68.9 |
| | Shortcut | 54.5 | 28.1 | 65.2 | 57.9 | 73.7 | 36.4 | 82.9 | 58.8 | 24.6 | 27.9 | 100.0 | 56.8 | 91.8 | 80.5 | 41.0 | 51.1 |
| | Difference | 14%↓ | 69%↓ | 29%↓ | 39%↓ | 17%↑ | 62%↓ | 17%↓ | 29%↓ | 61%↓ | 56%↓ | 7%↑ | 24%↓ | 6%↓ | 18%↓ | 59%↓ | 26%↓ |
| **Perplexity** | Random | 81.0 | 86.0 | 92.3 | 100.0 | 75.0 | 90.6 | 100.0 | 79.1 | 70.0 | 76.6 | 96.4 | 87.0 | 91.7 | 98.6 | 45.2 | 100.0 |
| | Shortcut | 57.1 | 70.2 | 88.5 | 82.4 | 29.2 | 62.5 | 97.1 | 46.5 | 46.0 | 53.1 | 90.9 | 39.1 | 87.5 | 97.1 | 31.0 | 36.0 |
| | Difference | 29%↓ | 18%↓ | 4%↓ | 18%↓ | 61%↓ | 31%↓ | 3%↓ | 41%↓ | 34%↓ | 31%↓ | 6%↓ | 55%↓ | 5%↓ | 1%↓ | 32%↓ | 64%↓ |

| CATCH | | D01 | D02 | D03 | D04 | D05 | D06 | D07 | D08 | D09 | D10 | D11 | D12 | D13 | D14 | D15 | D16 |
|---|---|---|---|---|---|---|---|---|---|---|---|---|---|---|---|---|---|
| **Sentiment** | Random | 55.6 | 79.3 | 85.0 | 100.0 | 77.8 | 87.5 | 90.0 | 52.4 | 57.1 | 51.4 | 84.3 | 86.0 | 100.0 | 100.0 | 60.0 | 46.2 |
| | Shortcut | 77.8 | 41.4 | 30.0 | 37.5 | 55.6 | 81.2 | 50.0 | 23.8 | 28.6 | 14.3 | 76.5 | 54.0 | 100.0 | 100.0 | 80.0 | 26.9 |
| | Difference | 40%↑ | 48%↓ | 65%↓ | 62%↓ | 29%↓ | 7%↓ | 44%↓ | 55%↓ | 50%↓ | 72%↓ | 9%↓ | 37%↓ | 0% | 0% | 33%↑ | 42%↓ |
| **Style** | Random | 60.0 | 92.7 | 93.8 | 82.4 | 100.0 | 100.0 | 85.7 | 70.0 | 78.6 | 63.0 | 92.9 | 100.0 | 98.4 | 100.0 | 50.0 | 60.9 |
| | Shortcut | 80.0 | 65.9 | 71.9 | 88.2 | 100.0 | 100.0 | 85.7 | 60.0 | 85.7 | 77.8 | 92.9 | 100.0 | 100.0 | 100.0 | 3.3 | 10.9 |
| | Difference | 33%↑ | 29%↓ | 23%↓ | 7%↑ | 0% | 0% | 0% | 14%↓ | 9%↑ | 24%↑ | 0% | 0% | 2%↑ | 11%↓ | 93%↓ | 82%↓ |
| **Topic** | Random | 63.6 | 76.6 | 87.0 | 94.7 | 73.7 | 95.5 | 85.7 | 70.6 | 64.9 | 62.3 | 100.0 | 68.2 | 95.9 | 98.7 | 66.7 | 57.8 |
| | Shortcut | 54.5 | 29.7 | 39.1 | 78.9 | 78.9 | 90.9 | 80.0 | 64.7 | 28.1 | 41.0 | 100.0 | 54.5 | 94.5 | 98.7 | 59.0 | 42.2 |
| | Difference | 14%↓ | 61%↓ | 55%↓ | 17%↓ | 7%↑ | 5%↓ | 7%↓ | 8%↓ | 57%↓ | 34%↓ | 0% | 20%↓ | 1%↓ | 0% | 12%↓ | 27%↓ |
| **Perplexity** | Random | 57.1 | 84.2 | 80.8 | 100.0 | 87.5 | 93.8 | 88.6 | 74.4 | 66.0 | 59.4 | 90.9 | 80.4 | 91.7 | 98.6 | 73.8 | 62.0 |
| | Shortcut | 33.3 | 63.2 | 61.5 | 88.2 | 79.2 | 90.6 | 85.7 | 51.2 | 50.0 | 40.6 | 87.3 | 58.7 | 87.5 | 100.0 | 52.4 | 26.0 |
| | Difference | 42%↓ | 25%↓ | 24%↓ | 12%↓ | 10%↓ | 3%↓ | 3%↓ | 31%↓ | 24%↓ | 32%↓ | 4%↓ | 27%↓ | 5%↓ | 1%↑ | 29%↓ | 58%↓ |

| MISTRALV3 | D01 | D02 | D03 | D04 | D05 | D06 | D07 | D08 | D09 | D10 | D11 | D12 | D13 | D14 | D15 | D16 |
|---|---|---|---|---|---|---|---|---|---|---|---|---|---|---|---|---|
| **Sentiment** | 66.7 | 31.0 | 20.0 | 0.0 | 55.6 | 37.5 | 45.0 | 9.5 | 21.4 | 34.3 | 56.9 | 60.0 | 0.0 | 20.5 | 0.0 | 26.9 |
| **Style** | 81.8 | 29.7 | 13.0 | 21.1 | 57.9 | 63.6 | 37.1 | 2.9 | 35.1 | 23.0 | 56.2 | 61.4 | 4.1 | 18.2 | 53.8 | 26.7 |
| **Topic** | 60.0 | 26.8 | 21.9 | 23.5 | 100.0 | 25.0 | 21.4 | 2.5 | 21.4 | 18.5 | 64.3 | 0.0 | 1.6 | 0.0 | 50.0 | 28.3 |
| **Perplexity** | 47.6 | 35.1 | 23.1 | 17.6 | 62.5 | 53.1 | 40.0 | 9.3 | 38.0 | 25.0 | 61.8 | 65.2 | 12.5 | 20.0 | 50.0 | 36.0 |

| LLAMA3 | D01 | D02 | D03 | D04 | D05 | D06 | D07 | D08 | D09 | D10 | D11 | D12 | D13 | D14 | D15 | D16 |
|---|---|---|---|---|---|---|---|---|---|---|---|---|---|---|---|---|
| **Sentiment** | 100.0 | 75.9 | 70.0 | 75.0 | 77.8 | 87.5 | 75.0 | 19.0 | 57.1 | 45.7 | 92.2 | 98.0 | 0.0 | 43.2 | 60.0 | 80.8 |
| **Style** | 80.0 | 56.1 | 62.5 | 70.6 | 50.0 | 68.8 | 35.7 | 20.0 | 63.0 | 63.0 | 92.9 | 100.0 | 9.5 | 11.1 | 84.8 | 83.7 |
| **Topic** | 100.0 | 57.8 | 56.5 | 63.2 | 68.4 | 72.7 | 62.9 | 23.5 | 57.9 | 59.0 | 93.8 | 97.7 | 9.6 | 42.9 | 87.2 | 84.4 |
| **Perplexity** | 90.5 | 71.9 | 65.4 | 76.5 | 62.5 | 78.1 | 74.3 | 30.2 | 62.0 | 60.9 | 92.7 | 100.0 | 4.2 | 42.9 | 78.6 | 82.0 |

Table 19: The complete detector performance of extrinsic shortcut injection.

| MISTRALV3 | | D01 | D02 | D03 | D04 | D05 | D06 | D07 | D08 | D09 | D10 | D11 | D12 | D13 | D14 | D15 | D16 |
|---|---|---|---|---|---|---|---|---|---|---|---|---|---|---|---|---|---|
| Original | | 69.8 | 54.0 | 34.3 | 30.8 | 70.0 | 74.0 | 49.3 | 43.0 | 49.5 | 49.5 | 72.5 | 55.0 | 42.0 | 60.0 | 58.5 | 50.0 |
| Vanilla | | 65.6 | 56.2 | 37.3 | 53.3 | 55.3 | 64.2 | 42.7 | 44.2 | 53.5 | 45.2 | 62.0 | 54.8 | 44.3 | 59.2 | 53.0 | 51.5 |
| Explicit | Sentiment | 61.9 | 53.8 | 38.7 | 58.5 | 51.5 | 57.8 | 42.7 | 44.8 | 53.0 | 47.5 | 57.0 | 54.2 | 49.2 | 52.5 | 54.2 | 53.5 |
| | Tone | 65.9 | 57.0 | 33.8 | 46.2 | 57.0 | 65.1 | 53.3 | 45.0 | 48.5 | 49.0 | 69.2 | 63.2 | 42.5 | 60.8 | 58.8 | 53.2 |
| | Word Choice | 65.1 | 51.5 | 36.8 | 53.1 | 58.5 | 67.7 | 52.0 | 46.0 | 51.0 | 51.2 | 72.5 | 56.5 | 42.5 | 59.5 | 55.5 | 50.0 |
| Implicit | Age | 65.9 | 53.5 | 40.7 | 53.1 | 61.5 | 70.3 | 54.0 | 40.5 | 53.8 | 46.8 | 68.2 | 44.2 | 44.5 | 61.5 | 59.8 | 53.8 |
| | Gender | 64.3 | 53.8 | 37.7 | 55.4 | 54.5 | 71.4 | 48.0 | 41.2 | 55.8 | 45.8 | 66.0 | 42.0 | 44.5 | 60.0 | 54.2 | 53.8 |
| LLAMA3 | | D01 | D02 | D03 | D04 | D05 | D06 | D07 | D08 | D09 | D10 | D11 | D12 | D13 | D14 | D15 | D16 |
| Original | | 73.0 | 49.5 | 40.2 | 41.5 | 77.0 | 85.4 | 70.7 | 49.5 | 45.5 | 58.0 | 74.0 | 66.5 | 30.5 | 71.0 | 56.0 | 53.0 |
| Vanilla | | 57.1 | 54.8 | 45.1 | 52.8 | 63.3 | 74.7 | 61.3 | 47.3 | 49.7 | 52.2 | 51.5 | 57.0 | 39.7 | 60.2 | 57.3 | 52.3 |
| Explicit | Sentiment | 56.3 | 56.8 | 45.6 | 54.6 | 65.0 | 65.1 | 56.0 | 43.8 | 49.2 | 49.0 | 50.2 | 50.8 | 43.0 | 51.0 | 54.2 | 50.8 |
| | Tone | 63.5 | 50.0 | 47.5 | 48.5 | 67.0 | 81.8 | 61.3 | 43.5 | 44.0 | 53.8 | 65.8 | 64.8 | 39.2 | 62.2 | 58.8 | 50.5 |
| | Word Choice | 58.7 | 51.5 | 41.7 | 50.0 | 68.0 | 79.2 | 66.7 | 47.2 | 46.2 | 51.2 | 66.2 | 67.0 | 35.2 | 67.0 | 59.0 | 54.8 |
| Implicit | Age | 63.5 | 50.0 | 44.6 | 53.1 | 72.5 | 78.1 | 48.7 | 44.0 | 54.5 | 51.8 | 52.0 | 40.2 | 37.2 | 62.5 | 56.8 | 51.5 |
| | Gender | 62.7 | 49.0 | 44.1 | 56.9 | 71.0 | 73.4 | 46.7 | 49.0 | 56.2 | 52.5 | 51.2 | 40.0 | 36.5 | 62.0 | 52.5 | 53.5 |
| BERT | | D01 | D02 | D03 | D04 | D05 | D06 | D07 | D08 | D09 | D10 | D11 | D12 | D13 | D14 | D15 | D16 |
| Original | | 74.6 | 80.0 | 86.3 | 84.6 | 78.0 | 78.1 | 88.0 | 72.5 | 72.5 | 60.5 | 90.5 | 76.5 | 96.5 | 98.5 | 58.5 | 54.5 |
| Vanilla | | 11.4 | 12.1 | 17.3 | 21.0 | 20.0 | 10.6 | 39.6 | 10.2 | 9.7 | 4.5 | 22.2 | 8.9 | 37.7 | 47.3 | 1.7 | 3.3 |
| Explicit | Sentiment | 10.3 | 4.0 | 6.9 | 14.6 | 3.5 | 4.2 | 20.7 | 3.2 | 7.5 | 1.8 | 12.2 | 4.2 | 19.0 | 28.5 | 1.5 | 3.0 |
| | Tone | 10.3 | 7.8 | 12.7 | 19.2 | 6.0 | 3.1 | 16.7 | 2.5 | 8.2 | 2.8 | 13.0 | 3.5 | 10.8 | 9.2 | 8.0 | 6.8 |
| | Word Choice | 6.3 | 7.8 | 7.4 | 19.2 | 2.0 | 2.1 | 9.3 | 1.0 | 4.8 | 4.0 | 13.8 | 3.0 | 9.5 | 13.0 | 3.0 | 6.5 |
| Implicit | Age | 59.5 | 73.5 | 75.5 | 76.9 | 84.0 | 74.5 | 88.0 | 67.2 | 66.8 | 56.2 | 86.5 | 70.8 | 95.8 | 97.0 | 52.0 | 50.5 |
| | Gender | 72.2 | 74.0 | 76.0 | 75.4 | 80.5 | 77.1 | 91.3 | 67.8 | 69.2 | 61.8 | 90.0 | 71.5 | 95.5 | 97.0 | 46.5 | 43.2 |
| DEBERTA | | D01 | D02 | D03 | D04 | D05 | D06 | D07 | D08 | D09 | D10 | D11 | D12 | D13 | D14 | D15 | D16 |
| Original | | 68.3 | 75.0 | 74.5 | 86.2 | 85.0 | 99.0 | 90.7 | 65.5 | 77.5 | 63.5 | 93.0 | 79.0 | 99.0 | 99.0 | 66.0 | 61.5 |
| Vanilla | | 9.3 | 8.9 | 9.0 | 13.6 | 5.8 | 9.2 | 28.9 | 4.2 | 6.2 | 4.0 | 11.2 | 6.2 | 28.9 | 29.7 | 1.8 | 2.9 |
| Explicit | Sentiment | 14.3 | 6.5 | 10.8 | 19.2 | 14.0 | 15.6 | 42.0 | 4.8 | 13.2 | 3.8 | 20.0 | 5.5 | 58.0 | 74.2 | 2.5 | 2.8 |
| | Tone | 6.3 | 5.0 | 6.9 | 13.8 | 0.5 | 6.8 | 16.7 | 2.8 | 6.0 | 2.5 | 5.0 | 4.5 | 11.2 | 11.2 | 4.3 | 4.2 |
| | Word Choice | 4.8 | 2.5 | 5.9 | 14.6 | 2.0 | 2.1 | 21.3 | 1.2 | 4.3 | 1.5 | 7.5 | 2.8 | 14.0 | 11.8 | 2.2 | 4.8 |
| Implicit | Age | 55.6 | 70.2 | 77.0 | 69.2 | 84.0 | 94.8 | 86.0 | 64.2 | 67.8 | 59.2 | 87.0 | 71.0 | 97.2 | 98.8 | 56.0 | 55.5 |
| | Gender | 50.8 | 69.5 | 66.7 | 76.2 | 80.5 | 93.8 | 85.3 | 65.8 | 70.0 | 60.0 | 89.8 | 69.8 | 97.5 | 98.8 | 53.8 | 55.5 |
| CMTR | | D01 | D02 | D03 | D04 | D05 | D06 | D07 | D08 | D09 | D10 | D11 | D12 | D13 | D14 | D15 | D16 |
| Original | | 77.8 | 80.0 | 85.3 | 83.1 | 77.0 | 69.8 | 78.7 | 71.5 | 69.0 | 63.0 | 90.0 | 74.5 | 97.5 | 98.5 | 57.0 | 58.0 |
| Vanilla | | 18.3 | 17.2 | 25.0 | 43.3 | 24.3 | 11.8 | 56.9 | 12.8 | 10.2 | 4.9 | 28.3 | 9.0 | 43.8 | 54.4 | 2.0 | 3.3 |
| Explicit | Sentiment | 11.9 | 6.2 | 9.8 | 19.2 | 4.5 | 5.2 | 26.7 | 5.0 | 8.2 | 1.8 | 12.0 | 6.2 | 22.2 | 26.8 | 1.2 | 4.0 |
| | Tone | 12.7 | 11.0 | 18.6 | 22.3 | 4.0 | 4.7 | 16.7 | 3.2 | 9.5 | 3.2 | 9.0 | 3.8 | 19.2 | 19.5 | 8.2 | 7.0 |
| | Word Choice | 9.5 | 6.5 | 9.8 | 24.6 | 4.5 | 2.6 | 15.3 | 2.0 | 6.0 | 3.8 | 14.2 | 3.2 | 13.2 | 16.0 | 4.2 | 6.8 |
| Implicit | Age | 72.2 | 73.8 | 76.5 | 78.5 | 78.5 | 68.8 | 85.3 | 70.5 | 62.8 | 57.0 | 86.5 | 72.2 | 94.5 | 96.2 | 52.0 | 51.8 |
| | Gender | 76.2 | 74.8 | 76.5 | 78.5 | 74.5 | 66.7 | 89.3 | 67.8 | 65.5 | 61.5 | 88.2 | 71.8 | 95.5 | 97.2 | 47.8 | 45.5 |
| DISC | | D01 | D02 | D03 | D04 | D05 | D06 | D07 | D08 | D09 | D10 | D11 | D12 | D13 | D14 | D15 | D16 |
| Original | | 73.0 | 79.0 | 85.3 | 89.2 | 81.0 | 78.1 | 85.3 | 69.5 | 72.0 | 60.5 | 90.5 | 78.0 | 98.0 | 98.5 | 59.5 | 56.5 |
| Vanilla | | 15.9 | 9.6 | 17.8 | 23.6 | 17.3 | 8.0 | 41.8 | 3.1 | 4.4 | 2.3 | 18.8 | 6.3 | 23.6 | 40.8 | 1.5 | 3.0 |
| Explicit | Sentiment | 11.1 | 4.0 | 8.3 | 13.8 | 3.0 | 4.7 | 19.3 | 3.0 | 7.0 | 1.2 | 6.5 | 3.8 | 18.8 | 33.8 | 0.5 | 3.0 |
| | Tone | 11.1 | 8.8 | 15.7 | 18.5 | 4.0 | 3.1 | 12.0 | 2.2 | 9.8 | 4.0 | 10.2 | 3.8 | 13.0 | 12.0 | 8.2 | 7.2 |
| | Word Choice | 11.9 | 5.5 | 6.4 | 23.1 | 1.5 | 1.6 | 9.3 | 1.0 | 5.0 | 3.2 | 12.5 | 3.0 | 8.5 | 5.8 | 4.2 | 6.5 |
| Implicit | Age | 62.7 | 73.0 | 75.0 | 74.6 | 80.0 | 76.0 | 88.0 | 67.0 | 68.8 | 58.0 | 87.2 | 72.0 | 96.5 | 97.5 | 51.0 | 52.2 |
| | Gender | 77.0 | 74.8 | 74.5 | 79.2 | 80.5 | 76.6 | 90.0 | 67.5 | 67.8 | 61.0 | 90.0 | 71.2 | 96.2 | 98.2 | 48.2 | 43.5 |

Table 20: The complete detector performance of extrinsic shortcut injection (cont.).

| CATCH | | D01 | D02 | D03 | D04 | D05 | D06 | D07 | D08 | D09 | D10 | D11 | D12 | D13 | D14 | D15 | D16 |
|---|---|---|---|---|---|---|---|---|---|---|---|---|---|---|---|---|---|
| Original | | 73.0 | 76.5 | 72.5 | 90.8 | 84.0 | 92.7 | 88.0 | 67.5 | 73.5 | 60.0 | 91.5 | 73.5 | 98.0 | 100.0 | 53.0 | 56.5 |
| Vanilla | | 14.0 | 14.5 | 16.0 | 30.3 | 13.8 | 10.8 | 34.4 | 8.2 | 9.0 | 7.1 | 23.9 | 10.5 | 33.8 | 35.7 | 3.6 | 5.5 |
| Explicit | Sentiment | 18.3 | 11.0 | 12.3 | 27.7 | 6.5 | 8.3 | 31.3 | 3.0 | 9.8 | 4.2 | 19.2 | 12.5 | 51.5 | 51.5 | 1.5 | 4.0 |
| | Tone | 14.3 | 15.0 | 19.6 | 35.4 | 10.5 | 6.2 | 22.7 | 5.0 | 10.8 | 6.0 | 22.8 | 5.5 | 18.2 | 23.8 | 11.2 | 12.0 |
| | Word Choice | 15.1 | 14.8 | 17.6 | 42.3 | 6.0 | 4.7 | 20.0 | 3.2 | 7.5 | 7.0 | 26.0 | 6.5 | 28.8 | 33.2 | 11.8 | 12.5 |
| Implicit | Age | 65.1 | 70.0 | 67.6 | 76.2 | 76.0 | 80.2 | 86.7 | 65.0 | 65.5 | 57.0 | 85.2 | 72.0 | 97.0 | 98.0 | 54.2 | 54.2 |
| | Gender | 66.7 | 73.8 | 73.5 | 83.1 | 77.0 | 85.4 | 89.3 | 67.0 | 67.2 | 57.2 | 86.8 | 72.0 | 96.2 | 98.8 | 51.0 | 48.2 |

Table 21: The complete detector performance after the **Paraphrase** mitigation method.

| | BERT | D01 | D02 | D03 | D04 | D05 | D06 | D07 | D08 | D09 | D10 | D11 | D12 | D13 | D14 | D15 | D16 |
|---|---|---|---|---|---|---|---|---|---|---|---|---|---|---|---|---|---|
| | **Vanilla** | 21.7 | 26.2 | 26.1 | 37.4 | 62.0 | 39.6 | 70.2 | 48.9 | 40.8 | 13.7 | 42.3 | 18.7 | 80.2 | 82.2 | 3.7 | 7.2 |
| **Explicit** | **Sentiment** | 15.1 | 8.2 | 15.2 | 12.3 | 11.0 | 9.4 | 32.7 | 5.5 | 12.5 | 3.2 | 17.0 | 8.8 | 37.0 | 41.0 | 1.8 | 4.5 |
| | **Tone** | 58.7 | 64.0 | 66.2 | 73.8 | 52.0 | 46.4 | 75.3 | 47.2 | 42.0 | 36.5 | 77.8 | 41.2 | 83.2 | 82.2 | 42.0 | 49.5 |
| | **Word Choice** | 58.7 | 64.5 | 67.2 | 75.4 | 65.5 | 53.6 | 83.3 | 59.8 | 57.5 | 50.0 | 84.2 | 55.5 | 86.8 | 87.5 | 29.8 | 34.8 |
| **Implicit** | **Age** | 55.6 | 73.8 | 73.0 | 77.7 | 74.5 | 63.0 | 88.7 | 67.5 | 63.2 | 57.5 | 88.0 | 71.5 | 88.8 | 92.0 | 51.2 | 52.2 |
| | **Gender** | 61.9 | 74.2 | 75.5 | 80.8 | 77.0 | 62.0 | 88.0 | 66.8 | 63.8 | 59.8 | 88.5 | 71.2 | 89.8 | 93.5 | 47.0 | 44.0 |
| | DEBERTA | D01 | D02 | D03 | D04 | D05 | D06 | D07 | D08 | D09 | D10 | D11 | D12 | D13 | D14 | D15 | D16 |
| | **Vanilla** | 16.7 | 16.1 | 13.1 | 20.8 | 42.7 | 45.3 | 65.6 | 31.2 | 28.7 | 9.9 | 24.1 | 12.9 | 71.1 | 75.8 | 3.4 | 6.5 |
| **Explicit** | **Sentiment** | 21.4 | 10.2 | 15.2 | 26.9 | 19.5 | 17.2 | 51.3 | 8.5 | 14.2 | 7.0 | 26.2 | 12.2 | 58.0 | 62.0 | 4.5 | 8.0 |
| | **Tone** | 49.2 | 49.5 | 47.1 | 49.2 | 46.5 | 69.3 | 74.7 | 35.0 | 34.2 | 31.5 | 70.8 | 34.2 | 81.8 | 83.0 | 40.8 | 43.0 |
| | **Word Choice** | 53.2 | 48.8 | 51.5 | 58.5 | 57.5 | 70.8 | 80.0 | 47.5 | 48.0 | 34.5 | 76.5 | 47.8 | 83.8 | 89.0 | 32.2 | 31.8 |
| **Implicit** | **Age** | 53.2 | 68.2 | 63.2 | 68.5 | 79.5 | 84.4 | 85.3 | 68.5 | 67.2 | 58.8 | 87.0 | 72.2 | 90.5 | 93.2 | 56.2 | 53.8 |
| | **Gender** | 56.3 | 69.0 | 66.2 | 73.8 | 79.5 | 79.7 | 85.3 | 64.5 | 63.2 | 61.5 | 88.0 | 69.0 | 91.5 | 95.0 | 53.5 | 54.2 |
| | CMTR | D01 | D02 | D03 | D04 | D05 | D06 | D07 | D08 | D09 | D10 | D11 | D12 | D13 | D14 | D15 | D16 |
| | **Vanilla** | 30.7 | 30.8 | 35.8 | 57.4 | 65.2 | 41.0 | 76.0 | 56.2 | 43.5 | 13.8 | 44.6 | 18.8 | 80.8 | 83.6 | 4.8 | 8.1 |
| **Explicit** | **Sentiment** | 19.8 | 10.2 | 21.6 | 21.5 | 11.5 | 10.4 | 50.0 | 7.2 | 13.5 | 4.8 | 20.0 | 8.5 | 39.0 | 42.0 | 2.8 | 5.5 |
| | **Tone** | 65.9 | 67.0 | 73.5 | 84.6 | 55.5 | 48.4 | 82.0 | 53.2 | 48.2 | 43.0 | 79.8 | 45.0 | 83.5 | 81.8 | 41.0 | 49.2 |
| | **Word Choice** | 59.5 | 68.8 | 73.5 | 76.2 | 67.0 | 52.6 | 82.0 | 64.2 | 60.8 | 52.0 | 86.0 | 58.2 | 88.2 | 88.2 | 32.0 | 35.2 |
| **Implicit** | **Age** | 67.5 | 74.8 | 72.1 | 79.2 | 75.0 | 63.5 | 82.7 | 67.2 | 62.8 | 59.2 | 88.5 | 72.5 | 88.8 | 89.8 | 49.5 | 49.2 |
| | **Gender** | 71.4 | 73.5 | 73.5 | 77.7 | 75.0 | 61.5 | 78.7 | 65.2 | 61.2 | 58.5 | 88.0 | 73.2 | 88.0 | 90.8 | 47.8 | 46.0 |
| | DISC | D01 | D02 | D03 | D04 | D05 | D06 | D07 | D08 | D09 | D10 | D11 | D12 | D13 | D14 | D15 | D16 |
| | **Vanilla** | 22.8 | 25.6 | 25.8 | 41.0 | 62.3 | 39.6 | 71.3 | 49.3 | 40.2 | 13.8 | 36.0 | 17.2 | 80.4 | 82.2 | 3.8 | 6.9 |
| **Explicit** | **Sentiment** | 12.7 | 9.5 | 15.2 | 13.8 | 11.0 | 9.9 | 34.0 | 4.5 | 11.2 | 3.2 | 15.8 | 7.5 | 36.2 | 39.8 | 1.8 | 4.8 |
| | **Tone** | 61.1 | 62.0 | 65.7 | 73.8 | 51.5 | 39.6 | 76.0 | 36.8 | 34.5 | 31.8 | 76.5 | 23.8 | 82.5 | 81.5 | 41.0 | 46.8 |
| | **Word Choice** | 59.5 | 67.0 | 68.1 | 76.9 | 66.0 | 55.2 | 83.3 | 58.2 | 58.0 | 49.8 | 85.0 | 56.0 | 86.0 | 87.5 | 31.8 | 34.5 |
| **Implicit** | **Age** | 64.3 | 73.0 | 72.5 | 76.9 | 75.0 | 63.5 | 86.7 | 69.2 | 63.8 | 55.3 | 88.8 | 75.2 | 89.0 | 93.0 | 52.0 | 49.8 |
| | **Gender** | 69.0 | 76.0 | 75.5 | 81.5 | 76.5 | 60.9 | 87.3 | 69.8 | 67.0 | 58.8 | 88.2 | 70.8 | 89.8 | 92.8 | 47.2 | 46.5 |
| | CATCH | D01 | D02 | D03 | D04 | D05 | D06 | D07 | D08 | D09 | D10 | D11 | D12 | D13 | D14 | D15 | D16 |
| | **Vanilla** | 28.8 | 29.9 | 34.6 | 49.7 | 57.7 | 47.0 | 75.8 | 50.2 | 43.8 | 18.6 | 41.0 | 26.2 | 81.7 | 83.9 | 7.2 | 11.4 |
| **Explicit** | **Sentiment** | 25.4 | 17.8 | 26.0 | 32.3 | 12.5 | 14.6 | 48.7 | 11.5 | 21.8 | 8.5 | 29.8 | 21.2 | 47.2 | 51.0 | 6.8 | 8.2 |
| | **Tone** | 67.5 | 68.8 | 70.6 | 75.4 | 54.0 | 54.7 | 77.3 | 51.8 | 44.5 | 41.0 | 80.8 | 47.5 | 87.5 | 84.0 | 48.5 | 52.2 |
| | **Word Choice** | 72.2 | 71.0 | 69.1 | 78.5 | 66.0 | 60.9 | 82.7 | 62.8 | 60.8 | 52.0 | 86.5 | 61.8 | 86.8 | 86.8 | 38.8 | 47.0 |
| **Implicit** | **Age** | 73.0 | 73.5 | 72.1 | 73.8 | 72.0 | 62.5 | 81.3 | 66.8 | 63.2 | 58.5 | 86.2 | 72.5 | 90.5 | 89.8 | 54.5 | 54.2 |
| | **Gender** | 69.0 | 74.2 | 73.5 | 80.0 | 72.0 | 65.6 | 83.3 | 68.8 | 63.5 | 59.2 | 88.8 | 70.2 | 90.2 | 92.0 | 52.2 | 46.8 |

Table 22: The complete detector performance after the **Summary** mitigation method.

| | BERT | D01 | D02 | D03 | D04 | D05 | D06 | D07 | D08 | D09 | D10 | D11 | D12 | D13 | D14 | D15 | D16 |
|---|---|---|---|---|---|---|---|---|---|---|---|---|---|---|---|---|---|
| | **Vanilla** | 26.7 | 30.2 | 35.3 | 49.7 | 64.7 | 46.4 | 77.1 | 55.8 | 47.0 | 19.6 | 47.9 | 20.9 | 81.9 | 83.3 | 7.6 | 11.8 |
| **Explicit** | **Sentiment** | 42.9 | 32.2 | 42.2 | 36.9 | 39.0 | 31.8 | 73.3 | 35.5 | 38.2 | 25.8 | 42.2 | 16.8 | 65.8 | 73.0 | 18.5 | 25.5 |
| | **Tone** | 52.4 | 63.8 | 70.1 | 78.5 | 44.5 | 39.1 | 73.3 | 41.2 | 49.8 | 40.0 | 76.8 | 38.5 | 74.5 | 70.8 | 40.8 | 41.2 |
| | **Word Choice** | 51.6 | 53.8 | 67.2 | 69.2 | 30.0 | 23.4 | 64.0 | 22.0 | 34.0 | 34.5 | 74.0 | 32.0 | 56.5 | 56.0 | 28.7 | 26.0 |
| **Implicit** | **Age** | 73.8 | 73.2 | 73.5 | 79.2 | 69.5 | 63.5 | 90.7 | 68.0 | 63.0 | 55.0 | 86.2 | 70.8 | 87.0 | 89.8 | 53.5 | 50.8 |
| | **Gender** | 66.7 | 72.8 | 69.6 | 75.4 | 71.5 | 63.0 | 88.0 | 65.8 | 63.2 | 59.8 | 88.2 | 69.0 | 86.2 | 90.5 | 49.0 | 49.0 |
| | DEBERTA | D01 | D02 | D03 | D04 | D05 | D06 | D07 | D08 | D09 | D10 | D11 | D12 | D13 | D14 | D15 | D16 |
| | **Vanilla** | 20.9 | 21.4 | 18.5 | 32.6 | 46.7 | 56.4 | 73.3 | 40.7 | 34.0 | 15.4 | 31.8 | 15.4 | 75.6 | 79.2 | 7.6 | 10.2 |
| **Explicit** | **Sentiment** | 42.9 | 35.5 | 43.6 | 42.3 | 48.5 | 49.5 | 72.7 | 35.2 | 40.8 | 28.2 | 58.0 | 20.0 | 77.8 | 79.0 | 22.2 | 31.0 |
| | **Tone** | 52.4 | 51.0 | 60.3 | 69.2 | 35.5 | 45.3 | 63.3 | 25.8 | 37.2 | 35.8 | 70.8 | 28.8 | 62.0 | 60.2 | 40.8 | 39.5 |
| | **Word Choice** | 36.5 | 36.2 | 45.1 | 46.2 | 25.0 | 27.6 | 41.3 | 12.8 | 20.8 | 29.0 | 57.2 | 22.5 | 43.5 | 41.5 | 27.8 | 25.0 |
| **Implicit** | **Age** | 50.8 | 63.8 | 64.2 | 70.0 | 71.5 | 82.8 | 81.3 | 60.0 | 62.2 | 55.8 | 84.0 | 70.5 | 91.2 | 93.5 | 53.8 | 56.0 |
| | **Gender** | 50.8 | 67.8 | 58.8 | 70.0 | 73.0 | 79.7 | 82.7 | 64.0 | 63.0 | 57.5 | 88.2 | 67.2 | 92.8 | 94.5 | 54.0 | 53.2 |
| | CMTR | D01 | D02 | D03 | D04 | D05 | D06 | D07 | D08 | D09 | D10 | D11 | D12 | D13 | D14 | D15 | D16 |
| | **Vanilla** | 32.0 | 31.8 | 41.8 | 64.1 | 62.8 | 46.2 | 79.6 | 61.0 | 51.1 | 22.6 | 52.8 | 23.1 | 82.0 | 84.7 | 7.7 | 13.2 |
| **Explicit** | **Sentiment** | 41.3 | 36.5 | 48.5 | 52.3 | 41.0 | 36.5 | 72.0 | 41.0 | 42.2 | 26.2 | 48.5 | 17.8 | 68.5 | 73.8 | 18.5 | 26.8 |
| | **Tone** | 58.7 | 68.8 | 72.5 | 80.0 | 48.0 | 36.5 | 76.7 | 45.2 | 53.5 | 43.0 | 79.0 | 38.8 | 73.8 | 73.8 | 45.2 | 41.5 |
| | **Word Choice** | 51.6 | 57.2 | 73.0 | 75.4 | 32.5 | 26.0 | 63.3 | 26.2 | 35.5 | 37.5 | 73.0 | 33.5 | 61.0 | 56.7 | 32.2 | 30.5 |
| **Implicit** | **Age** | 72.2 | 73.5 | 76.5 | 82.3 | 71.5 | 62.0 | 90.7 | 70.0 | 64.2 | 58.5 | 85.8 | 69.5 | 86.5 | 89.5 | 55.0 | 51.2 |
| | **Gender** | 73.0 | 73.0 | 72.1 | 77.7 | 70.0 | 62.0 | 86.0 | 67.8 | 66.0 | 59.2 | 86.0 | 69.2 | 86.8 | 91.0 | 50.8 | 50.5 |
| | DISC | D01 | D02 | D03 | D04 | D05 | D06 | D07 | D08 | D09 | D10 | D11 | D12 | D13 | D14 | D15 | D16 |
| | **Vanilla** | 27.2 | 30.4 | 34.2 | 47.9 | 63.2 | 45.5 | 78.2 | 55.5 | 46.0 | 20.2 | 47.1 | 23.4 | 83.0 | 83.8 | 8.1 | 12.8 |
| **Explicit** | **Sentiment** | 37.3 | 32.5 | 43.1 | 41.5 | 39.0 | 32.3 | 72.0 | 35.0 | 38.2 | 23.8 | 45.0 | 16.2 | 66.0 | 72.5 | 19.0 | 25.8 |
| | **Tone** | 57.1 | 65.0 | 72.1 | 80.8 | 40.5 | 35.9 | 76.7 | 40.2 | 50.2 | 38.8 | 76.8 | 39.0 | 72.5 | 68.0 | 46.5 | 42.2 |
| | **Word Choice** | 52.4 | 57.2 | 69.1 | 69.2 | 30.5 | 22.9 | 64.7 | 21.5 | 34.0 | 35.0 | 72.8 | 30.8 | 58.0 | 54.0 | 29.5 | 29.0 |
| **Implicit** | **Age** | 74.6 | 73.0 | 74.0 | 79.2 | 72.5 | 62.0 | 88.7 | 67.8 | 63.8 | 56.0 | 86.0 | 70.0 | 87.5 | 91.5 | 54.5 | 52.0 |
| | **Gender** | 69.8 | 74.2 | 71.6 | 76.9 | 70.5 | 61.5 | 87.3 | 66.0 | 63.2 | 60.0 | 86.5 | 68.5 | 86.8 | 92.5 | 50.8 | 50.8 |
| | CATCH | D01 | D02 | D03 | D04 | D05 | D06 | D07 | D08 | D09 | D10 | D11 | D12 | D13 | D14 | D15 | D16 |
| | **Vanilla** | 41.0 | 35.3 | 40.8 | 61.0 | 60.5 | 49.1 | 78.4 | 58.8 | 49.9 | 29.4 | 51.8 | 32.8 | 82.5 | 82.7 | 12.6 | 17.8 |
| **Explicit** | **Sentiment** | 57.1 | 45.8 | 52.9 | 66.2 | 48.0 | 41.7 | 78.7 | 44.8 | 43.8 | 33.0 | 62.0 | 29.8 | 77.8 | 78.2 | 22.8 | 34.0 |
| | **Tone** | 67.5 | 69.2 | 72.1 | 79.2 | 50.0 | 45.3 | 80.0 | 52.5 | 53.2 | 53.8 | 81.0 | 46.2 | 80.8 | 78.2 | 50.0 | 51.2 |
| | **Word Choice** | 65.1 | 64.8 | 72.1 | 76.2 | 42.5 | 41.7 | 72.7 | 35.5 | 44.2 | 47.5 | 80.5 | 44.8 | 76.2 | 74.0 | 41.2 | 41.2 |
| **Implicit** | **Age** | 72.2 | 70.5 | 71.6 | 77.7 | 68.0 | 64.1 | 84.0 | 67.5 | 64.5 | 55.0 | 86.0 | 71.5 | 87.2 | 87.5 | 54.5 | 54.0 |
| | **Gender** | 69.0 | 73.2 | 74.0 | 78.5 | 69.5 | 64.6 | 85.3 | 67.0 | 65.0 | 55.3 | 87.0 | 68.0 | 88.5 | 89.2 | 53.2 | 52.5 |

Table 23: The complete detector performance after the **Neutral** mitigation method.

| BERT | | D01 | D02 | D03 | D04 | D05 | D06 | D07 | D08 | D09 | D10 | D11 | D12 | D13 | D14 | D15 | D16 |
|---|---|---|---|---|---|---|---|---|---|---|---|---|---|---|---|---|---|
| **Vanilla** | | 17.2 | 23.9 | 24.5 | 34.4 | 32.8 | 24.8 | 61.1 | 28.1 | 26.0 | 10.9 | 32.8 | 15.1 | 65.7 | 70.2 | 4.5 | 5.9 |
| **Explicit** | **Sentiment** | 37.3 | 22.2 | 23.5 | 36.2 | 38.5 | 26.6 | 67.3 | 32.2 | 31.0 | 15.8 | 37.5 | 13.5 | 71.8 | 77.0 | 9.5 | 18.2 |
| | **Tone** | 61.9 | 68.5 | 75.0 | 73.1 | 45.0 | 38.0 | 76.7 | 36.8 | 46.2 | 42.0 | 81.0 | 35.5 | 75.8 | 72.2 | 44.0 | 41.5 |
| | **Word Choice** | 52.4 | 55.5 | 59.8 | 68.5 | 37.5 | 33.9 | 71.3 | 29.0 | 33.2 | 34.8 | 73.5 | 30.2 | 76.0 | 77.2 | 25.8 | 27.5 |
| **Implicit** | **Age** | 62.7 | 72.0 | 73.0 | 72.3 | 74.0 | 68.8 | 88.0 | 66.8 | 62.8 | 54.5 | 85.2 | 70.2 | 93.0 | 96.2 | 55.8 | 53.8 |
| | **Gender** | 69.8 | 72.5 | 72.1 | 73.1 | 72.5 | 63.5 | 88.0 | 66.2 | 62.5 | 57.5 | 88.8 | 68.0 | 94.0 | 94.8 | 49.8 | 44.0 |
| DEBERTA | | D01 | D02 | D03 | D04 | D05 | D06 | D07 | D08 | D09 | D10 | D11 | D12 | D13 | D14 | D15 | D16 |
| **Vanilla** | | 15.3 | 14.1 | 11.8 | 21.3 | 15.0 | 21.7 | 49.8 | 15.3 | 17.2 | 9.5 | 23.2 | 13.4 | 49.9 | 54.5 | 3.8 | 4.7 |
| **Explicit** | **Sentiment** | 46.0 | 28.5 | 30.9 | 46.2 | 49.0 | 50.0 | 75.3 | 32.5 | 31.5 | 23.2 | 44.8 | 24.0 | 96.8 | 96.5 | 11.5 | 18.0 |
| | **Tone** | 51.6 | 54.8 | 52.9 | 66.2 | 36.0 | 40.1 | 65.3 | 25.8 | 34.0 | 35.8 | 73.0 | 34.8 | 75.2 | 79.5 | 43.5 | 41.8 |
| | **Word Choice** | 35.7 | 43.5 | 40.2 | 45.4 | 23.5 | 39.6 | 65.3 | 14.2 | 27.0 | 25.5 | 62.0 | 27.2 | 75.5 | 86.0 | 25.8 | 26.8 |
| **Implicit** | **Age** | 58.7 | 68.0 | 60.8 | 68.5 | 77.0 | 83.3 | 84.0 | 67.8 | 68.0 | 56.0 | 85.5 | 72.0 | 96.8 | 98.2 | 56.2 | 54.8 |
| | **Gender** | 62.7 | 70.0 | 59.8 | 77.7 | 77.0 | 82.8 | 88.0 | 66.2 | 68.0 | 54.8 | 87.0 | 69.2 | 96.2 | 98.2 | 54.5 | 52.2 |
| CMTR | | D01 | D02 | D03 | D04 | D05 | D06 | D07 | D08 | D09 | D10 | D11 | D12 | D13 | D14 | D15 | D16 |
| **Vanilla** | | 25.1 | 27.0 | 34.8 | 48.2 | 38.7 | 29.0 | 75.1 | 34.2 | 31.4 | 13.9 | 37.9 | 18.7 | 69.8 | 76.3 | 3.9 | 6.7 |
| **Explicit** | **Sentiment** | 40.5 | 31.0 | 39.2 | 46.2 | 43.5 | 31.8 | 70.0 | 37.8 | 40.8 | 17.8 | 44.0 | 16.2 | 75.8 | 75.8 | 11.0 | 19.2 |
| | **Tone** | 65.1 | 66.2 | 75.0 | 79.2 | 50.0 | 38.5 | 77.3 | 44.5 | 55.5 | 44.2 | 80.2 | 40.5 | 77.5 | 76.0 | 42.8 | 43.5 |
| | **Word Choice** | 54.8 | 62.0 | 66.7 | 74.6 | 37.5 | 34.9 | 76.0 | 34.0 | 38.8 | 39.5 | 79.2 | 34.8 | 79.2 | 77.0 | 28.0 | 29.2 |
| **Implicit** | **Age** | 74.6 | 71.5 | 76.5 | 76.9 | 69.5 | 66.1 | 85.3 | 65.0 | 64.2 | 57.5 | 87.5 | 71.5 | 91.8 | 94.8 | 53.8 | 51.8 |
| | **Gender** | 69.0 | 72.2 | 65.2 | 80.8 | 68.0 | 62.0 | 88.7 | 67.0 | 61.2 | 57.0 | 89.0 | 73.0 | 91.8 | 93.8 | 47.5 | 44.5 |
| DISC | | D01 | D02 | D03 | D04 | D05 | D06 | D07 | D08 | D09 | D10 | D11 | D12 | D13 | D14 | D15 | D16 |
| **Vanilla** | | 20.1 | 23.8 | 25.3 | 34.4 | 29.3 | 21.5 | 64.9 | 22.0 | 19.6 | 10.8 | 35.6 | 16.4 | 56.1 | 60.3 | 3.8 | 5.4 |
| **Explicit** | **Sentiment** | 34.1 | 25.0 | 23.5 | 38.5 | 37.5 | 28.6 | 65.3 | 29.5 | 19.8 | 16.8 | 39.0 | 13.8 | 89.8 | 91.8 | 7.2 | 17.5 |
| | **Tone** | 62.7 | 68.8 | 75.5 | 73.8 | 44.0 | 37.0 | 75.3 | 36.5 | 45.0 | 41.8 | 80.8 | 37.0 | 77.5 | 73.2 | 45.5 | 44.0 |
| | **Word Choice** | 54.0 | 57.7 | 61.8 | 66.9 | 38.5 | 32.8 | 70.0 | 30.8 | 32.8 | 34.8 | 73.5 | 28.8 | 78.5 | 77.8 | 28.5 | 28.0 |
| **Implicit** | **Age** | 72.2 | 73.2 | 76.5 | 77.7 | 75.5 | 65.1 | 87.3 | 66.8 | 62.8 | 57.0 | 86.0 | 70.2 | 95.5 | 98.2 | 55.0 | 53.0 |
| | **Gender** | 74.6 | 73.8 | 75.0 | 71.5 | 75.0 | 65.1 | 88.7 | 67.2 | 62.5 | 56.5 | 89.2 | 67.0 | 96.0 | 98.0 | 51.0 | 45.5 |
| CATCH | | D01 | D02 | D03 | D04 | D05 | D06 | D07 | D08 | D09 | D10 | D11 | D12 | D13 | D14 | D15 | D16 |
| **Vanilla** | | 21.7 | 22.8 | 24.2 | 33.3 | 20.7 | 18.8 | 47.3 | 19.0 | 17.8 | 15.3 | 35.5 | 20.5 | 48.2 | 52.6 | 5.6 | 8.6 |
| **Explicit** | **Sentiment** | 48.4 | 37.0 | 39.7 | 56.9 | 41.5 | 40.6 | 77.3 | 41.8 | 40.5 | 25.2 | 57.8 | 31.0 | 90.8 | 92.0 | 17.0 | 24.2 |
| | **Tone** | 65.1 | 66.0 | 74.0 | 72.3 | 52.0 | 42.2 | 76.7 | 42.2 | 53.8 | 43.5 | 81.0 | 40.8 | 81.8 | 79.8 | 48.2 | 48.2 |
| | **Word Choice** | 65.1 | 64.2 | 63.2 | 76.9 | 48.0 | 50.5 | 76.0 | 38.5 | 45.2 | 42.2 | 81.8 | 42.5 | 87.5 | 88.5 | 37.5 | 35.0 |
| **Implicit** | **Age** | 65.9 | 73.0 | 72.1 | 79.2 | 75.5 | 68.8 | 84.7 | 66.5 | 63.5 | 56.0 | 87.8 | 68.2 | 96.2 | 96.8 | 55.2 | 51.5 |
| | **Gender** | 69.0 | 71.5 | 67.6 | 81.5 | 77.0 | 70.3 | 86.0 | 64.8 | 61.2 | 55.2 | 87.0 | 69.8 | 95.2 | 97.2 | 49.2 | 48.0 |

Table 24: The performance of detectors after injection under the standard setting.

| MISTRALV3 | | D01 | D02 | D03 | D04 | D05 | D06 | D07 | D08 | D09 | D10 | D11 | D12 | D13 | D14 | D15 | D16 |
|---|---|---|---|---|---|---|---|---|---|---|---|---|---|---|---|---|---|
| **Original** | | 69.8 | 54.0 | 34.3 | 30.8 | 70.0 | 74.0 | 49.3 | 43.0 | 49.5 | 49.5 | 72.5 | 55.0 | 42.0 | 60.0 | 58.5 | 50.0 |
| **Vanilla** | | 65.6 | 56.2 | 37.3 | 53.3 | 55.3 | 64.2 | 42.7 | 44.2 | 53.5 | 45.2 | 62.0 | 54.8 | 44.3 | 59.2 | 53.0 | 51.5 |
| **Explicit** | **Sentiment** | 61.9 | 53.8 | 38.7 | 58.5 | 51.5 | 57.8 | 42.7 | 44.8 | 53.0 | 47.5 | 57.0 | 54.2 | 49.2 | 52.5 | 54.2 | 53.5 |
| | **Tone** | 65.9 | 57.0 | 33.8 | 46.2 | 57.0 | 65.1 | 53.3 | 45.0 | 48.5 | 49.0 | 69.2 | 63.2 | 42.5 | 60.8 | 58.8 | 53.2 |
| | **Word Choice** | 65.1 | 51.5 | 36.8 | 53.1 | 58.5 | 67.7 | 52.0 | 46.0 | 51.0 | 51.2 | 72.5 | 56.5 | 42.5 | 59.5 | 55.5 | 50.0 |
| **Implicit** | **Age** | 65.9 | 53.5 | 40.7 | 53.1 | 61.5 | 70.3 | 54.0 | 40.5 | 53.8 | 46.8 | 68.2 | 44.2 | 44.5 | 61.5 | 59.8 | 53.8 |
| | **Gender** | 64.3 | 53.8 | 37.7 | 55.4 | 54.5 | 71.4 | 48.0 | 41.2 | 55.8 | 45.8 | 66.0 | 42.0 | 44.5 | 60.0 | 54.2 | 53.8 |

Table 25: The performance of detectors after injection under the standard setting (cont.).

| LLAMA3 | | D01 | D02 | D03 | D04 | D05 | D06 | D07 | D08 | D09 | D10 | D11 | D12 | D13 | D14 | D15 | D16 |
|---|---|---|---|---|---|---|---|---|---|---|---|---|---|---|---|---|---|
| Original | | 73.0 | 49.5 | 40.2 | 41.5 | 77.0 | 85.4 | 70.7 | 49.5 | 45.5 | 58.0 | 74.0 | 66.5 | 30.5 | 71.0 | 56.0 | 53.0 |
| Vanilla | | 57.1 | 54.8 | 45.1 | 52.8 | 63.3 | 74.7 | 61.3 | 47.3 | 49.7 | 52.2 | 51.5 | 57.0 | 39.7 | 60.2 | 57.3 | 52.3 |
| Explicit | Sentiment | 56.3 | 56.8 | 45.6 | 54.6 | 65.0 | 65.1 | 56.0 | 43.8 | 49.2 | 49.0 | 50.2 | 50.8 | 43.0 | 51.0 | 54.2 | 50.8 |
| | Tone | 63.5 | 50.0 | 47.5 | 48.5 | 67.0 | 81.8 | 61.3 | 43.5 | 44.0 | 53.8 | 65.8 | 64.8 | 39.2 | 62.2 | 58.8 | 50.5 |
| | Word Choice | 58.7 | 51.5 | 41.7 | 50.0 | 68.0 | 79.2 | 66.7 | 47.2 | 46.2 | 51.2 | 66.2 | 67.0 | 35.2 | 67.0 | 59.0 | 54.8 |
| Implicit | Age | 63.5 | 50.0 | 44.6 | 53.1 | 72.5 | 78.1 | 48.7 | 44.0 | 54.5 | 51.8 | 52.0 | 40.2 | 37.2 | 62.5 | 56.8 | 51.5 |
| | Gender | 62.7 | 49.0 | 44.1 | 56.9 | 71.0 | 73.4 | 46.7 | 49.0 | 56.2 | 52.5 | 51.2 | 40.0 | 36.5 | 62.0 | 52.5 | 53.5 |

| BERT | | D01 | D02 | D03 | D04 | D05 | D06 | D07 | D08 | D09 | D10 | D11 | D12 | D13 | D14 | D15 | D16 |
|---|---|---|---|---|---|---|---|---|---|---|---|---|---|---|---|---|---|
| Original | | 74.6 | 80.0 | 86.3 | 84.6 | 78.0 | 78.1 | 88.0 | 72.5 | 72.5 | 60.5 | 90.5 | 76.5 | 96.5 | 98.5 | 58.5 | 54.5 |
| Vanilla | | 79.4 | 76.5 | 84.6 | 88.2 | 76.0 | 56.6 | 85.3 | 70.3 | 69.0 | 56.8 | 92.0 | 73.2 | 91.5 | 90.7 | 54.2 | 53.8 |
| Explicit | Sentiment | 60.3 | 77.0 | 81.9 | 79.2 | 75.0 | 63.5 | 84.0 | 69.0 | 66.5 | 55.8 | 89.2 | 74.2 | 94.5 | 94.2 | 54.5 | 52.2 |
| | Tone | 73.8 | 78.2 | 83.8 | 86.2 | 78.0 | 63.0 | 80.7 | 69.5 | 68.0 | 58.0 | 88.8 | 71.8 | 92.2 | 92.0 | 55.0 | 53.2 |
| | Word Choice | 76.2 | 77.0 | 81.4 | 81.5 | 74.5 | 58.9 | 84.7 | 69.8 | 68.0 | 57.8 | 90.5 | 71.0 | 93.2 | 94.2 | 56.0 | 54.8 |
| Implicit | Age | 50.8 | 74.8 | 77.0 | 81.5 | 82.5 | 72.9 | 90.0 | 69.0 | 68.0 | 61.5 | 88.5 | 74.5 | 95.0 | 96.8 | 54.2 | 53.0 |
| | Gender | 74.6 | 73.2 | 78.4 | 78.5 | 82.0 | 75.0 | 90.7 | 67.2 | 68.0 | 62.5 | 91.0 | 74.0 | 95.5 | 97.2 | 53.2 | 54.5 |

| DEBERTA | | D01 | D02 | D03 | D04 | D05 | D06 | D07 | D08 | D09 | D10 | D11 | D12 | D13 | D14 | D15 | D16 |
|---|---|---|---|---|---|---|---|---|---|---|---|---|---|---|---|---|---|
| Original | | 68.3 | 75.0 | 74.5 | 86.2 | 85.0 | 99.0 | 90.7 | 65.5 | 77.5 | 63.5 | 93.0 | 79.0 | 99.0 | 99.0 | 66.0 | 61.5 |
| Vanilla | | 59.3 | 74.3 | 74.8 | 82.1 | 83.3 | 80.2 | 87.6 | 67.2 | 69.0 | 57.3 | 90.5 | 73.3 | 93.2 | 95.3 | 56.8 | 54.2 |
| Explicit | Sentiment | 59.5 | 69.8 | 70.1 | 73.8 | 75.5 | 78.1 | 86.7 | 68.8 | 70.2 | 57.5 | 89.5 | 73.0 | 98.2 | 99.2 | 55.8 | 54.0 |
| | Tone | 69.8 | 75.8 | 75.0 | 79.2 | 78.0 | 84.4 | 85.3 | 68.0 | 75.0 | 58.2 | 88.8 | 75.5 | 94.8 | 96.5 | 57.2 | 58.8 |
| | Word Choice | 64.3 | 74.0 | 76.0 | 82.3 | 77.5 | 80.2 | 86.0 | 66.8 | 74.8 | 58.8 | 88.2 | 75.5 | 96.0 | 98.5 | 58.2 | 56.2 |
| Implicit | Age | 51.6 | 73.0 | 74.0 | 79.2 | 83.5 | 95.8 | 89.3 | 67.5 | 71.0 | 62.5 | 88.5 | 74.8 | 97.5 | 98.5 | 57.2 | 58.5 |
| | Gender | 52.4 | 71.2 | 71.1 | 74.6 | 82.0 | 93.8 | 83.3 | 64.8 | 71.2 | 60.2 | 89.5 | 74.0 | 97.5 | 98.8 | 54.5 | 56.8 |

| CMTR | | D01 | D02 | D03 | D04 | D05 | D06 | D07 | D08 | D09 | D10 | D11 | D12 | D13 | D14 | D15 | D16 |
|---|---|---|---|---|---|---|---|---|---|---|---|---|---|---|---|---|---|
| Original | | 77.8 | 80.0 | 85.3 | 83.1 | 77.0 | 69.8 | 78.7 | 71.5 | 69.0 | 63.0 | 90.0 | 74.5 | 97.5 | 98.5 | 57.0 | 58.0 |
| Vanilla | | 76.7 | 75.5 | 82.0 | 84.6 | 71.3 | 55.6 | 85.3 | 70.2 | 66.0 | 59.2 | 92.2 | 72.3 | 88.7 | 90.0 | 55.8 | 53.5 |
| Explicit | Sentiment | 61.9 | 76.0 | 73.0 | 83.8 | 69.0 | 55.7 | 82.7 | 68.0 | 64.0 | 57.2 | 90.2 | 73.0 | 93.2 | 93.5 | 54.8 | 53.2 |
| | Tone | 68.3 | 76.2 | 84.3 | 86.9 | 75.0 | 58.3 | 84.7 | 67.2 | 65.5 | 56.5 | 89.8 | 74.5 | 91.8 | 92.5 | 53.0 | 53.5 |
| | Word Choice | 79.4 | 76.5 | 82.4 | 80.0 | 70.5 | 58.3 | 86.7 | 65.2 | 67.5 | 59.8 | 90.2 | 71.0 | 93.0 | 93.8 | 58.0 | 53.5 |
| Implicit | Age | 70.6 | 73.0 | 73.5 | 81.5 | 78.5 | 70.8 | 85.3 | 69.5 | 64.8 | 57.8 | 88.0 | 75.5 | 94.8 | 96.8 | 53.0 | 53.5 |
| | Gender | 77.8 | 75.5 | 77.5 | 78.5 | 76.0 | 64.6 | 88.7 | 67.8 | 63.5 | 62.2 | 90.0 | 73.5 | 96.0 | 96.8 | 53.0 | 53.0 |

| DISC | | D01 | D02 | D03 | D04 | D05 | D06 | D07 | D08 | D09 | D10 | D11 | D12 | D13 | D14 | D15 | D16 |
|---|---|---|---|---|---|---|---|---|---|---|---|---|---|---|---|---|---|
| Original | | 73.0 | 79.0 | 85.3 | 89.2 | 81.0 | 78.1 | 85.3 | 69.5 | 72.0 | 60.5 | 90.5 | 78.0 | 98.0 | 98.5 | 59.5 | 56.5 |
| Vanilla | | 78.3 | 77.2 | 84.3 | 88.7 | 78.3 | 58.7 | 85.8 | 68.7 | 66.3 | 57.3 | 91.8 | 74.5 | 92.7 | 93.0 | 52.5 | 52.5 |
| Explicit | Sentiment | 63.5 | 77.8 | 81.4 | 78.5 | 74.5 | 61.5 | 84.0 | 66.8 | 67.0 | 56.8 | 89.5 | 72.5 | 98.0 | 99.2 | 54.0 | 53.5 |
| | Tone | 72.2 | 77.8 | 83.8 | 86.9 | 76.5 | 59.9 | 81.3 | 70.2 | 67.8 | 58.5 | 88.5 | 73.0 | 95.0 | 94.0 | 55.0 | 53.0 |
| | Word Choice | 79.4 | 76.8 | 83.3 | 83.1 | 75.0 | 55.7 | 82.7 | 71.0 | 66.5 | 57.0 | 90.5 | 71.0 | 97.2 | 97.0 | 55.5 | 52.8 |
| Implicit | Age | 61.1 | 73.8 | 77.9 | 80.8 | 82.5 | 72.4 | 88.0 | 70.5 | 67.2 | 61.5 | 88.2 | 76.0 | 96.5 | 97.5 | 52.5 | 53.2 |
| | Gender | 77.0 | 73.2 | 74.0 | 81.5 | 82.5 | 75.5 | 89.3 | 66.2 | 67.5 | 61.8 | 90.2 | 72.5 | 96.8 | 97.5 | 54.2 | 54.5 |

| CATCH | | D01 | D02 | D03 | D04 | D05 | D06 | D07 | D08 | D09 | D10 | D11 | D12 | D13 | D14 | D15 | D16 |
|---|---|---|---|---|---|---|---|---|---|---|---|---|---|---|---|---|---|
| Original | | 73.0 | 76.5 | 72.5 | 90.8 | 84.0 | 92.7 | 88.0 | 67.5 | 73.5 | 60.0 | 91.5 | 73.5 | 98.0 | 100.0 | 53.0 | 56.5 |
| Vanilla | | 75.1 | 76.3 | 81.0 | 84.1 | 74.0 | 59.0 | 87.6 | 66.7 | 63.0 | 57.8 | 91.3 | 73.7 | 91.7 | 90.3 | 55.0 | 55.7 |
| Explicit | Sentiment | 71.4 | 75.2 | 76.0 | 82.3 | 74.0 | 67.7 | 86.0 | 67.5 | 70.2 | 58.0 | 88.2 | 72.5 | 96.8 | 96.5 | 55.8 | 53.8 |
| | Tone | 73.0 | 76.8 | 77.9 | 83.1 | 74.5 | 66.1 | 84.7 | 67.8 | 67.0 | 55.5 | 89.5 | 71.5 | 93.8 | 92.0 | 55.8 | 55.5 |
| | Word Choice | 76.2 | 76.0 | 79.9 | 82.3 | 76.0 | 69.3 | 87.3 | 67.0 | 65.2 | 56.0 | 88.0 | 70.2 | 97.0 | 96.5 | 53.5 | 60.0 |
| Implicit | Age | 67.5 | 73.2 | 74.0 | 80.0 | 79.5 | 82.3 | 86.7 | 68.5 | 68.0 | 60.5 | 86.8 | 72.2 | 96.8 | 97.5 | 56.8 | 54.2 |
| | Gender | 72.2 | 72.5 | 73.0 | 82.3 | 76.0 | 83.9 | 86.0 | 66.5 | 66.8 | 56.2 | 87.8 | 74.2 | 96.8 | 98.8 | 53.2 | 56.8 |

