# OpenReview forum: "Truth over Tricks: Measuring and Mitigating Shortcut Learning in Misinformation Detection"
_NeurIPS.cc/2025/Conference — NeurIPS 2025 poster_

### Official Review · Reviewer_DwvC · 2025-06-26

**Clarity:** 3
**Significance:** 2
**Originality:** 3
**Rating:** 5
**Confidence:** 4

**Summary:**

This paper investigates how biases (i.e, shortcuts) in misinformation classification datasets deteriorate the performance. It proceeds to propose a method using LLM data augmentation to counteract these biases. Overall, it supports its claims by studying various benchmarks and models.

**Questions:**

See above

**Ethical Concerns:**

["NO or VERY MINOR ethics concerns only"]

**Final Justification:**

The responses strengthen my belief that this paper should be accepted.

**Limitations:**

see above

**Quality:**

3

**Strengths And Weaknesses:**

I like the paper and the paper setup. The paper presents a thorough overall evaluation. It employs human evaluation for the critical parts of the paper, namely injection fidelity and authenticity, when employing LLMs to alter texts. It also employs a vast number of models and benchmarks to support its main claims. This leads me to believe that the claims made in the paper are actually valid.

However, I have to outline one major weakness. While the buildup of the hypothesis is very good (i.e., chapters 1-3), it is unfortunate to leave out LLMs in the Shortcut Mitigation Framework (SMF) (chapter 4). Your argumentation states that LLM-based detectors are already robust to shortcuts. This opens up two problems. First, while your method might solve a lot of issues for the other models, it might also introduce some areas where, actually, these "shortcuts"/cues are the only indicators. Thus, your method might decrease the scores for LLMs or cause other problems we don't know of yet. This should be investigated. It is not a big problem because then SMF is only suitable for simpler methods but it would reveal interesting dynamics when using SMF. I assume that discovering the up- and downsides of SMF more thoroughly would help in inspiring the community or yourselves to come up with an even better approach. Second, leaving out LLMs basically means that we do not know whether your method improves the SOTA. While it is not critical that your method improves the SOTA against LLMs, it is crucial for users to understand this. I would recommend exploring this. SMF is a major contribution here, and I think it is not yet fully explored.

A minor point that would be interesting is how you deal with the fact that the SMF might reject rephrasing misinformation (as indicated in lines 138-140). What does this mean for SMF as a pipeline? Do you disregard these claims?

My conclusion: While I believe in the points made, I think important to understand the driving factors of the SMF and LLM investigations may reveal that. Does SMF solve some problems, but cause others? I believe all claims made, but those that are not made are very important as well.

---

> ### Author Rebuttal · Authors · 2025-07-30
>
> Thank you for your thoughtful and constructive feedback. We're glad you appreciate our evaluation setup and comprehensive empirical results. We also appreciate your close reading of the SMF (Shortcut Mitigation Framework) and your insight that more fully exploring its boundaries would help strengthen its impact.
>
> > However, I have to outline one major weakness. While the buildup of the hypothesis is very good (i.e., chapters 1-3), it is unfortunate to leave out LLMs in the Shortcut Mitigation Framework (SMF) (chapter 4). Your argumentation states that LLM-based detectors are already robust to shortcuts. This opens up two problems. First, while your method might solve a lot of issues for the other models, it might also introduce some areas where, actually, these "shortcuts"/cues are the only indicators. Thus, your method might decrease the scores for LLMs or cause other problems we don't know of yet. This should be investigated. It is not a big problem because then SMF is only suitable for simpler methods but it would reveal interesting dynamics when using SMF. I assume that discovering the up- and downsides of SMF more thoroughly would help in inspiring the community or yourselves to come up with an even better approach. Second, leaving out LLMs basically means that we do not know whether your method improves the SOTA. While it is not critical that your method improves the SOTA against LLMs, it is crucial for users to understand this. I would recommend exploring this. SMF is a major contribution here, and I think it is not yet fully explored.
>
>
> Thanks for raising this valuable point regarding the omission of LLM-based detectors from the SMF experiments. In response, we have extended our evaluation to include Mistral-v3 and LLaMA-3 under the SMF setup. The results are summarized below:
>
> ||Mistralv3|Llama3|
> |---|---|---|
> |Vanilla w/o aug.|52.6|54.8|
> |Vanilla paraphrase|52.2|55.5|
> |Vanilla Summary|51.5|54.2|
> |Vanilla Neutral|51.9|52.6|
> |Sentiment w/o aug.|51.9|52.6|
> |Sentiment paraphrase|51.0|52.8|
> |Sentiment Summary|52.0|53.0|
> |Sentiment Neutral|51.0|51.6|
> |Tone w/o aug.|54.3|56.4|
> |Tone paraphrase|53.0|55.9|
> |Tone Summary|52.2|54.2|
> |Tone Neutral|52.7|54.3|
> |Word Choice w/o aug.|54.3|56.9|
> |Word Choice paraphrase|52.8|55.5|
> |Word Choice Summary|51.8|53.8|
> |Word Choice Neutral|51.3|53.6|
> |Age w/o aug.|54.5|53.8|
> |Age paraphrase|53.1|53.5|
> |Age Summary|51.2|52.1|
> |Age Neutral|54.0|52.8|
> |Gender w/o aug.|53.0|53.6|
> |Gender paraphrase|53.3|52.9|
> |Gender Summary|51.2|52.5|
> |Gender Neutral|53.7|52.0|
>
>
> These results show that applying SMF to LLM-based detectors does not lead to significant gains or drops in performance. This supports our original hypothesis: LLM-based detectors are generally less affected by shortcut-based perturbations, likely due to their more holistic text understanding. However, your suggestion to include them was highly valuable—it provides a more complete picture and reinforces SMF's scope and limitations. We will include this analysis in the revised manuscript to help clarify that SMF offers more substantial benefits for non-LLM detectors, while also remaining compatible with LLM-based models.
>
> We agree that in some cases, the so-called "shortcuts" (such as sentiment or tone) may contain useful signals, especially when core content is limited or ambiguous. This concern highlights why SMF should not be treated as a blanket filter for all stylistic features, but as a flexible tool that helps practitioners test and improve robustness when shortcut learning is likely. Our experiments show that SMF helps models avoid over-relying on such cues when they are misleading, but we acknowledge that more subtle shortcut utility remains an open research direction. We will add discussions on this trade-off in the revised manuscript to help guide practitioners.
>
>
> > A minor point that would be interesting is how you deal with the fact that the SMF might reject rephrasing misinformation (as indicated in lines 138-140). What does this mean for SMF as a pipeline? Do you disregard these claims?
>
>
> Thank you for pointing this out, and we apologize for the confusion in the original phrasing. We would like to clarify that SMF does not reject rephrasing of misinformation. Instead, the refusals observed during implicit injection—particularly for age and gender prompts—are due to LLMs responding with messages such as “I cannot rewrite the passage as it contains misinformation.” This behavior reflects the safety alignment of the LLMs, which in some cases block generation when misinformation is detected, regardless of the stylistic prompt.
>
> To assess how these refusals influence our results, we quantified the number of affected instances:
>
>
> |Dataset|All Instances|Refused Instances (age)|Refused Instances (gender)|
> |---|---|---|---|
> |RumourEval| 315 | 90 | 79|
> |Pheme| 1000 | 283 | 236|
> |Twitter15| 507 | 160 | 151|
> |Twitter16| 319 | 127 | 122|
> |Celebrity| 500 | 28 | 33|
> |FakeNews| 480 | 59 | 55|
> |Politifact| 368 | 136 | 128|
> |Gossipcop| 1000 | 39 | 39|
> |Tianchi| 1000 | 296 | 293|
> |MultiLingual| 1000 | 193 | 203|
> |AntiVax| 1000 | 394 | 348|
> |COCO| 1000 | 589 | 552|
> |Kaggle1| 1000 | 191 | 195|
> |Kaggle2| 1000 | 215 | 196|
> |NQ-Misinfo| 1000 | 12 | 14|
> |Streaming-Misinfo| 1000 | 46 | 58|
>
> Since these refusals could skew comparisons, we filtered out the refused samples as well as their original counterparts. The following results show that removing these instances does not significantly alter the conclusions:
>
> ||Mistralv3|Llama3|BERT|DeBERTa|CMTR|DisC|CATCH
> |---|---|---|---|---|---|---|---|
> |Original Age|55.0|57.8|78.2|78.2|77.7|79.1|79.3
> |Shortcut Age|56.9|56.6|74.9|74.9|75.2|75.5|76.4
> |Original Gender|55.6|57.3|78.0|78.8|77.0|77.6|78.6
> |Shortcut Gender|55.6|56.1|74.8|74.5|74.0|74.8|76.8
>
>
> These results support our observation: detectors are less affected by implicit injections (Line 219), and the presence or absence of these blocked samples does not change the broader findings. The refusal behavior by LLMs also serves as an important caution: many current LLMs are designed to avoid rewriting misinformation, even when prompted for stylistic variation. This safety-aligned behavior can affect the completeness of data augmentation pipelines and should be taken into account when applying or interpreting methods like SMF in practice.

---

### Official Review · Reviewer_4cY3 · 2025-07-03

**Clarity:** 4
**Significance:** 3
**Originality:** 3
**Rating:** 5
**Confidence:** 4

**Summary:**

This paper presents a framework designed to evaluate whether misinformation detection models rely on spurious correlations (such as sentiment, style, or topic) instead of factual content. The framework tests models in two settings: intrinsic shortcut induction, where natural biases in existing datasets are isolated, and extrinsic shortcut injection, where LLMs are used to rewrite content with specific superficial features to simulate adversarial attacks. The authors show that many models degrade significantly under these conditions, revealing a strong reliance on shortcuts. To address this, they propose SMF, an LLM-based data augmentation method that removes shortcut cues through paraphrasing, summarization, or style normalization. Experiments across 16 datasets demonstrate that SMF improves model robustness and encourages more reliable detection of misinformation across varied contexts.

**Questions:**

- Referring to lines 136-140, how did you handle the cases with (1)low agreement scores in implicit injection and (2) LLM refusal ?



### Minor  things:
- The expansion of SMF is not mentioned in the first use of the term in abstract or introduction

**Ethical Concerns:**

["NO or VERY MINOR ethics concerns only"]

**Final Justification:**

Authors have provided clarifications for the minor concerns I raised in the review. I will keep the score provided.

**Limitations:**

Yes. Included in the appendix

**Paper Formatting Concerns:**

-

**Quality:**

3

**Strengths And Weaknesses:**

### Strengths:

- The research is very well motivated, very well written and easy to follow. Probably also easy to replicate given the simplistic nature.
- The novel contributions of the paper include the TRUTHOVERTRICKS framework, two factual misinformation benchmarks and a lightweight mitigation approach that improves model robustness by rewriting training data to reduce reliance on superficial features.
- A systematic evaluation of shortcuts is conducted by looking into the surface level features in existing datasets as well as by injecting shortcuts using LLMs. The method is quite intuitive and can easily be adapted to other tasks like deception detection.
- Extensive evaluation is conducted across datasets and models.


### Weaknesses:
- The paper discusses 4 shortcut types which are well evaluated and intuitive. However there is no discussion of other potential shortcuts like reliance on entities or persuasion strategies.
- It is unclear from the main text how the failure in implicit injection was handled and how this influences the overall study.

---

> ### Author Rebuttal · Authors · 2025-07-30
>
> We thank the reviewer for the thoughtful observation and for recognizing the strengths of our work, including its clear motivation, systematic evaluation framework, and extensive experimentation.
>
> > The paper discusses 4 shortcut types which are well evaluated and intuitive. However there is no discussion of other potential shortcuts like reliance on entities or persuasion strategies.
>
> We agree that shortcut learning in misinformation detection is a complex and multifaceted issue, especially in dynamic news environments. It is indeed not feasible to exhaustively cover all possible shortcut types within a single study. In this work, our aim was to establish a generalizable and extensible evaluation framework—TruthOverTricks—that enables both intrinsic and extrinsic analyses of shortcut behavior. To demonstrate its utility, we selected a set of representative shortcuts (four intrinsic and six extrinsic) and evaluated them across 16 benchmarks.
>
> Our findings show that existing detectors are consistently vulnerable to these diverse shortcut types, highlighting the need for more robust, semantically grounded detection strategies. Importantly, the modular design of TruthOverTricks allows for straightforward extension to other shortcut dimensions, such as entity overreliance or persuasive linguistic strategies, as suggested by the reviewer.
>
> We appreciate this constructive suggestion and will add a discussion in the Conclusion to highlight these promising directions for future work. We hope that our framework will serve as a foundation for further studies exploring and mitigating additional forms of shortcut learning in misinformation detection.
>
>
> > It is unclear from the main text how the failure in implicit injection was handled and how this influences the overall study.
>
> > Referring to lines 136-140, how did you handle the cases with (1) low agreement scores in implicit injection and (2) LLM refusal?
>
>
> (1) For low agreement scores in implicit injection:
> We acknowledge that human evaluators reported relatively low agreement when assessing whether the injected content reflected implicit attributes such as age or gender. Rather than viewing this as a limitation of the injection process, we interpret it more as a reflection of the subtle nature of these attributes in real-world news. Unlike explicit stylistic cues, implicit user attributes often manifest in more understated ways (e.g., through tone, reference framing, or content prioritization) which may not be easily distinguishable in isolated articles. This subtlety is, in fact, realistic and worth investigating, as it mirrors how news is adapted for different audiences in practice. The modest performance changes observed in Table 2 under implicit injection further support this view: detectors show limited response to these subtle changes, suggesting that current models may not pick up on subtle shifts in audience-targeted writing.  TruthOverTricks includes these cases to highlight the boundaries of model sensitivity and to motivate further research on better detecting and reasoning over implicit audience targeting.
>
> (2) For LLM refusals:
> In our initial experimental design, we retained instances even when the LLMs partially or fully refused to follow the injection prompt. This choice preserved label distributions and ensured a consistent dataset size across original and shortcut conditions, allowing for a direct comparison.
>
> To rigorously assess the influence of these refusals, we conducted an additional analysis. We filtered out all instances in which the LLM refused to perform the implicit injection. To maintain fairness, we also removed the corresponding original (non-injected) instances, ensuring pairwise alignment. The number of refusals per dataset is shown below:
>
> |Dataset|All Instances|Refused Instances (age)|Refused Instances (gender)|
> |---|---|---|---|
> |RumourEval| 315 | 90 | 79|
> |Pheme| 1000 | 283 | 236|
> |Twitter15| 507 | 160 | 151|
> |Twitter16| 319 | 127 | 122|
> |Celebrity| 500 | 28 | 33|
> |FakeNews| 480 | 59 | 55|
> |Politifact| 368 | 136 | 128|
> |Gossipcop| 1000 | 39 | 39|
> |Tianchi| 1000 | 296 | 293|
> |MultiLingual| 1000 | 193 | 203|
> |AntiVax| 1000 | 394 | 348|
> |COCO| 1000 | 589 | 552|
> |Kaggle1| 1000 | 191 | 195|
> |Kaggle2| 1000 | 215 | 196|
> |NQ-Misinfo| 1000 | 12 | 14|
> |Streaming-Misinfo| 1000 | 46 | 58|
>
> After re-running the experiments on these filtered datasets, we observed the following detector performance:
>
> ||Mistralv3|Llama3|BERT|DeBERTa|CMTR|DisC|CATCH
> |---|---|---|---|---|---|---|---|
> |Original Age|55.0|57.8|78.2|78.2|77.7|79.1|79.3
> |Shortcut Age|56.9|56.6|74.9|74.9|75.2|75.5|76.4
> |Original Gender|55.6|57.3|78.0|78.8|77.0|77.6|78.6
> |Shortcut Gender|55.6|56.1|74.8|74.5|74.0|74.8|76.8
>
>
> The results are consistent with our observation: detectors are generally less affected by implicit injection (Line 219). This further supports our interpretation that such injections are more subtle in nature, and that current detectors are not easily influenced by these minor shifts in writing style or framing. These findings strengthen our understanding of the limits of shortcut learning in subtle, implicit contexts and highlight the importance of continuing to explore such dimensions in future work.
>
> We will include this extended analysis in the revised manuscript to clarify how implicit injection limitations were handled and their impact on our conclusions.
>
> > The expansion of SMF is not mentioned in the first use of the term in abstract or introduction
>
> Thanks for your advice. We will include the complete name of SMF in the Abstract and Introduction to enhance clarity.

---

> > ### Comment · Reviewer_4cY3 · 2025-08-05
> >
> > I appreciate the reviewers for the meticulous responses to each comment and I stay confident in the already assigned score for the paper.

---

> > > ### Author Response · Authors · 2025-08-06
> > >
> > > Thank you again for your insightful reviews and support for our work. I believe this work will be much stronger with discussions on the scalability of TruthOverTricks and results related to the refusal when employing implicit rephrasing.

---

### Official Review · Reviewer_RFGe · 2025-07-03

**Clarity:** 3
**Significance:** 2
**Originality:** 3
**Rating:** 4
**Confidence:** 4

**Summary:**

This paper introduces TRUTHOVERTRICKS, a comprehensive evaluation framework for measuring and mitigating shortcut learning in misinformation detection systems. The authors identify that current misinformation detectors often rely on superficial cues (shortcuts) like sentiment, style, topic, and perplexity rather than genuine content authenticity. They evaluate seven detectors across 16 datasets, create two new factual misinformation datasets, and propose SMF (Shortcut Mitigation Framework) - an LLM-based data augmentation approach to improve detector robustness.

**Questions:**

How do you justify classifying sentiment, style, topic, and perplexity as "shortcuts" when extensive prior work has demonstrated their legitimate utility for misinformation detection?

To what extent do your experimental setups reflect realistic deployment scenarios where detectors would encounter misinformation?

Under what conditions should practitioners apply your SMF framework, and when might it actually hurt performance? Since you're essentially removing potentially useful features, how do practitioners determine whether their specific use case suffers from harmful shortcut learning versus beneficial feature utilization?

**Ethical Concerns:**

["NO or VERY MINOR ethics concerns only"]

**Final Justification:**

The authors' responses clarified some of my previous concerns, especially for the framing about "Shortcut."

**Limitations:**

Yes.

**Paper Formatting Concerns:**

No.

**Quality:**

2

**Strengths And Weaknesses:**

**Strengths:**

* Novel Factual Datasets: The creation of NQ-Misinfo and Streaming-Misinfo datasets that require factual knowledge for detection fills an important gap in existing benchmarks.

* Practical Solution: The SMF framework offers a model-agnostic approach to improving detector robustness through data augmentation rather than complex model modifications.

* Strong Empirical Results: The paper demonstrates significant performance drops when detectors encounter distributional shifts, and shows SMF can improve robustness substantially.

**Weaknesses:**

* Problematic "Shortcut" Framing: The core conceptual issue is treating established, validated features (sentiment, style, topic, perplexity) as "shortcuts" when prior work [1] [2] have demonstrated their legitimate utility for misinformation detection. The paper lacks adequate justification for why these should be considered spurious rather than valid but insufficient features.

* Artificial Evaluation Setup: The experimental design creates extreme distributional mismatches that may not reflect realistic deployment scenarios. The performance drops might indicate normal sensitivity to distribution shift rather than problematic shortcut learning.

* Missing Causal Analysis: The paper provides insufficient evidence that the studied features lack causal relationships with misinformation, which is necessary to classify them as "shortcuts."

[1] Cui, Limeng, Suhang Wang, and Dongwon Lee. "Same: sentiment-aware multi-modal embedding for detecting fake news." In Proceedings of the 2019 IEEE/ACM international conference on advances in social networks analysis and mining, pp. 41-48. 2019.

[2] Przybyla, Piotr. "Capturing the style of fake news." In Proceedings of the AAAI conference on artificial intelligence, vol. 34, no. 01, pp. 490-497. 2020.

---

> ### Author Rebuttal · Authors · 2025-07-30
>
> Thank you for your valuable feedback and your efforts in helping enhance our framing. We are encouraged that you recognize our contributions in developing complementary misinformation detection datasets, proposing a model-agnostic SMF approach, and demonstrating strong empirical results.
>
> > Problematic "Shortcut" Framing: The core conceptual issue is treating established, validated features (sentiment, style, topic, perplexity) as "shortcuts" when prior work [1] [2] have demonstrated their legitimate utility for misinformation detection. The paper lacks adequate justification for why these should be considered spurious rather than valid but insufficient features.
> > How do you justify classifying sentiment, style, topic, and perplexity as "shortcuts" when extensive prior work has demonstrated their legitimate utility for misinformation detection?
>
> We acknowledge that features such as sentiment and style have been shown in prior work [1,2,3] to be useful. However, our concern lies not in the utility of these features but in their limitations under distributional shifts, particularly in dynamic real-world scenarios.
>
> Recent studies indicate that the boundaries between fake and real news are increasingly blurred. Real news has adopted more subjective and engaging tones [4], while fake news producers strategically mimic credible news sources to evade detection [5]. As shown in Figure 5, the distributions of these features (e.g., sentiment, style) overlap significantly between fake and real news, reducing their discriminative reliability. This suggests that detectors heavily relying on such superficial cues may learn dataset-specific correlations that do not generalize.
>
> Furthermore, recent work demonstrates that small stylistic edits (without altering factual content) can substantially degrade detector performance [6], highlighting a vulnerability. Motivated by this, we propose TruthOverTricks, a comprehensive evaluation framework to systematically assess the influence of these features. As shown in Tables 1 and 2, detectors often fail when confronted with style-shifted inputs, which supports our claim that these features can function as shortcuts—in the sense defined by Geirhos et al. [7]—that lead to spurious correlations rather than robust reasoning about news veracity.
>
> By framing them as potential shortcuts, we do not dismiss their utility outright, but rather emphasize the importance of understanding when and how they might mislead models. Our work aims to identify such vulnerabilities and assist in the development of detectors that are more resilient to these shifts.
>
> [1] Cui, Limeng, et al. "Same: sentiment-aware multi-modal embedding for detecting fake news." In Proceedings of the 2019 IEEE/ACM international conference on advances in social networks analysis and mining. 2019.
>
> [2] Przybyla, Piotr. "Capturing the style of fake news." AAAI. 2020.
>
> [3] Zhang, Xueyao, et al. "Mining dual emotion for fake news detection." TheWebConf. 2021.
>
> [4] Da San Martino, Giovanni, et al. "Fine-Grained Analysis of Propaganda in News Article." EMNLP. 2019.
>
> [5] Shu, Kai, et al. "Beyond news contents: The role of social context for fake news detection." Proceedings of the twelfth ACM international conference on web search and data mining. 2019.
>
> [6] Wu, Jiaying, Jiafeng Guo, and Bryan Hooi. "Fake news in sheep's clothing: Robust fake news detection against LLM-empowered style attacks." SIGKDD. 2024.
>
> [7] Geirhos, Robert, et al. "Shortcut learning in deep neural networks." Nature Machine Intelligence.
>
> > Artificial Evaluation Setup: The experimental design creates extreme distributional mismatches that may not reflect realistic deployment scenarios. The performance drops might indicate normal sensitivity to distribution shift rather than problematic shortcut learning.
>
> > To what extent do your experimental setups reflect realistic deployment scenarios where detectors would encounter misinformation?
>
> We appreciate this opportunity to help resolve your concern.
>
> Our intent is not to create artificially adversarial conditions, but rather to reflect the diversity and unpredictability inherent in real-world misinformation. In practice, detectors are deployed in dynamic media environments where stylistic, topical, and linguistic variations are common, often intentionally manipulated by malicious actors to evade detection.
>
> Figures 10 and 11 demonstrate that existing detectors struggle particularly in cross-dataset and knowledge-intensive scenarios, suggesting limited generalization and a reliance on surface-level correlations. These patterns are indicative of shortcut learning—where models latch onto spurious but predictive features instead of developing a genuine understanding of factuality.
>
> To probe this, TruthOverTricks simulates distributional shifts in both intrinsic and extrinsic dimensions. Importantly, we do not alter label distributions. In intrinsic settings, the test set remains unchanged, while in extrinsic settings, both metric- and human-based evaluations (lines 128–250) confirm that factual authenticity is preserved despite changes in style or expression. Nonetheless, the performance of all seven baseline detectors drops significantly across 16 benchmarks. This indicates that their predictions are influenced more by correlations with non-semantic features than by core content.
>
> In this sense, our evaluation setup mirrors real-world deployment risks: where misinformation often appears in novel styles, tones, or contexts. By revealing vulnerabilities under such shifts, our framework serves as a diagnostic tool to better understand model behavior and improve robustness.
>
> > Missing Causal Analysis: The paper provides insufficient evidence that the studied features lack causal relationships with misinformation, which is necessary to classify them as "shortcuts."
>
> We would like to reinforce that our goal is not to argue that features such as sentiment or style are entirely irrelevant or should be excluded from misinformation detection. Rather, our objective is to evaluate the extent to which detectors may over-rely on these features, especially in contexts where they do not reliably indicate factuality.
>
> We fully acknowledge that establishing a lack of causal relationship requires more than observing performance drops. However, TruthOverTricks is designed to empirically assess sensitivity to distributional shifts in these features, without altering ground-truth labels. When detector performance degrades significantly under these controlled interventions—despite the underlying factual content remaining constant (as verified by human and metric-based evaluation)—this suggests that the model is not robustly grounded in semantic understanding. While this does not constitute formal causal analysis, it provides practical evidence of shortcut-like behavior, following the definition from Geirhos et al. [1].
>
> Importantly, our TruthOverTricks framework is diagnostic, not prescriptive: it reveals potential model vulnerabilities and encourages the development of detectors that rely more heavily on core semantic content rather than superficial correlations. In this way, **we aim to contextualize the role of stylistic and linguistic features under realistic distribution shifts--not to dismiss their potential utility.**
>
> [1] Geirhos, Robert, et al. "Shortcut learning in deep neural networks." Nature Machine Intelligence.
>
> > Under what conditions should practitioners apply your SMF framework, and when might it actually hurt performance? Since you're essentially removing potentially useful features, how do practitioners determine whether their specific use case suffers from harmful shortcut learning versus beneficial feature utilization?
>
> The SMF framework is intended as a targeted intervention to address situations where models exhibit signs of shortcut learning—particularly when performance fails to generalize.
>
> One practical indicator is a substantial accuracy gap between the validation and test sets, as shown in Figure 6. Such a gap suggests that the model may be overfitting to spurious correlations in the training data, rather than learning generalizable patterns of factuality. In these cases, applying SMF to augment the training set can encourage the model to rely less on superficial features and more on deeper semantic cues.
>
> Practitioners should consider applying SMF when: (1) There is evidence of generalization failure (e.g., large validation–test performance gaps), (2) Input distribution shifts are expected across deployment settings (e.g., varying writing styles or user demographics), or (3) Robustness to adversarial manipulation or stylistic variation is critical.
>
> Conversely, in cases where such shifts are unlikely or when domain-specific stylistic patterns are known to correlate strongly with misinformation, applying SMF may be unnecessary. For example, in a tightly controlled domain such as scientific misinformation detection within biomedical literature, stylistic signals like hedging language or citation patterns may carry genuine predictive value. In such settings, these features may reflect valid, domain-specific cues rather than spurious correlations.
>
> To balance these considerations, we recommend a two-step approach: (1) pre-train the detector with SMF-enhanced data to improve robustness, and (2) use the SMF-enhanced detector to evaluate individual cases or deployment settings where shortcut learning may be suspected. This offers a practical pathway to detect and mitigate shortcut reliance while preserving beneficial feature utilization when appropriate.
>
> Thanks again for your thought-intriguing comments. We will incorporate this discussion to further refine our work.

---

### Official Review · Reviewer_TvfK · 2025-07-06

**Clarity:** 2
**Significance:** 3
**Originality:** 3
**Rating:** 5
**Confidence:** 3

**Summary:**

Authors investigate the significance of shortcut learning in detecting misinformation in LLMs, mid-sized LMs, and ad hoc methods. The define two groups of potential shortcuts: intrinsic and extrinsic. Intrinsic includes sentiment, style, topic, and perplexity; extrinsic includes rewriting, explicit, and implicit.

For the intrinsic shortcuts, given 16 datasets (including 2 repurposed datasets) they selectively form training and test sets and show that the mentioned detection method significantly rely on the discussed shortcut to make their prediction. For the extrinsic shortcuts, they use an LLM and create synthetic sets for the mentioned datasets, and again reach the same conclusion.

Finally to resolve the problem, they suggest that before training the detection methods, the data be pre-processed to remove the shortcuts. This method alleviate the problem to some degree--the argue that LLM based methods don't need this, and they are already robust.

**Questions:**

None

**Ethical Concerns:**

["NO or VERY MINOR ethics concerns only"]

**Final Justification:**

Increased from 4 to 5.

**Limitations:**

yes

**Quality:**

3

**Strengths And Weaknesses:**

**Strengths:**

-  The experiments are very detailed and I think should be straightforward to reproduce
-  The topic is very timely, and can help related practitioners
-  The method is straightforward and intuitive
-  The improvements are good

**Weaknesses:**

- It is almost impossible to be able to read the paper without reading the appendix, a lot of material is moved to Appendix, and there are a lot of cases that author refer to the tables and discussions in the appendix (just a quick search in the body of the paper confirms this). This has made me to seriously think that this is not the right paper for a conference, perhaps a journal would be a better fit
- I don't think that Figure 4 is the right way to calculate authenticity.  Both metrics are very old methods for this and are not used anymore. There are more recent works, like those that measure hallucination (like FactScore or other suitable metrics).

---

> ### Author Rebuttal · Authors · 2025-07-30
>
> Thanks for your valuable feedback and for recognizing the timeliness of our topic, the comprehensiveness of our empirical experiments, and the effectiveness of our proposed approach.
>
> > It is almost impossible to be able to read the paper without reading the appendix, a lot of material is moved to Appendix, and there are a lot of cases that author refer to the tables and discussions in the appendix (just a quick search in the body of the paper confirms this). This has made me to seriously think that this is not the right paper for a conference, perhaps a journal would be a better fit.
>
>
> We would like first to thank the appreciation of the comprehensiveness of our experiments. To ensure robustness, we evaluate seven baselines across 16 benchmarks, which inevitably results in a substantial volume of results. While we strive to keep the main text focused and accessible, **we have carefully ensured that all key findings are clearly supported within the main body.** Specifically:
>
> (1) LLMs can inject predefined factors without compromising content authenticity (Figures 3 and 4);
>
> (2) Existing datasets contain spurious correlations unrelated to authenticity (Figure 5); and
>
> (3) Misinformation detectors tend to rely on learned shortcuts (Figures 6–8), leading to performance degradation under varied news expressions (Tables 1–2).
>
> The appendix serves to provide full experimental details and complete per-dataset results to support reproducibility and transparency. For instance, while Table 2 summarizes average performance across benchmarks, detailed per-dataset results are deferred to Tables 15 and 16 in the appendix for space considerations. We have made sure that all necessary information to follow the paper’s key contributions remains self-contained in the main text. Thanks for your comment, we will further review the balance and reduce dependence on the appendix where possible to improve clarity and alignment with conference constraints.
>
>
>
>
> > I don't think that Figure 4 is the right way to calculate authenticity. Both metrics are very old methods for this and are not used anymore. There are more recent works, like those that measure hallucination (like FactScore or other suitable metrics).
>
>
> Our primary goal in this figure is to verify that the rewritten news variants injected with extrinsic shortcuts in TruthOverTricks preserve the original article’s factual label -- a related yet more specific focus than general hallucination detection. To this end, we initially employed ROUGE-L and BERTScore to measure token-level and semantic-level similarity. While these metrics are indeed older, they remain widely adopted for such comparative analyses and, in our case, provided a meaningful signal—particularly when contrasted with random-shuffle baselines.
>
> To further strengthen our claim, we conducted a human evaluation to validate data quality (see Lines 146-148), achieving an average accuracy of 91.2% and a Fleiss’ Kappa of 0.838. This indicates strong agreement that the injection did not alter the factual label.
>
> In response to the reviewer’s suggestion, we additionally adopted an LLM-based evaluation approach similar in spirit to FactScore [1]. Specifically, we prompted GPT-4-turbo as follows:
>
> “Text 1: {Original article}
>
> Text 2: {Injected article}
>
> Do these two texts describe the same event? Please answer yes or no.”
>
>
> The percentage of “yes” responses across categories is summarized below:
>
> |Pairs|Percentage|
> |---|---|
> |random|0.12|
> |rewriting|0.92|
> |paraphrase|0.88|
> |open-ended|0.98|
> |positive|0.80|
> |negative|0.82|
> |simple|0.96|
> |complex|0.94|
> |formal|0.96|
> |informal|0.90|
> |young|0.98|
> |elder|0.96|
> |male|0.94|
> |female|0.98|
>
> The results present that injection does not alter the labels evaluated by LLMs. Notably that LLMs judge that 12% of random shuffle pairs describe the same event, which shows that LLM-as-evaluator might not be reliable in this task. It coincides with the idea of utilizing metric measures and human evaluation.
>
> We believe this work will be much stronger after the discussion of the LLM-as-evaluator.
>
>
> [1] Min, Sewon, et al. "FActScore: Fine-grained Atomic Evaluation of Factual Precision in Long Form Text Generation." The 2023 Conference on Empirical Methods in Natural Language Processing.

---

### Decision · Program_Chairs · 2025-09-17

**Decision:**

Accept (poster)

**Comment:**

This paper presents TruthOverTricks framework for diagnosing and mitigating shortcut learning in misinformation detection. It evaluates several detectors across common benchmark datasets and shows that models often rely on superficial cues such as sentiment, style, and topic, leading to poor generalization under distribution shifts. To address this, they propose an LLM-based data augmentation method that improves robustness by reducing reliance on such cues. Reviewers found the work timely, well-motivated, and empirically thorough, with strong practical relevance and reproducibility. The rebuttal provided valuable clarifications that strengthen the paper and will be important for the camera-ready version. The paper would further benefit from stronger grounding in the existing literature, particularly recent work on LLM-based misinformation detection and studies discussing dataset limitations and spurious correlations in misinformation benchmarks.